# Multi-Scale Image Diffusion Transformers: Explainability Leads to Faster Training

## Abstract

Diffusion models have significantly advanced image synthesis but often face high computational demands and slow convergence rates during training. To tackle these challenges, we propose the Multi-Scale Diffusion Transformer (MDiT), which incorporates heterogeneous, asymmetric, scale-specific transformer blocks to reintroduce explicit inductive structural biases into diffusion transformers (DiTs). Using explainable AI techniques, we demonstrate that DiTs inherently learn these biases, exhibiting distinct encode-decode behaviors, effectively functioning as semantic autoencoders. Our optimized MDiT architecture leverages this understanding to achieve a $\geq 3\times$ increase in convergence speed on FFHQ-256x256 and ImageNet-256x256, culminating in a $7\times$ training speedup on ImageNet compared with state-of-the-art models. This acceleration significantly reduces the computational requirements for training, measured in FLOPs, enabling more efficient resource use and enhancing performance on smaller datasets. Additionally, we develop a variance matching regularization technique to correct sample variance discrepancies which can occur in latent diffusion models, enhancing image contrast and vibrancy, and further accelerating convergence.

## 1 Introduction

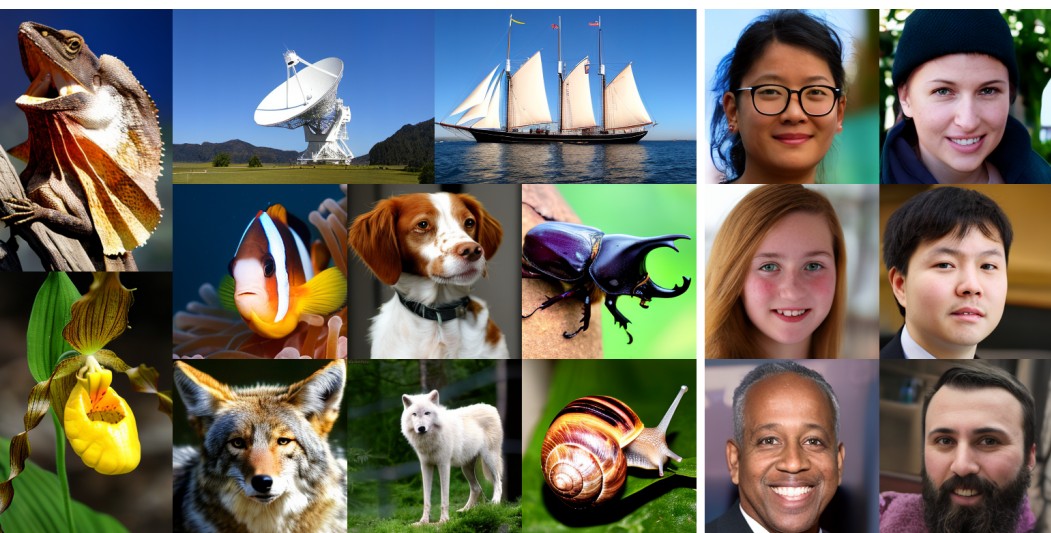

Figure 1: Generated Samples on ImageNet 256x256 (left) and FFHQ 256x256 (right), with $12.5\times$ and $3.2\times$ fewer training FLOPS than comparable diffusion models. Best viewed zoomed in.

The advent of diffusion-based generative models has significantly advanced the field of image synthesis. Models such as Imagen (Saharia et al., 2022), Stable Diffusion (Rombach et al., 2021), and DALL-E 2 (Ramesh et al., 2022) have set new benchmarks by leveraging the robustness of U-Net Convolutional Neural Network (CNN)-based architectures (Ronneberger et al., 2015), which are particularly effective for capturing multi-scale detail. Meanwhile, transformer-based approaches like

DiT (Peebles & Xie, 2022), DiffiT (Hatamizadeh et al., 2023), and SD3 (Esser et al., 2024) have since surpassed their CNN-based counterparts in both efficiency and in capturing complex dependencies within image data. However, despite their operational efficiency, transformer-based models often exhibit slower convergence, which necessitates extensive training iterations (Dosovitskiy et al., 2021; Chen et al., 2024). This significant computational expenditure constrains their accessibility within the research community and for smaller organizations, limiting their uptake and slowing innovation.

A key advantage of the shift towards diffusion transformers (DiTs) has been the elimination of inductive biases (Peebles & Xie, 2022) inherent in CNN-UNets, resulting in a simpler, *homogeneous* network structure. However, it is well-documented that images inherently possess three fundamental properties: translation invariance, locality, and multi-scale features. The absence of architectural structures that enforce these biases in vision transformers (ViTs) necessitates *implicitly* learning these properties (Ben-Shaul et al., 2023; Raghu et al., 2021), which may incur unnecessary computational overhead and limit model capacity. Reintroducing these biases into ViTs, therefore, has been shown to enhance performance relative to computational cost (Liu et al., 2021; Hassani et al., 2023).

Consequently, this paper poses two pivotal questions: 1) Do DiTs similarly exhibit this *implicitly* learned behavior as observed in ViTs? and 2) Can such biases be *explicitly* reintroduced to diffusion transformers while maintaining their generality and enhancing training efficiency?

In the rest of the paper we primarily focus on the impact of transformer network architecture on DiTs, distinct from algorithmic improvements. We utilize the latent space Min-SNR weighting strategy (Hang et al., 2023), with $x_0$ prediction - a training objective where $x_0$ represents the original, clean latent data sample in diffusion processes. This approach offers a consistent prediction target across diffusion timesteps and facilitates direct classification probe training at $t = 0$, where the network is predominantly engaged in a reconstruction task. The training efficiency gains are thus *compounding with* the enhanced convergence provided by Min-SNR. Finally, we introduce a regularization term that improves image contrast and vibrancy when training with Min-SNR, which is particularly impactful for unconditional models that cannot leverage classifier-free guidance (Ho & Salimans, 2021).

Our main contributions are as follows:

- We propose a heterogeneous multi-scale diffusion transformer architecture (MDiT), employing distinct transformer blocks for image feature processing, achieving enhanced detail capture earlier in training and accelerating convergence by $3.47\times$ on ImageNet-256.

- We develop an explainability framework for the MDiT architecture by employing partial-head rotary position embeddings, inspired by GPT-J (Wang & Komatsuzaki, 2021), and layer-wise classification probes, which we use to explain the depth-wise functional behavior of diffusion transformers and further optimize our architecture for enhanced image synthesis.

- We introduce a variance matching regularization technique, which corrects a sample variance discrepancy with latent diffusion models trained with Min-SNR, improving image contrast and vibrancy, and further accelerating convergence by 3% on ImageNet-256.

## 2 RELATED WORK

**Foundational Diffusion Models:** Diffusion models, introduced by Ho et al. (2020), iteratively reconstruct images from noise via a reverse diffusion process. Song et al. (2021) improved efficiency with fewer, larger steps, while Ramesh et al. (2022) introduced text conditioning for guided generation. Rombach et al. (2021) shifted diffusion to a latent space for high-resolution outputs with lower computational cost, and Podell et al. (2024) advanced control with added conditioning such as scale.

**Transformer-Based Diffusion Models:** Hoogeboom et al. (2023) replaced U-Net cores with vision transformers, reducing FLOPS significantly. Peebles & Xie (2022) introduced Diffusion Transformers (DiTs), utilizing vision transformers throughout for efficient scaling. Crowson et al. (2024) further adapted DiTs for image space, implementing a U-Net-like structure with nested patch embeddings. Other advances include enhanced self-attention (Hatamizadeh et al., 2023), multi-modal transformers (Esser et al., 2024), and integrating Mixture of Experts (Xue et al., 2023).

**Efficiency Enhancements in Training Diffusion Models:** Various methods have been developed to reduce the training costs of diffusion models. These include progressive training strategies (Chen

et al., 2024), loss scaling based on signal-to-noise ratio (Hang et al., 2023), alternative training objectives (Dao et al., 2023; Ma et al., 2024), sub-image patch training (Wang et al., 2023), and salient feature patch masking techniques (Sehwag et al., 2024).

# 3 A MULTI-SCALE HETEROGENEOUS DIFFUSION TRANSFORMER

Diffusion transformers (DiTs) have demonstrated widespread success across generative modeling tasks, excelling in producing high-quality outputs. However, their architectural rigidity poses several limitations, including inefficiencies in parameter utilization (Crowson et al., 2024), challenges with multi-scale feature representation due to their isotropic nature, and difficulties in adapting to diverse modalities such as text conditioning and zero-shot aspect ratio changes (Chen et al., 2024). While prior works have addressed subsets of these issues, we propose the Multi-scale Diffusion Transformer (MDiT) to tackle them holistically by reintroducing inductive biases, improving parameter efficiency, and enhancing flexibility. This architecture serves as a testbed to explore whether explicitly reintroducing such biases can enhance the generality and training efficiency of diffusion transformers.

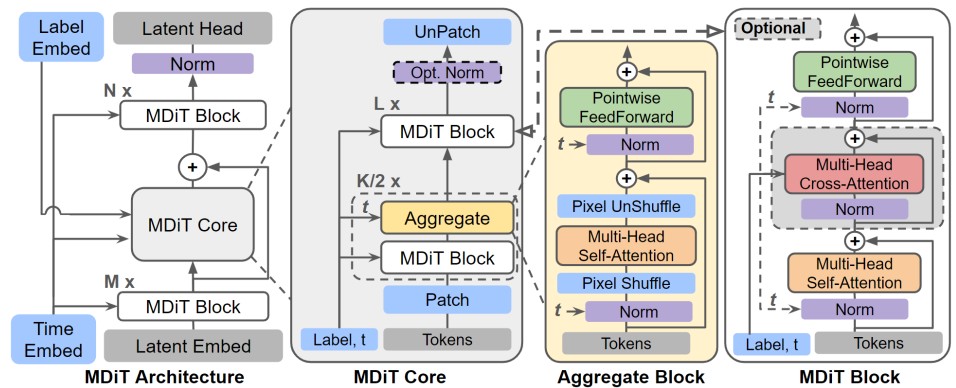

Figure 2: MDiT multi-scale architecture showing the hierarchical structure from left to right.

**Key Architectural Contributions:** MDiT introduces two key innovations: a shallow U-Net-like structure and aggregate blocks within the core. The shallow U-Net design reduces the hierarchy to two levels, reintroducing the inductive bias of scale and decreasing the parameter overhead associated with deeper U-Net hierarchies. Unlike typical diffusion transformers that use a 2x2 patch embedding, MDiT incorporates a 1x1 point-wise patch embedding in the outer U-Net level, enabling fine-grained feature processing at the full latent resolution. Aggregate blocks within the core complement this by efficiently capturing a third feature scale, performing down-sampling within the attention layers to bridge spatial representations without the additional overhead of deeper U-Net levels.

**Architecture Overview:** As depicted in Figure 2, MDiT is structured with two levels: an outer level and a core, connected by a skip connection that treats the core as a "macro block." The outer level processes features at the full latent resolution, using $M$ and $N$ blocks before and after the core, respectively. The core operates at a 2x downsampled spatial resolution, where aggregate blocks alternate with MDiT blocks, parameterized by $K$, followed by a stack of $L$ additional MDiT blocks. The parameter set $\{M, N, K, L\}$ enables heterogeneous configurations while also providing coverage with the isotropic case $\{0, 0, 0, L\}$ used in DiTs. This equivalence enables controlled experiments to evaluate whether isotropic DiTs implicitly learn the spatial inductive properties of images.

**Hybrid Conditioning Scheme:** To support flexible conditioning modalities, including text, MDiT employs a hybrid conditioning scheme. Cross-attention is applied within the core blocks for class conditioning, as it restricts conditioning to areas of high semantic focus while efficiently managing the $\mathcal{O}(HW)$ scaling of cross-attention. In the outer level and aggregate blocks, modulated pre-layer norm is retained, consistent with standard diffusion transformers (Peebles & Xie, 2022; Esser et al., 2024; Crowson et al., 2024). To further reduce parameter count and computational overhead in cross-attention enabled blocks, the time embedding is folded directly into an auxiliary token, replacing the need for pre-layer norm modulation in these blocks. Additional details can be found in Appendices G.1 and G.4, with text conditioning experiments on CC3M (Sharma et al., 2018) in Appendix C.

### 3.1 Augmenting the Diffusion Transformer Blocks for Improved Explainability

In developing the MDiT blocks, we follow HDiT (Crowson et al., 2024) by building upon the LLaMA style transformer blocks (Touvron et al., 2023). However, to support the explainability analysis in Section 4, our implementation differs from HDiT and LLaMA in the following two ways:

**Partial Head Axial-RoPE:** Inspired by GPT-J (Wang & Komatsuzaki, 2021), our model employs partial head Rotary Positional Embeddings (RoPE) (Su et al., 2022) to achieve 2D translation invariance by selectively applying positional embeddings to a subset of self-attention head channels. While similar to HDiT (Crowson et al., 2024), we expand upon their approach by providing an explanation for its effectiveness and limitations in Section 4. Further differing from HDiT, we utilize fixed rotary frequencies centered in the upper-left corner, rather than a normalized resolution with a centered origin, thereby allowing for easier extrapolation to arbitrary aspect ratios (see Appendix I).

**Normalization on Q and K Vectors:** We apply a layer normalization without affine scaling to the Q an K vectors in all attention layers as proposed by Dehghani et al. (2023), rather than utilizing a RMS normalization with learnable affine scaling as in Esser et al. (2024); Crowson et al. (2024). Layer norm was chosen to enforce a zero mean, placing all vectors on a unit hyper-sphere (Riechers, 2024), ensuring the attention vector L2 energy remains constant across layers – ideal for comparisons.

Notably, these changes do not significantly impact performance; Additional details in Appendix G.1.

### 3.2 Shallow U-Net: Semantic Compression and Efficient Representation

In diffusion transformers, the transition from processing low-level details to higher-level semantic information mirrors the dynamics observed in variational auto-encoders (VAEs), where data flows into and out of an internal latent space (Kingma & Welling, 2013; Esser et al., 2021). This resemblance suggests that standard patch embeddings (linear projections) in DiTs are insufficient for capturing complex semantic tasks, placing excessive demands on downstream transformer blocks. The shallow U-Net in MDiT mirrors VAE-like dynamics, with the outer level compressing features for the core and reconstructing them on the output. This approach is equivalent to replacing standard patch embeddings with increased-capacity transformer blocks, reintroducing scale-awareness while reducing the burden on the core. Empirical evidence for this interpretation is detailed in Section 4 and Appendix M.

Processing image tokens at the full latent resolution comes with an additional cost in the self-attention layers, which we overcome by adopting neighborhood self-attention (Hassani et al., 2023) in the outer blocks. This adaptation significantly reduces computational complexity from $\mathcal{O}(N^2)$ to $\mathcal{O}(Nk^2)$, with $k = 7$ strategically selected to balance FLOPS, roughly equating two outer MDiT blocks to one core MDiT block. Moreover, the combination of neighborhood attention with Axial RoPE enables scaling to larger image dimensions without additional fine-tuning of the outer blocks, while also supporting larger resolutions by adapting the patch and un-patch blocks (Appendix I & J).

### 3.3 Aggregate Blocks: Enhancing Structure at Medium Scales

Aggregate Blocks are interleaved within the MDiT core to represent medium-scale spatial features that are challenging to capture at the core's token resolution. Each block processes inputs in a 2x downsampled space using pixel shuffle, applies multi-head self-attention (MHSA), and restores the original resolution with pixel unshuffle. A point-wise feedforward network (FFN) is then applied at the input scale (Eqn.1). The FFN remains unscaled to maintain parameter efficiency, while the number of attention heads is increased by 1.5x, equivalent to scaling the hidden dimension in the down-sampling operation. Aggregate Blocks mirror the dynamics found in U-Nets, yet provide a lightweight solution for medium-scale feature representation without the overhead of introducing a third U-Net level. Parameter details are summarized in Table 1 for MDiT-B and MDiT-L configurations.

$$h = y + \text{FFN}(\text{Norm}(y)), \quad y = x + \text{UnShuffle}(\text{MHSA}(\text{Shuffle}(\text{Norm}(x)))) \tag{1}$$

To qualitatively assess the impact of the aggregate blocks, we analyzed the radial spectral power by computing the 2D FFT of the output hidden states from each core block and flattening the absolute values to a 1D diagonal. This measurement provides insight into how different configurations manage spectral energy across processing layers. As shown in Figure 3, configurations with Aggregate Blocks, such as $\{K, L\} = \{4, 5\}$, do not significantly alter the spectral energy immediately after the first or

Table 1: MDiT scaling for Base (B) and Large (L) models following DiT (Peebles & Xie, 2022).

| Model | MDiT-B | | MDiT-L | |
|---|---|---|---|---|
| Parameter | Outer | Core | Outer | Core |
| Hidden dim $d$ | 384 | 768 | 512 | 1024 |
| Head dim $d_k$ | 64 | 64 | 64 | 64 |
| Heads $h$ | 6 | 12 | 8 | 16 |
| Agg. Heads $h_A$ | – | 18 | – | 24 |

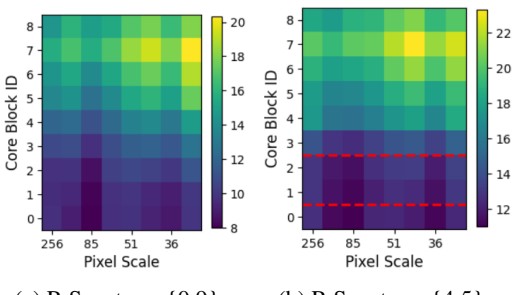

(a) R.Spectrum {0,9}.  (b) R.Spectrum {4,5}.

Figure 3: Radial spectral power of core block output activations for {K,L} at sampling step 12/25 (highest core contribution). Aggregate blocks are shown with a dashed red line, with output above.

second aggregate block. However, subsequent layers exhibit a noticeable increase in the uniformity of the spectral distribution and overall spectral energy. At the medium scale – approximately 85 pixels, or about one-third resolution – there is a clear increase in spectral energy at the final output when Aggregate Blocks are used compared to configurations without them. This suggests that while Aggregate Blocks may not directly boost spectral energy, they encode information in their outputs that subsequent MDiT blocks leverage to enhance the spectral distribution. Improved uniformity in the spectral distribution likely aids in medium-scale structure later in the sequence, enabling the model to achieve more semantically meaningful states with fewer residual updates.

### 3.4 BOOSTING FIDELITY WITH VARIANCE MATCHING REGULARIZATION

Latent diffusion models are adept at generating high-quality images; however, specific training configurations such as Min-SNR (Hang et al., 2023), can result in outputs that appear washed-out with reduced contrast. Our empirical analysis revealed deviations between the variances of generated samples and those of the true data (see Appendix H), a discrepancy that compromises the visual fidelity of the outputs. To address this issue, we introduced a variance matching regularization term to our loss function. This term aims to correct the per-sample, per-latent-channel misalignments, and enhance image quality:

$$\mathcal{L} = \mathcal{L}_{\text{MSE}} + \lambda_{\text{VAR}} \cdot \frac{1}{C} \sum_i \left| \sigma_i^2 - \hat{\sigma}_i^2 \right| \quad (2)$$

In this equation, $C$ denotes the number of latent channels, $\lambda_{\text{VAR}}$ is the loss-weighting factor, and $\sigma_i^2$ and $\hat{\sigma}_i^2$ represent the true and generated per-channel variances, respectively.

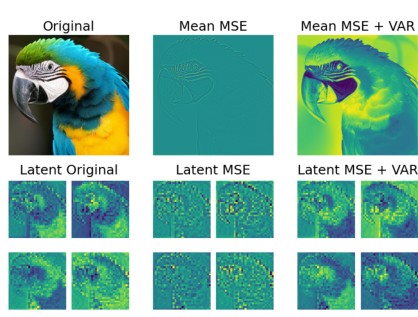

Figure 4: Gradient comparison with MSE and MSE + Variance Matching. Showing mean RGB space and latent space (4-channels) from the Stable Diffusion Variation Autoencoder (Rombach et al., 2021). Best viewed zoomed in.

In addition to correcting channel misalignment, variance matching enhances the training gradient signal by emphasizing critical features such as lighting boundaries, larger-scale details, and object edges. This broadens the impact across image scales, in contrast to the fine-detail focus of Mean Squared Error (MSE) loss. For illustration (see Figure 4), an early diffusion model prediction ($x_0$) can be simulated by blurring a ground-truth image. While MSE loss primarily highlights high-frequency errors, variance matching strengthens the gradient signal to capture a broader range of detail levels. This ensures significant visual elements receive enhanced emphasis during early training. Further extensions to rectified flows (Liu et al., 2022; Esser et al., 2024) are explored in Appendix H.2.

### 4 SEMANTIC AUTOENCODING BEHAVIOR OF DIFFUSION TRANSFORMERS

Using our MDiT framework, we observe that the DiT analog ({0,0,0,L}) inherently transitions from encoding positional to semantic information as a function of transformer depth. Interestingly, this behavior is followed by a reduction in semantic emphasis in the final blocks, mirroring the functionality of an autoencoder. This *implicitly* learned behavior suggests a natural encoding-decoding process within DiTs and has significant implications for enhancing training efficiency.

## 4.1 EXPLAINING DEPTH-WISE FOCUS WITH PARTIAL-HEAD ROPE

To enforce translation invariance in images, we utilize Axial RoPE to encode position information within our MDiT architecture. This method extends traditional RoPE (Su et al., 2022) by concatenating the 1-D embeddings in the X and Y directions of the image sequence, applying them directly to the self-attention layers (see Fig.5d). We then selectively apply Axial RoPE to a subset of feature channels within each attention-head, allowing the model to ignore positional information if needed.

**Partial Head RoPE Mechanism:** RoPE views the head features in multi-head self-attention blocks as complex numbers, where $d$ real features becomes $d/2$ complex pairs. Complex rotations, governed by RoPE coefficients $R(m) = e^{im\theta}$, are multiplied, introducing phase shifts to the vectors $q_m$ and $k_n$. This mechanism results in relative offsets of $m - n$, which reinforce translation invariance. In the case of Partial Head RoPE, we treat $\theta$ as zero for channels above a specific threshold ($r_{\dim} = d_k/4 = 16$ in our implementation), effectively bifurcating the channels into those that encode positional data and those that do not. Additionally, the behavior of these complex pairs under RoPE implies that positional information is disregarded when $x = R(m) \cdot x = 0$, allowing the magnitude of the complex pairs to serve as a measure of the encoding's contribution to position or semantic focus.

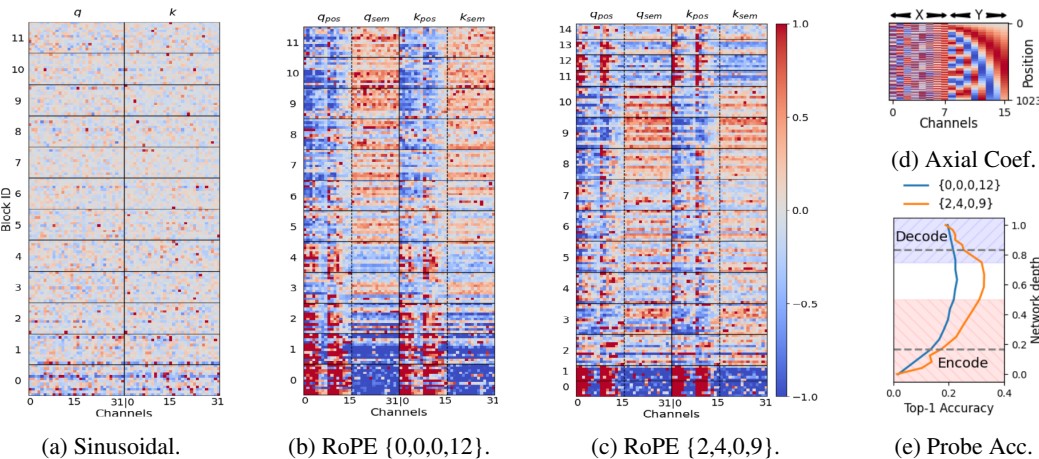

|(a) Sinusoidal. | (b) RoPE {0,0,0,12}. | (c) RoPE {2,4,0,9}. | (e) Probe Acc.|

Figure 5: (a-c) Complex magnitude ($||\cdot||^2 - 2$) of Q and K vectors of the MHSA heads for sinusoidal and RoPE position embeddings with {M,N,K,L} configurations. Red and Blue indicates strong and weak activation, respectively. (d) Axial RoPE Coefficients. (e) Block-wise probe accuracy for the models in (b) and (c), with the encode/decode region highlighted, and core within dashed lines.

**Functional Classifications:** We probe the self-attention layers with random normal activation tensors to compute a mean channel-wise complex magnitude of the Q and K vectors for each self-attention head, as illustrated in Figure 5b. Enabled by the bifurcation in channel functionality through Partial Head RoPE, distinct patterns emerge that are not observed with traditional sinusoidal embeddings, shown in Figure 5a. This leads to a per head classification into three types: *Positional focus* - characterized by weak activation above $r_{\dim}$; *Semantic focus* - noted for weak activation below $r_{\dim}$; and *Hybrid focus* - identified by moderate activation both above and below $r_{\dim}$.

**Depth-wise Behavior:** In the homogeneous configuration {0,0,0,12}, our analysis indicates that the initial blocks are predominantly position-focused, with a gradual transition to a blend of semantic and hybrid focuses in later blocks. Conversely, in the configuration {2,4,0,9}, depicted in Figure 5c, spatial encoding tasks are primarily handled by the outer blocks, enabling the core blocks to focus predominantly on semantic and hybrid processing. Additionally, some output blocks in the {2,4,0,9} configuration exhibit a hybrid focus, suggesting enhanced conditional fine-detail feature decoding.

**Impact on Capacity:** While partial-head RoPE maintains comparable performance to sinusoidal position embeddings (Appendix F), it may subtly reduce the model's capacity. This reduction is observable in Figures 5b and 5c, where certain channels demonstrate significantly weakened activation, thus limiting their contribution to the attention mechanism. Notably, this phenomenon is not evident in Figure 5a, indicating that models employing RoPE may experience a constrained number of effectively active self-attention neurons, dependent on the choice of $r_{\dim}$.

**Generalizability:** The proposed attention probe analysis extends to RoPE-based transformer models, including LLaMA (Touvron et al., 2023) and GPT-J (Wang & Komatsuzaki, 2021), though its effectiveness may be reduced in the absence of unit-normalized logits. Further insights into its application on Large Language Models and long-context fine-tuning are discussed in Appendix K.

### 4.2 MLP Probes for Semantic Analysis

In order to cross-validate the findings from our RoPE analysis, we employed classification probes as proposed by Alain & Bengio (2017), using them as an independent method to assess the semantic encoding capabilities of our MDiT architecture. We utilized two-layer MLP classifiers with an average pooling input layer, trained on the hidden state outputs from the MDiT blocks. The ImageNet-trained MDiT backbones, frozen for this task, were set to $t = 0$ (no noise) and $c = null$ (unconditional), effectively operating in an unconditional reconstruction mode, enabled by the $x_0$ training objective. Top-1 probe accuracy was then evaluated using the ImageNet validation set for both the homogeneous {0,0,0,12} and multi-scale {2,4,0,9} cases, mapping semantic encoding with network depth.

The results, illustrated in Figure 5e, offer several insights: Firstly, the top-1 accuracy curve generally peaks at approximately 60% of the network depth, highlighting the point of most effective semantic encoding. Secondly, the curve illustrates distinct "semantic encode" and "semantic decode" phases, reflective of an autoencoder's functionality. Thirdly, the multi-scale configuration achieves a significantly higher peak in accuracy, benefiting from the focus shift enabled by the outer blocks. Furthermore, there is a clear correspondence between the semantic peaks in Figure 5e and the blocks identified as highly semantic-focused in the RoPE plots (Figure 5c), especially blocks 8 and 9. This correlation validates the RoPE analysis, confirming that blocks with heightened semantic focus are indeed associated with improved semantic representations, as measured by the MLP probes.

## 5 Empirical Evaluation of MDiT Efficiencies

We adopt the Min-SNR strategy (Hang et al., 2023) setting $\gamma = 5$, to significantly accelerate training on the $x_0$ objective - where the diffusion model predicts the original, clean images (latents). Our experiments utilize the FFHQ dataset (Karras et al., 2019) for unconditional images and ImageNet (Deng et al., 2009) for conditional images, with all images standardized to a resolution of 256x256 pixels. All models are trained within the latent space of the pre-trained Variational Autoencoder from Stable Diffusion (Rombach et al., 2021), with a latent space size of $4 \times 32 \times 32$, reflecting a downsampling factor of 8 from the original image dimensions.

**Training Hyperparameters:** Consistent with the Min-SNR approach, we implement a cosine noise schedule with $t_{max} = 1000$ and employ the AdamW optimizer with a weight decay of $1 \times 10^{-2}$. Diverging from typical settings, we adjust $\beta_1$ to 0.9 and $\beta_2$ to 0.95, necessary for stability and supporting an increased constant learning rate of $4 \times 10^{-4}$ with a batch size of 256 images. Additionally, we evaluate on an Exponential Moving Average (EMA) model using a decay of 0.9999.

**Evaluation Protocol:** Model performance is assessed by generating 50k images for each checkpoint, following the protocol by Karras et al. (2019). We utilize the DDIM sampler (Song et al., 2021) for $x_0$, DDPM (Song et al., 2021) for $\epsilon$ (eps), and Euler for rectified flow (rf) objectives. Both $x_0$ and rf use with 50 and 100 steps for plots and statistical tables respectively; $\epsilon$ uses 100 or 250 steps as stated. All measurements are *without* classifier free guidance (CFG) (Ho & Salimans, 2021) unless otherwise stated. We calculate several key metrics: Fréchet Inception Distance (FID) (Heusel et al., 2017), sFID (Nash et al., 2021), Inception Score (Salimans et al., 2016), and Precision/Recall (Kynkäänniemi et al., 2019). We also calculate the DINO-FID (D-FID) score using the DINO V2-L model (Oquab et al., 2024), which Stein et al. (2023) have shown to better align with human assessments.

### 5.1 Increasing Convergence Rate

To demonstrate the efficiency of our MDiT architecture, we conducted a comparative analysis against DiT(Peebles & Xie, 2022), which serves as a homogeneous transformer baseline. Both models were trained under identical hyperparameters, using the $x_0$ objective with Min-SNR to ensure a fair comparison. The results, depicted in Figure 6, indicate significant improvements in training speed: a $3\times$ speedup on the FFHQ dataset (Fig.6a), a $4\times$ speedup on ImageNet with the B-scale model

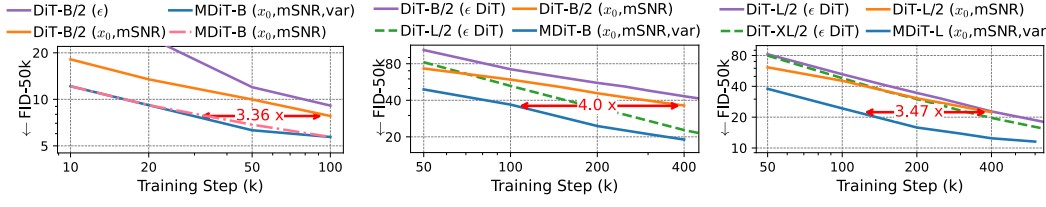

(a) FFHQ Model Convergence.  (b) B-Scale ImageNet Convergence.  (c) L-Scale ImageNet Convergence.

Figure 6: Log-Log FID-50K convergence plots for FFHQ-256 and ImageNet-256 datasets. Showing MDiT, DiT baseline with $x_0$ prediction and Min-SNR (mSNR), and DiT with $\epsilon$ prediction.

(Fig.6b), and a $3.47\times$ speedup with the L-scale model (Fig.6c). These outcomes underscore that our MDiT model not only achieves faster convergence rates but also maintains this performance advantage across different datasets and scales. Additionally, for comparative analysis, we include the training dynamics from Peebles & Xie (2022), trained under the $\epsilon$ objective. This inclusion contextualizes our findings, demonstrating that MDiT's improvements result in compounded speedups.

## 5.2 ARCHITECTURAL ABLATIONS

We evaluate key architectural components of MDiT through ablations summarized in Table 2. First, we assess the shift from DiT to LLaMA blocks (with GeGLU), which significantly reduces parameter count. Next, we isolate the effects of the MDiT blocks, Cross-Attention, RoPE, and the proposed multi-scale architecture (Outer and Aggregate blocks). Results show that while the shift from DiT to MDiT blocks offers substantial gains, the single largest contributing factor is the multi-scale architecture (-22%). Further details provided in Appendix F.

Table 2: Ablations on ImageNet.

| Method | FID ↓ (Rel.%) |
|---|---|
| DiT-B/2 | 39.78 (+0%) |
| LLaMA Blocks | 39.51 (-0%) |
| MDiT Blocks | 31.27 (-21%) |
| + Cross-Attn. | 28.05 (-10%) |
| + RoPE | 28.05 (-0%) |
| + Outer Blocks | 22.85 (-19%) |
| + Agg. Blocks | 21.77 (-5%) |

## 5.3 IMPACT OF MULTI-SCALE

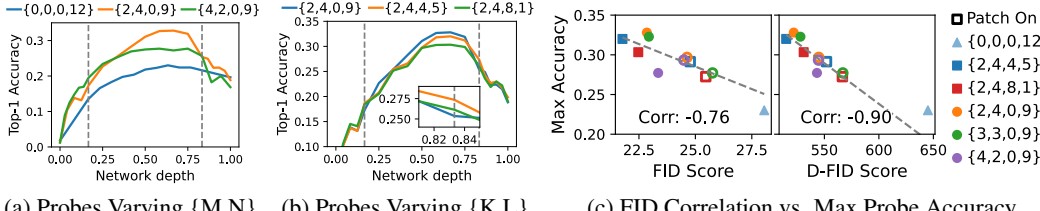

(a) Probes Varying {M,N}.  (b) Probes Varying {K,L}.  (c) FID Correlation vs. Max Probe Accuracy.

Figure 7: (a-b) Comparison of MLP probe accuracy for different values of {M,N,K,L} vs. normalized network depth for $t = 0$. The MDiT core region is marked by vertical dashed lines. (c) Correlation plots of maximum probe accuracy vs. FID and D-FID scores at 300k training steps on ImageNet-256. Open shapes are the patch-on set (see Appendix E), included to improve correlation measure[1].

Evaluating the impact of multi-scale blocks in our MDiT architecture, was achieved through systematically varying the architectural configurations defined by {M, N, K, L} on ImageNet-256. This evaluation focused on analyzing the roles of input and output blocks (M and N), as well as the placement of aggregation blocks (K and L), to determine their contributions to model performance. Our findings indicate that output blocks are more critical than input blocks (i.e., N>M), as observed through semantic probe accuracies across network depths in Fig.7a. This pattern suggests that the MDiT core primarily encodes and processes global information, which is then effectively utilized by the output blocks acting as local decoders. Furthermore, positioning aggregation blocks early in the model, exemplified by the {2,4,4,5} configuration, proved more effective than more dispersed placements ({2,4,8,1}), as depicted by the semantic probes in Fig.7b. Although the top-1 accuracy of {2,4,4,5} is lower compared to {2,4,0,9}, the early inclusion of aggregation blocks enhances the transmission of semantic signals to the output blocks, thereby improving local decoding behavior.

---

[1]Configuration {0,0,0,12} was plotted for completeness and is not included in the correlation measure.

Correlating these architectural impacts with semantic probe accuracies and image fidelity metrics, as shown in Figure 7c, we observed a significant correlation between maximum probe accuracy and both FID and D-FID scores. The stronger correlation with D-FID (-0.90) compared to FID (-0.76) suggests that D-FID provides a more accurate reflection of semantic accuracy. This evidence supports the effectiveness of our multi-scale approach in enhancing semantic encoding capabilities.

## 5.4 IMPACT OF VARIANCE MATCHING

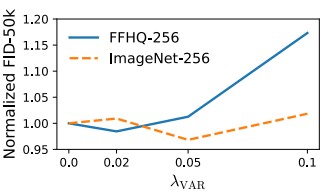
(a) FID vs. $\lambda_{\mathrm{VAR}}$ Curve.

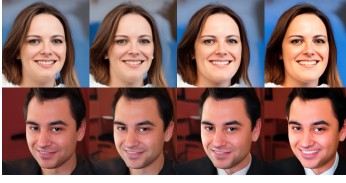
(b) FFHQ Variance Impact.

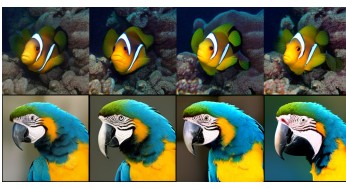
(c) ImageNet Variance Impact.

Figure 8: Impact of variance matching: (a) FID vs. $\lambda_{\mathrm{VAR}}$ scaled to $\lambda_{\mathrm{VAR}} = 0.0$ for comparison; (b-c) Image samples for FFHQ and ImageNet (cfg=3.0) for $\lambda_{\mathrm{VAR}} = 0.0, 0.02, 0.05, 0.1$ (left to right). Comparing MDiT-B models using {M,N,K,L}={2,4,4,5} at 50k (b) and 300k (c) training steps.

To evaluate the effectiveness of variance matching regularization, we varied the loss weighting factor, $\lambda_{\mathrm{VAR}}$, and observed its impact on the Fréchet Inception Distance (FID) and image quality. Figure 8a demonstrates how FID changes with $\lambda_{\mathrm{VAR}}$, normalized to a baseline of $\lambda_{\mathrm{VAR}} = 0.0$. Sample outputs for the FFHQ and ImageNet datasets at different $\lambda_{\mathrm{VAR}}$ settings (0.0, 0.02, 0.05, 0.1) are shown in Figures 8b and 8c. These images demonstrate the visual impact of variance correction, with enhancements in contrast and detail noticeable at moderate $\lambda_{\mathrm{VAR}}$ levels, but a tendency towards oversaturation at higher weights. The results suggest a dataset-specific response to $\lambda_{\mathrm{VAR}}$.

Additionally, for ImageNet, we observed potential adverse effects of variance matching at high CFG scales, where images can appear slightly blurry. This issue may be linked to challenges with $x_0$ prediction and classifier free guidance, as noted in Saharia et al. (2022). To address this, we use a negative conditioning with a resolution condition $< 100\%$, which proved effective (Appendix H.1).

## 5.5 COMPARISON WITH STATE-OF-THE-ART

**FFHQ-256:** For the FFHQ dataset, we employed MDiT-B with a configuration of {2,4,4,5} and a variance regularization weight, $\lambda_{\mathrm{VAR}} = 0.02$. Sample images and detailed evaluation metrics are presented in Figure 1 and Table 3, respectively. Notably, MDiT-B surpasses PDM's (Lu et al., 2023) FID score after 13 million images, while using 6.4 times fewer training FLOPS and a comparable model size. Upon extending to 26 million images, MDiT-B demonstrates similar performance to LDM (Rombach et al., 2021), achieving this with 3.15 times fewer FLOPS and half the model size.

Table 3: Evaluation results on FFHQ 256x256 dataset. Showing model type (Conv-Net and Transformer diffusion), sampling steps (NFE), parameter count (NPar), images seen during training, FLOPS per forward (FLF) and during training (TFL). Marking **overall best** and 100M scale best.

| Method | Type | NFE | NPar | Imgs | FLF | TFL | FID↓ | D-FID↓ |
|---|---|---|---|---|---|---|---|---|
| LDM-4 (Rombach et al., 2021) | Diff·C | 200 | 274M | 27M | 90G | 2.43E | 4.98 | **226.72** |
| P2 Diffusion (Choi et al., 2022) | Diff·C | 500 | **94M** | 18M | 270G | 4.86E | 6.97 | – |
| PDM+CS (Lu et al., 2023) | Diff·C | 100 | 113M | **10M** | 250G | 2.50E | 6.11 | – |
| LFM (DiT/L) (Dao et al., 2023) | Diff·T | 88 | 457M | 26M | 81G | 2.10E | **4.55** | – |
| **MDiT-B (ours)** | Diff·T | 50 | 111M | 13M | 30G | **0.39E** | 5.92 | 280.89 |
| **MDiT-B (ours)** | Diff·T | 50 | 111M | 26M | 30G | 0.77E | 5.48 | 227.60 |

**ImageNet-256:** On ImageNet, MDiT-B and MDiT-L were configured with {2,4,4,5} and {4,8,8,10}, respectively, with a variance regularization weight of $\lambda_{\mathrm{VAR}} = 0.05$. Sample images and evaluation metrics are presented in Figure 1 and Table 4, respectively. Notably, MDiT-B achieves a lower FID score than DiT-L (Peebles & Xie, 2022) at 400k training steps, utilizing $3.4\times$ fewer training FLOPS.

Furthermore, MDiT-L surpasses LDM (Rombach et al., 2021) in all metrics while using only $0.75\times$ the images and FLOPS, significant given the typically slower convergence of transformers compared to convolutional models. Additionally, with extended training, MDiT-L achieves performance competitive with DiT-XL, requiring $12.5\times$ fewer FLOPS and $11.6\times$ fewer images.

Table 4: Evaluation results on ImageNet 256x256 dataset. Showing parameter count (NPar), images seen during training, training FLOPS (TFL), FID, sFID, DINO-FID, IS, Precision & Recall. Marking **XL-scale best**, and L-scale best. $^\alpha$See App. B; $^\beta$250 DDPM and $^\gamma$100 Euler steps.

| Method | NPar | Imgs | TFL | FID↓ | sFID↓ | D-FID↓ | IS↑ | Prec./Rec.↑ | |
|---|---|---|---|---|---|---|---|---|---|
| LDM-4 (Rombach et al., 2021) | 400M | 214M | 22.17E | 10.56 | – | – | 103.49 | 0.71 | 0.62 |
| + cfg=1.5 | 400M | 214M | 22.17E | 3.60 | – | 112.4 | 247.67 | 0.87 | 0.48 |
| DiT-L/2 (Peebles & Xie, 2022) | 458M | 103M | 8.31E | 23.33 | – | – | – | – | – |
| DiT-XL/2 | 675M | 1.8B | 213.0E | 9.62 | 6.85 | – | 121.50 | 0.67 | **0.67** |
| + cfg=1.5 | 675M | 1.8B | 213.0E | 2.27 | 4.60 | **79.36** | **278.24** | 0.83 | 0.57 |
| ViT-XL (Hang et al., 2023) | 451M | 1.1B | 192.0E | 8.10 | – | – | – | – | – |
| + cfg=1.5 | 451M | 1.1B | 192.0E | **2.06** | – | – | – | – | – |
| HDiT-L (Crowson et al., 2024) | 557M | 742M | 146.9E | 6.92 | – | – | 135.20 | – | – |
| + cfg=1.3 | 557M | 742M | 146.9E | 3.21 | – | – | 220.60 | – | – |
| SiT-XL (+cfg) (Ma et al., 2024) | 675M | 1.8B | 213.5E | **2.06** | **4.50** | – | 270.27 | 0.82 | 0.59 |
| **MDiT-B (ours)** | 137M | 77M | 2.44E | 19.09 | 10.11 | 509.78 | 62.96 | 0.61 | 0.62 |
| **MDiT-L (ours)** | 455M | 154M | 16.98E | 10.34 | 7.32 | 232.37 | 107.93 | 0.69 | 0.63 |
| + cfg=1.5 | 455M | 154M | 16.98E | 3.32 | 7.11 | 97.56 | 261.63 | 0.85 | 0.51 |
| + cfg=1.5 (best)$^\alpha$ | 455M | 206M | 22.76E | 2.88 | 4.63 | 84.21 | 276.94 | 0.86 | 0.51 |
| **MDiT-XL-eps (ours)$^{\alpha,\beta}$** | 572M | **256M** | **38.00E** | 7.64 | 5.34 | 197.14 | 134.51 | 0.70 | 0.65 |
| + cfg=1.5$^{\alpha,\beta}$ | 572M | **256M** | **38.00E** | 2.77 | 4.59 | 81.88 | 269.28 | **0.85** | 0.54 |
| **MDiT-XL-rf (ours)$^{\alpha,\gamma}$** | 572M | **256M** | **38.00E** | 6.85 | 4.59 | 191.09 | 119.53 | 0.69 | **0.67** |
| + cfg=1.5$^{\alpha,\gamma}$ | 572M | **256M** | **38.00E** | 2.32 | 4.55 | 85.51 | 258.04 | 0.83 | 0.57 |

**Additional Evaluation on ImageNet-256:** To better compare with DiT, we adopted the 3-channel CFG strategy proposed by Peebles & Xie (2022), achieving a FID of 2.55 with MDiT-L at 800k steps (206M images). While this approach enhances FID, it adversely affects other performance metrics and falls short of DiT-XL, due to capacity constraints. In response, we trained two MDiT-XL models, configured with {4,9,8,12}, using distinct strategies: the $\epsilon$ (eps) objective and rectified flows (rf). Omitting Min-SNR and variance matching to better isolate architectural performance, these models achieved competitive performance with DiT-XL at 1 million training steps. Although the $\epsilon$ model exhibited a higher FID, it aligned better with DiT-XL across other metrics: sFID, IS, and notably D-FID, which is less sensitive to image artifacts and better correlated with human assessments than FID. These advances represent an effective $7\times$ training speedup compared to DiT, and $5\times$ reduction in both training images and FLOPS when compared with ViT-XL in Min-SNR (Hang et al., 2023).

**Additional Observations and Results:** Further comparisons on ImageNet, insights into convergence and scaling, and more image samples are detailed in Appendices A, B, and L, respectively.

# 6 CONCLUSION AND FUTURE DIRECTIONS

We proposed the Multi-Scale Diffusion Transformer (MDiT), which integrates heterogeneous, asymmetric, scale-specific transformer blocks to mitigate the high computational demands and slow convergence rates typical of diffusion transformers. Utilizing explainable AI techniques, we demonstrated that diffusion transformers naturally adopt structural biases, effectively functioning as semantic autoencoders. This understanding enabled MDiT to achieve a $\geq 3\times$ increase in convergence speed on FFHQ-256x256 and ImageNet-256x256, culminating in a $7\times$ training speedup compared to state-of-the-art models while significantly reducing training FLOPs. Additionally, we developed a variance matching regularization technique that enhances image contrast and vibrancy. Our results highlight substantial potential for further architectural improvements in model efficiency. Future research could explore a more exhaustive architectural sweep, investigate longer-term training and convergence properties, and test behavior at higher resolutions. Studies could also examine alternative training objectives (Karras et al., 2024; Ma et al., 2024) and enhanced inference techniques (Kynkäänniemi et al., 2024). The proposed new directions hold promise for extending the capabilities of diffusion-based image synthesis models, potentially enhancing both their efficiency and depth of understanding.

ETHICS STATEMENT

This work introduces the Multi-Scale Diffusion Transformer (MDiT), which enhances the training efficiency of image synthesis models, requiring fewer computational resources and less data. These improvements allow for more rapid experimentation and validation, benefiting fields where high-quality data is scarce, such as synthetic medical image generation. However, the increased accessibility of advanced image synthesis models also raises ethical concerns. In particular, the potential misuse of this technology for creating deepfakes, spreading misinformation, or violating privacy and security presents significant risks. These concerns reflect broader societal challenges surrounding the development and application of powerful AI technologies.

REPRODUCIBILITY STATEMENT

We have made substantial efforts to ensure the reproducibility of our work. The Multi-Scale Diffusion Transformer (MDiT) architecture is described in the main paper, with additional architectural details, including specific parameters and training hyper-parameters, thoroughly expanded upon in Appendix G. Dataset details, including preprocessing steps, and we also provide pseudocode for the more complex components of the functional behavior in Appendix E & G.5. While source code for the model and training will be provided in the future, the descriptions and resources in the paper and appendices should allow for the reproduction of our experiments.

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

## A MORE SoTA COMPARISONS

Additional comparisons with State-of-the-Art models; shown previously for brevity, are detailed in Table 5 for ImageNet-256x256. We further include the metrics for our DiT-B/2 and DiT-L/2 experiments trained with the same hyper-parameters as MDiT using Min-SNR on the $x_0$ objective.

Table 5: Evaluation Results for diffusion models on ImageNet 256x256 dataset. Showing parameter count (NPar), images seen during training, FID, sFID, DINO-FID, IS, Precision, and Recall. The sampler used for each is shown in square brackets if significant. Showing **XL-scale best**, L-scale best, *B-scale best*, and 3-channel guidance (3C). $^\alpha$See App. B; $^\beta$100 Euler and $^\gamma$250 DDPM steps.

| Method | NPar | Train Imgs | Train FLOPS | FID↓ | sFID↓ | D-FID↓ | IS↑ | Prec./Rec.↑ | |
|---|---|---|---|---|---|---|---|---|---|
| LDM-4 (Rombach et al., 2021) | 400M | 214M | 22.17E | 10.56 | – | – | 103.49 | 0.71 | 0.62 |
| + cfg=1.5 | 400M | 214M | 22.17E | 3.60 | – | 112.4 | 247.67 | 0.87 | 0.48 |
| DiT-B/2 (Peebles & Xie, 2022) | 130M | 103M | 2.37E | 43.47 | – | – | – | – | – |
| DiT-L/2 | 458M | 103M | 8.31E | 23.33 | – | – | – | – | – |
| DiT-XL/2 | 675M | 103M | 18.61E | 19.47 | – | – | – | – | – |
| DiT-XL/2 | 675M | 1.8B | 213.0E | 9.62 | 6.85 | – | 121.50 | 0.67 | **0.67** |
| + cfg=1.5,3C | 675M | 1.8B | 213.0E | 2.27 | 4.60 | 79.36 | 278.24 | 0.83 | 0.57 |
| ViT-XL (Hang et al., 2023) [Heun] | 451M | 1.1B | 192.0E | 8.10 | – | – | – | – | – |
| + cfg=1.5 | 451M | 1.1B | 192.0E | 2.06 | – | – | – | – | – |
| ViT-B (+cfg) | 88M | 512M | 11.78E | 10.0 | – | – | – | – | – |
| LFM (DiT/B) (Dao et al., 2023) | 130M | 1.15B | 26.46E | 20.38 | – | – | – | – | 0.56 |
| + cfg=1.5 | 130M | 1.15B | 26.46E | 4.46 | – | – | – | – | 0.42 |
| Patch Diffusion (Wang et al., 2023) | 280M | 2.5B | 97.5E | 7.64 | 5.36 | – | 130.23 | 0.73 | 0.63 |
| + cfg=1.3 | 280M | 2.5B | 97.5E | 2.72 | 4.86 | – | 243.25 | 0.84 | 0.57 |
| HDiT-L (Crowson et al., 2024) | 557M | 742M | 146.9E | 6.92 | – | – | 135.20 | – | – |
| + cfg=1.3 | 557M | 742M | 146.9E | 3.21 | – | – | 220.60 | – | – |
| DiffiT (Hatamizadeh et al., 2023) | | | | | | | | | |
| + cfg | 590M | 1.53B | 174.6E | **1.73** | 4.54 | – | 276.49 | 0.80 | 0.62 |
| + cfg [DDPM] | 590M | 1.53B | 174.6E | 2.20 | – | – | – | – | – |
| SiT-XL (Ma et al., 2024) [Heun] | 675M | 1.8B | 213.5E | 9.35 | 6.38 | – | 126.06 | 0.67 | 0.68 |
| + cfg=1.5 | 675M | 1.8B | 213.5E | 2.15 | 4.60 | – | 258.09 | 0.81 | 0.60 |
| SiT-XL [Euler-Maruyama] | 675M | 1.8B | 213.5E | 8.61 | 6.32 | – | 131.65 | 0.68 | 0.67 |
| + cfg=1.5 | 675M | 1.8B | 213.5E | 2.06 | **4.50** | – | 270.27 | 0.82 | 0.59 |
| **DiT-B/2 (ours) [DDIM]** | 130M | *103M* | *2.37E* | 30.71 | 5.59 | 700.93 | 39.33 | 0.62 | 0.48 |
| **MDiT-B (ours)** | 137M | 77M | 2.44E | 19.09 | 10.11 | 509.78 | 62.96 | 0.61 | 0.62 |
| **MDiT-B (ours)** | 137M | *103M* | 3.27E | 17.36 | 9.82 | 471.34 | 68.64 | 0.62 | 0.62 |
| + cfg=1.5 | 137M | *103M* | 3.27E | *4.33* | *4.78* | *234.75* | 193.84 | 0.82 | 0.50 |
| **DiT-L/2 (ours) [DDIM]** | 458M | 103M | 8.31E | 17.41 | 5.01 | 462.25 | 59.86 | 0.66 | 0.59 |
| **MDiT-L (ours)** | 455M | 103M | 11.38E | 9.40 | 7.85 | 270.12 | 98.79 | 0.69 | 0.63 |
| **MDiT-L (ours)** | 455M | 154M | 16.98E | 10.34 | 7.32 | 232.37 | 107.93 | 0.69 | 0.63 |
| + cfg=1.5 | 455M | 154M | 16.98E | 3.32 | 7.11 | 97.56 | 261.63 | 0.85 | 0.51 |
| + cfg=1.5,3C$^\alpha$ | 455M | 206M | 22.76E | 2.55 | 4.47 | 99.55 | 237.85 | 0.83 | 0.55 |
| + cfg=1.5 (best)$^\alpha$ | 455M | 206M | 22.76E | 2.88 | 4.63 | 84.21 | 276.94 | 0.86 | 0.51 |
| **MDiT-XL-eps$^{\alpha,\gamma}$ (ours)** | 572M | **256M** | **38.00E** | 7.64 | 5.34 | 197.14 | 134.51 | 0.70 | 0.65 |
| + cfg=1.5$^{\alpha,\gamma}$ | 572M | **256M** | **38.00E** | 3.23 | 4.59 | 76.67 | **301.96** | **0.87** | 0.51 |
| + cfg=1.5,3C$^{\alpha,\gamma}$ | 572M | **256M** | **38.00E** | 2.77 | 4.59 | 81.88 | 269.28 | 0.85 | 0.54 |
| **MDiT-XL-rf$^{\alpha,\beta}$ (ours) [Euler]** | 572M | **256M** | **38.00E** | 6.85 | 4.59 | 191.09 | 119.53 | 0.69 | **0.67** |
| + cfg=1.5$^{\alpha,\beta}$ | 572M | **256M** | **38.00E** | 2.64 | 4.71 | **74.70** | 286.27 | 0.85 | 0.55 |
| + cfg=1.5,3C$^{\alpha,\beta}$ | 572M | **256M** | **38.00E** | 2.32 | 4.55 | 85.51 | 258.04 | 0.83 | 0.57 |

## B VISUALIZING CONVERGENCE

### B.1 EARLY CONVERGENCE WITH MDiT-B

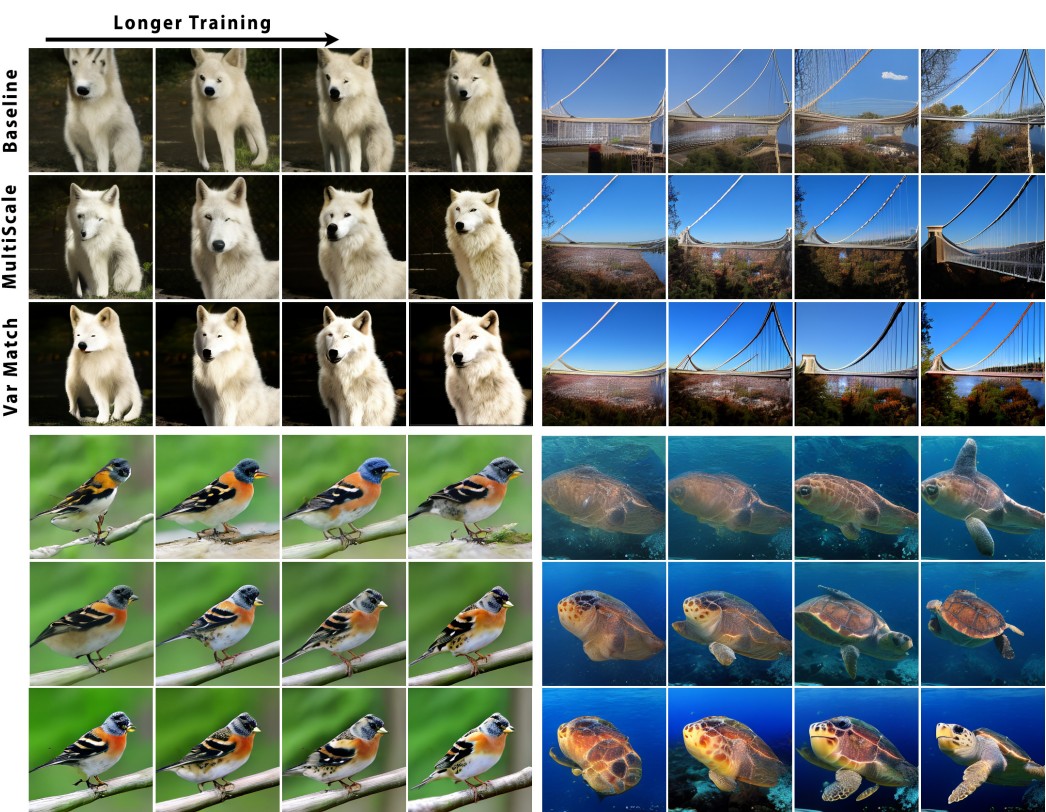

Figure 9: Visualizing Convergence Speedup with MDiT-B on ImageNet-256. Comparing DiT-B (basline), with MDiT-B, and MDiT-B with variance matching. Samples generated with 100 DDIM steps using $\eta = 1.0$, and a cfg=3.0. Showing samples at 50k, 100k, 200k, and 400k training steps.

### B.2 LATE CONVERGENCE WITH MDiT-L AND VARIANCE MATCHING

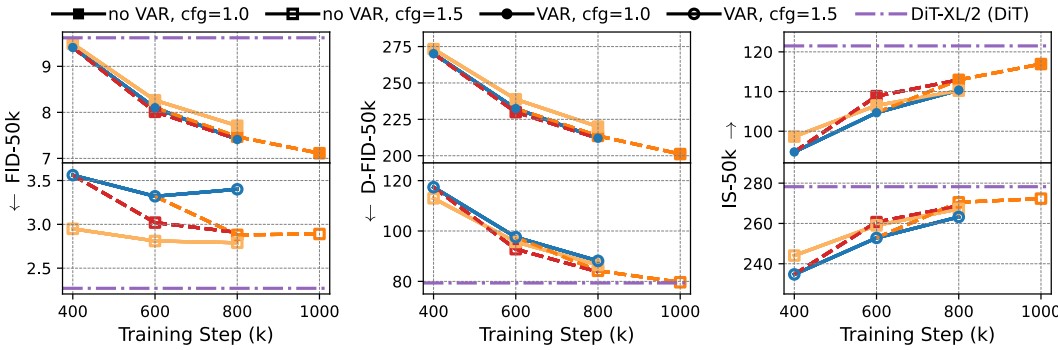

Figure 10: Convergence behavior of MDiT-L trained on ImageNet-256 with and without variance matching. Each line color (blue, yellow, orange, red) shows a different path of finetune resumes (dashed lines) with variance matching enabled or disabled. Evaluated using 100 DDIM steps and $\eta = 0.0$. Final DiT-XL/2 (Peebles & Xie, 2022) metrics (7M steps) shown by dot-dash line.

To understand whether the MDiT-L model achieved long-term convergence, we evaluate it under four distinct conditions to further establish the effects of variance matching and training dynamics on model performance. These conditions are: continuous variance matching (VAR on), no variance matching (VAR off), and discontinuation of variance matching after 400k (VAR off at 400k) and 600k (VAR off at 600k) steps. The differential impacts of these configurations on FID, DINO-FID (D-FID), and Inception Score (IS), both with and without classifier free guidance (CFG), are illustrated in Figure 10. Our analysis reveals that all configurations with CFG stabilize at a FID score of approximately 2.8 around 600k steps, indicating a practical convergence point. Conversely, configurations without CFG continue to improve in D-FID and IS, suggesting potential overfitting benefits these metrics under extended training durations.

The absence of variance matching yields improved performance under CFG, but poorer performance without CFG, highlighting a complex interaction between variance matching and CFG. Intriguingly, when variance matching is discontinued at 400k and 600k steps, a nuanced trade-off emerges: all metrics improve, reaching optimal scores at a slightly degraded FID under CFG compared to the version trained from the start without variance matching. Furthermore, we observe an improvement in high saturation seen in Figure 38 when variance matching is discontinued, approaching the original ImageNet color gamut while retaining better contrast and vibrancy. This indicates that disabling variance matching in later training stages can enhance overall model performance, suggesting a strategic approach to the application of variance matching in training diffusion models.

Moreover, applying a 3-channel CFG method, as proposed by Peebles & Xie (2022), the FID score under CFG conditions improves from 2.79 to 2.55. However, this adjustment results in substantial declines in D-FID and IS by 12 and 29 points, respectively. Comparatively, these results suggest a capacity limitation of MDiT-L, which is expected when comparing L and XL model sizes.

### B.3 LATE CONVERGENCE WITH MDIT-XL

Following the convergence analysis for MDiT-L, we track the key evaluation metrics for both MDiT-XL models, trained on the $\epsilon$ (MDiT-XL-eps) and rectified flow (MDiT-XL-rf) objectives, as a function training step. However, unlike with the previous section, we only track metrics under a CFG scale of 1.5, opting to include with and without 3-channel guidance as proposed by Peebles & Xie (2022). This choice was to prioritize computational resources within our training and evaluation budget. The model behavior can be seen in Figure 11, with each color curve representing a different sampler or step count as indicated in the legend.

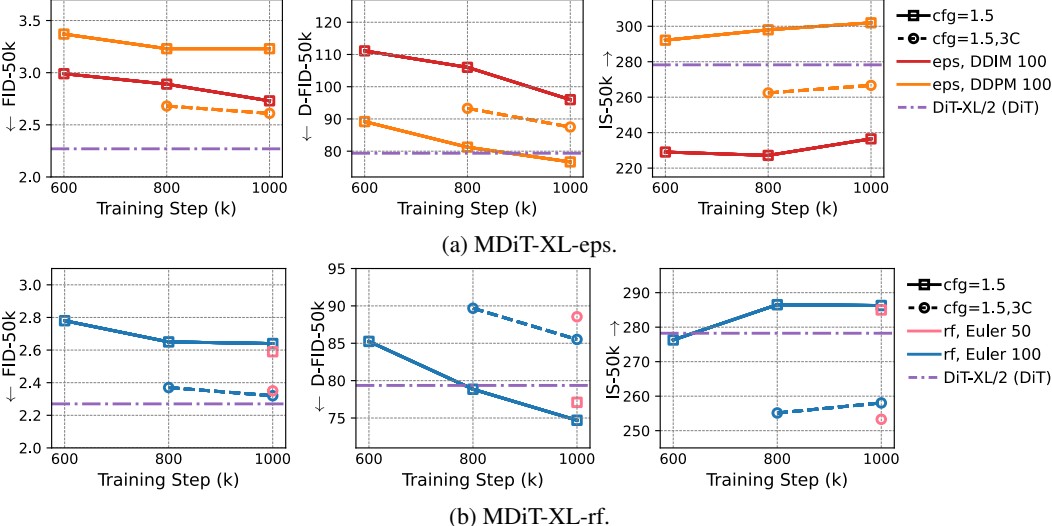

(a) MDiT-XL-eps.

(b) MDiT-XL-rf.

Figure 11: Convergence behavior of MDiT-XL trained on ImageNet-256. (a) Trained on the $\epsilon$ objective, showing curves with and without 3-channel guidance using 100 DDPM sampling steps. (b) Trained with rectified flow matching, showing curves with and without 3-channel guidance using 100 Euler steps. Final DiT-XL/2 (Peebles & Xie, 2022) metrics (7M steps) shown by dot-dash line.

From these plots, we make three key observations:

**Poor DDIM performance on $\epsilon$ objective:** The DDIM sampler performance is significantly degraded when compared to DDPM on the MDiT-XL-eps model. While it does show continual improvement with longer training, the DDPM sampler far exceeds it on all metrics except FID, and completely exceeds the DDIM sampler under 3-channel guidance. We believe this is a result of our training objective, where we follow Peebles & Xie (2022) and predict both the mean and variance of the noise distribution. Consequentially, the DDPM sample is able to make use of this additional information, while it is discarded in the deterministic DDIM sampler. This explains why DDIM performs well on FID, as its deterministic process favors sharp and focused samples that match the real data distribution closely but often at the cost of diversity. In contrast, we observe superior performance with DDIM over DDPM on our $x_0$ models, which are only trained to predict the distribution mean. In this case, the absence of variance modeling aligns more naturally with DDIM's deterministic sampling process.

**Metric shift under 3-channel guidance:** All models exhibit a metric shift when comparing performance with and without 3-channel guidance, where an improved FID score is traded off with degraded D-FID and IS metrics. This behavior is expected, as the 3-channel guidance approximates a CFG scale of $c' = 1 + \frac{3}{4}(c-1)$, meaning a CFG weight of $c = 1.5$ translates to $c' = 1.375$. Well-trained diffusion transformers, however, typically achieve a FID minimum at around $c \approx 1.3$ (Peebles & Xie, 2022; Crowson et al., 2024). D-FID, however, achieves its minimum at a higher weight due to the stronger focus on semantic feature extraction in the DINO-v2 model, while the inception score (IS) scales directly with the CFG weight. Thus, optimizing for FID at lower CFG values necessarily leads to suboptimal D-FID and IS, reflecting the different priorities of these metrics: FID measures inception feature alignment, D-FID emphasizes semantic alignment, and IS captures class alignment.

**Early convergence behavior:** Similar to the previous section, both models exhibit effective FID stagnation under CFG (DDPM and Euler), while other metrics continue to improve. Thus, while extended training improves D-FID and IS, it may degrade the FID score, signaling that the models are approaching practical convergence on ImageNet.

## B.4 SAMPLING CONVERGENCE FOR MDiT-XL

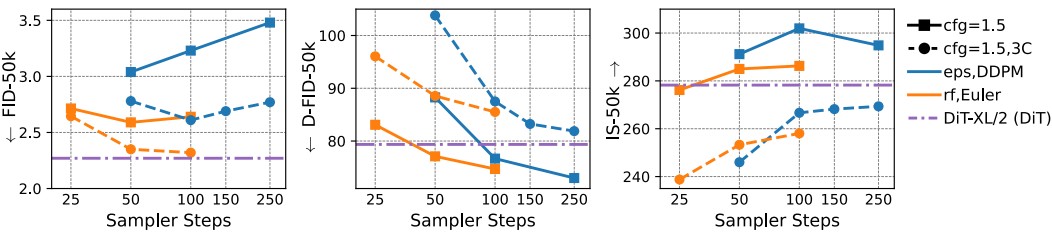

Figure 12: Sampling step-count convergence for MDiT-XL at 1M training steps. Showing metric behavior as a function of sampling step count for each model, with and without 3-channel guidance. MDiT-XL-eps uses the DDPM sampler and MDiT-XL-rf uses the Euler sampler. Final DiT-XL/2 (Peebles & Xie, 2022) metrics (7M steps) shown by dot-dash line.

In examining the MDiT-XL model, trained specifically on the $\epsilon$ (eps) objective and sampled with the DDPM sampler, an unexpected pattern emerged where FID scores were higher than anticipated, despite favorable outcomes in DINO-FID (D-FID), Inception Score (IS), and sFID metrics. To uncover the underlying cause, we investigated the metric performance as a function of the number of sampling steps as shown in Figure 12. Our analysis revealed that while increasing sampling steps initially improved the FID for MDiT-XL-eps, this metric began to degrade after reaching a certain threshold, deviating from expected trends observed in similar studies such as DiT (Peebles & Xie, 2022) and DDIM (Song et al., 2021). This divergence in metric responses likely stems from sample drift inherent in the stochastic DDPM process, which can introduce subtle image deviations particularly penalized by the FID metric, while less affecting other metrics such as D-FID and IS.

Further insights were gained by comparing the behavior of MDiT-XL-rf, trained using rectified flows, to MDiT-XL-eps. When 3-channel guidance was applied, MDiT-XL-rf continually improved across all metrics, including FID, starkly contrasting its performance with full-channel guidance, exhibiting similar behavior to MDiT-XL-eps with an earlier transition point. This highlights the complexities of

evaluating performance using FID and echoes the conclusions and motivations for the development of D-FID as a more robust metric in Stein et al. (2023). Such findings underscore the necessity for diverse evaluation techniques to fully understand model behaviors and inform future enhancements in model architecture and training strategies.

## B.5    COMPUTATIONAL SCALING BEHAVIOR

We investigate the computational scaling properties of the proposed MDiT architecture, focusing on its efficiency during inference and training. For inference, we analyze FLOPs as a function of image resolution, comparing MDiT to DiT and HDiT. For training, we evaluate FID-50k as a function of total training compute. This evaluation provides a quantitative foundation for assessing MDiT's scaling behavior relative to existing models.

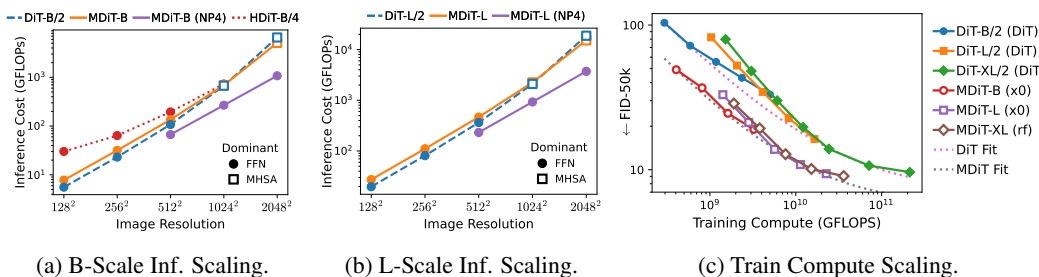

|    (a) B-Scale Inf. Scaling.    |    (b) L-Scale Inf. Scaling.    |    (c) Train Compute Scaling.    |

Figure 13: Scaling behavior comparison of MDiT. (a-b) showing the inference-time cost vs generated image resolution for the B-scale and L-scale models. (c) showing FID-50k vs training compute. Inference scaling compares MDiT against DiT (Peebles & Xie, 2022) and HDiT (Crowson et al., 2024). Further showing where each model becomes attention dominated, and including the method described in Appendix J.2 (NP4). For training compute, comparing against DiT and showing the best-fit scaling curves for each family using the proposed power-law from Henighan et al. (2020).

**Inference Scaling:** Inference scaling is evaluated by analyzing FLOPs as a function of image resolution for the B-scale (Fig. 13a) and L-scale (Fig. 13b) MDiT configurations. Comparisons are made against DiT (Peebles & Xie, 2022) for both scales and HDiT (Crowson et al., 2024) for the B-scale models. We also include results from the Natten + 4x4 patch finetune (NP4), detailed in Appendix J.2. At lower resolutions, MDiT incurs slightly higher FLOPs than DiT due to the additional cross-attention and pixel-shuffle projection layers in the aggregate blocks. However, at attention-dominant resolutions – where scaling transitions from $\mathcal{O}(N)$ to $\mathcal{O}(N^2)$ – MDiT achieves better performance compared to DiT. With the NP4 variant, MDiT retains $\mathcal{O}(N)$ scaling across all resolutions and achieves a significant reduction in FLOPs. These improvements result from MDiT's shallow U-Net structure, which delegates global attention to the aggregate blocks rather than distributing it uniformly across all core self-attention layers. Additionally, both MDiT variants outperforms HDiT in the B-scale comparison, likely due to HDiT's increased overhead from pixel-space compression, which MDiT avoids by operating in the latent space and utilizing a VAE decoder.

Table 6: Fit Parameters for Training Compute Scaling.

| Family | $L(C)$ |
|--------|--------|
| DiT    | $6.84 + \left(\frac{C}{8.39 \times 10^{-5}}\right)^{-0.586}$ |
| MDiT   | $5.81 + \left(\frac{C}{4.03 \times 10^{-5}}\right)^{-0.625}$ |

**Training Scaling:** Training scaling is analyzed by examining the FID-50k metric (without CFG) as a function of total compute, comparing all three MDiT and DiT scales: B, L, and XL. As shown in Figure 13c, MDiT exhibits similar scaling behavior to DiT but is shifted toward lower training compute, reflecting improved training efficiency. To quantify this comparison, we fit the power-law

formulation proposed by Henighan et al. (2020), expressed as:

$$L(x) = L_\infty + \left(\frac{x_0}{x}\right)^{\alpha_x} \tag{3}$$

where $x$ is the independent variable, representing compute ($C$) in our analysis, $L_\infty$ is the irreducible loss (minimum FID), and $x_0$ and $\alpha_x$ are fit parameters. The fitted values for both model families are presented in Table 6. Both DiT and MDiT achieve comparable exponents ($\alpha_x \sim -0.6$), while MDiT exhibits a $2\times$ smaller $x_0$, which accounts for its shift toward lower compute. While MDiT also fits a lower $L_\infty$, this measure's robustness is uncertain due to fit divergence at higher compute levels, likely caused by the limited availability of larger models in both families. However, the parallel nature of the fits suggests that the systematic biases affecting both families are comparable, allowing for meaningful relative comparisons between the two.

## C  Text to Image on CC3M

To bolster the performance improvement claims previously demonstrated with the FFHQ and ImageNet datasets, we extend the Multi-Scale Diffusion Transformer (MDiT) to a text-to-image (T2I) synthesis task using the Conceptual Captions 3M (CC3M) dataset (Sharma et al., 2018)[2]. This adaptation employs the same foundational architecture as our large-scale model (MDiT-L) applied to ImageNet-256, with the modification of incorporating T5-FLAN-L (Chung et al., 2022) embeddings for handling text conditioning. Replacement of the class embedding layer with text embeddings maintains the same embedding dimensions (1024), ensuring architectural consistency while allowing the generation of images directly conditioned on textual descriptions. This approach showcases the architecture's capability to handle more intricate conditioning scenarios without significant structural changes.

### C.1  Image Examples and Performance Benchmarking

In the evaluation of the text-to-image model on the CC3M dataset, we adhered to the training setup and evaluation protocol as detailed in Section 5, with the model undergoing training for 200,000 steps due to computational constraints. However, we choose to train this model without variance matching given the lack of low-resolution images (less than $0.08\%$ of the dataset) necessary to establish a sufficient auxiliary scale conditioning. Example generations with out-of-distribution prompts are shown in Figure 14.

For consistency with established practices in the field, we opted for 50 DDIM sample steps for evaluation, deviating from the 100 steps used previously for the FFHQ and ImageNet datasets. Evaluations were conducted on two distinct validation sets: the CC3M validation set (Sharma et al., 2018), which comprises approximately 13,000 images and was not used during training, and the MS-COCO 2014 validation set (Lin et al., 2014), containing 30,000 images. We chose Fréchet Inception Distance (FID) (Heusel et al., 2017), DINO-FID (D-FID) (Stein et al., 2023), and CLIP-L/14 similarity scores (Radford et al., 2021) as our metrics, computing these for both validation sets to provide a comprehensive view of the model's performance across different datasets. Notably, all evaluations employed classifier-free guidance (CFG) (Ho & Salimans, 2021) with a scaling factor of 2.5. Table 7 shows comparisons with other state-of-the-art models.

While the results presented in Table 7 show promise, they fall sort of their state-of-the-art counterparts. We attribute this gap largely due to the lack of training steps, further noting that training efficiency may be hindered by a combination of the dataset and poor alignment of captions within (see Fig. 39 for examples), in addition to the choice of T5-FLAN-L as the text encoder. Saharia et al. (2022) showed a strong interdependence on both FID and CLIP score as a function of text encoder, where T5-L to T5-XXL could account for a 0.02 and 2 point improvement on CLIP and FID scores, respectively. Nevertheless, this experiment shows promise in improving the training efficiency for T2I models.

---

[2]We used the snapshot uploaded to `https://huggingface.co/datasets/pixparse/cc3m-wds`, as many of the original links are no longer valid ($\sim 30\%$). Auto-cropping was used to remove white boarders applied to the validation images in the snapshot, allowing for a fair FID score given the training augmentations.

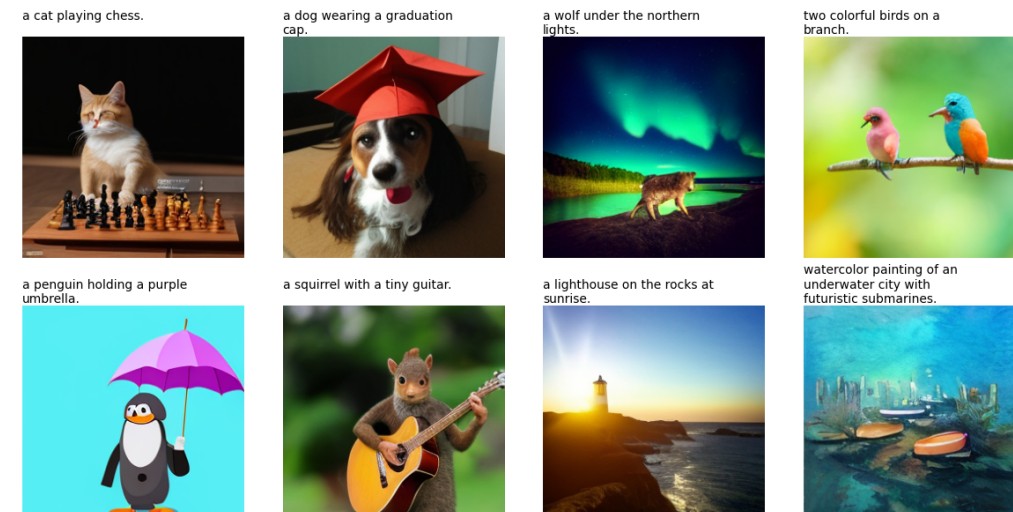

Figure 14: Select image samples from the MDiT-L CC3M model at 200k training steps, with 50 DDIM steps using $\eta = 1.0$, cfg=4.0.

Table 7: Comparative performance of different models on COCO-30k and CC3M-13k validation datasets. *CC3M results from (Chang et al., 2023). [†]Computed on 10k images. [§]Computed with CLIP-B/16.

| Method | Params | Zero-shot COCO-30k | | | CC3M-13k | | |
|---|---|---|---|---|---|---|---|
| | | FID↓ | D-FID↓ | CLIP↑ | FID↓ | D-FID↓ | CLIP↑ |
| LDM-4 (Rombach et al., 2021) | 645M | 12.63 | – | – | 17.01* | – | 0.24* |
| SDv1.5 (Rombach et al., 2021) | 890M | 9.62 | – | 0.257[†] | – | – | – |
| Muse (Base) (Chang et al., 2023) | 632M | – | – | – | 6.8 | – | 0.25 |
| Karlo (Lee et al., 2022) | 3.3B | 13.95 | – | 0.319[§] | 14.43 | – | 0.308[§] |
| RAPHAEL (Xue et al., 2023) | 3.0B | 6.61 | – | 0.33[†§] | – | – | – |
| Imagen (Saharia et al., 2022) | 2.6B | 7.27 | – | 0.265[†] | – | – | – |
| PIXART-$\alpha$ (Chen et al., 2024) | 600M | 7.32 | – | 0.260[†] | – | – | – |
| **MDiT-L (ours)** | 454M | 16.06 | 458.68 | 0.233 0.291[§] | 11.21 | 256.11 | 0.213 0.273[§] |

# D MEASURING CORE CONTRIBUTION

In our analysis, particular emphasis is placed on diffusion sampling step 12/25, identified through preliminary experiments as a transitional point during inference. At this step, the MDiT core exhibits the highest semantic contribution relative to the skip connection used by the outer blocks. This pattern aligns with U-Net-like architectures in diffusion models, where early steps address large-scale semantic content and scene composition, and later steps enhance fine details. This distinction also justifies the use of aggregation blocks over an additional down-sampling layer. To quantitatively evaluate the contributions across sample time-steps, we generate 10,000 samples with varying seeds and image classes, and compute statistics on the MDiT core output (post-upsample) and skip-connections. The results for configurations {2,4,0,9} and {2,4,4,5} after 300,000 training steps of MDiT-B on ImageNet are presented in Figure 15.

To systematically analyze the core and skip connection outputs at each diffusion step, we computed several key statistics. The L2 norm measures the magnitude of vectors, serving as an indicator of activation strength. The relative L2 norm evaluates the contribution ratio between the core output and skip connection activations, providing insight into the relative energy distribution between these two sources. Additionally, the channel-wise relative contribution (next section) is computed to emphasize channels that contain concentrated semantic power, which may not be fully captured by the L2 norm alone. We also calculate the mean and standard deviation for these metrics across the 10,000 samples

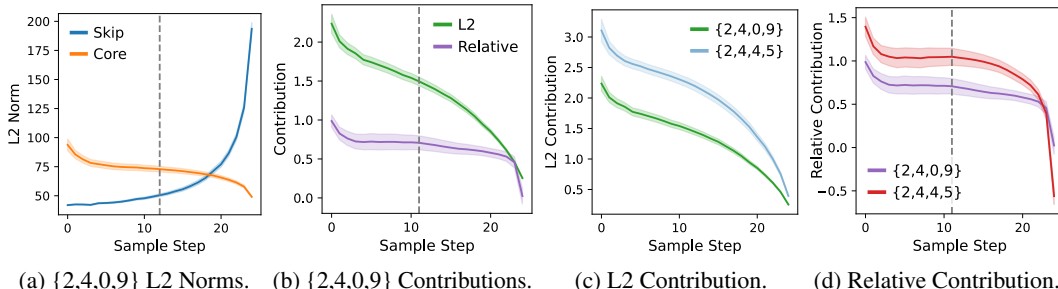

(a) {2,4,0,9} L2 Norms.     (b) {2,4,0,9} Contributions.     (c) L2 Contribution.     (d) Relative Contribution.

Figure 15: Core contribution of MDiT-B trained on ImageNet at 300k steps. Showing mean and 1-sigma spread for 10k samples, with step 12/25 indicated using a vertical dashed line. (a) L2 norms of core output and skip connection for the {2,4,0,9} configuration; (b) L2 norm ratio and channel-wise relative contribution for the {2,4,0,9} configuration; (c) L2 norm ratio for the {2,4,0,9} and {2,4,4,5} configurations; (d) Channel-wise relative contribution for the {2,4,0,9} and {2,4,4,5} configurations.

to capture the central tendency and variability of the contributions. This approach reflects the model's consistency and its sensitivity to varying input conditions. A vertical dashed line at step 12/25 in Figure 15 highlights the significance of this step, as suggested by the activation metrics.

Analysis of the activation metrics reveals distinct patterns of contribution across diffusion steps. The L2 norm shows a monotonic decrease as the sampling progresses, aligning with the expected behavior due to an increase in signal-to-noise ratio (SNR) during the reverse diffusion process. In contrast, the relative contribution remains predominantly flat, suggesting that the core continues to make meaningful semantic contributions even as the ratio of L2 norms drops below unity. This suggests that significant semantic conditioning is effectively maintained in the core up until the final sampling steps. Notably, both the L2 and relative contributions from the configuration {2,4,4,5} exceed those of {2,4,0,9}. These results echo the findings in Section 5.3, where earlier aggregation blocks were shown to enhance semantic signal transmission at the MDiT core output, substantiating the efficacy of this architectural choice.

### D.1 CHANNEL-WISE RELATIVE CONTRIBUTION

Given the unique characteristics of the MDiT architecture, traditional metrics such as the L2 norm provide incomplete insights into the interplay between core and skip connection outputs. To address this gap, we utilize a channel-wise relative contribution metric. This measure is specifically designed to assess power dominance between the core output and the skip connection in a manner that accounts for non-uniform signed activation distributions. Such distributions are expected when certain feature channels convey more semantic information than others, highlighting the necessity for a metric that can evaluate the directional and magnitude disparities between activations effectively. The metric is defined as follows:

$$
\rho(c, s) = \begin{cases} c - s & \text{if } c \geq 0 \text{ and } s \geq 0, \\ s - c & \text{if } c < 0 \text{ and } s < 0, \\ c + s & \text{if } c \geq 0 \text{ and } s < 0, \\ -c - s & \text{if } c < 0 \text{ and } s \geq 0. \end{cases} \tag{4}
$$

This function is designed to apply the appropriate operation based on the sign of the activations in the core $c$ and skip connections $s$:

- **Subtraction** $(c-s)$ when both activations are of the same sign, reflecting a direct comparison of their magnitudes, with a negation when they are both negative.

- **Addition** $(c + s)$ when the core is positive and the skip is negative, indicating the total magnitude of opposing contributions.

- **Negative addition** $(-c - s)$ when the core is negative and the skip is positive, emphasizing the skip connection's dominant positive contribution over the core's negative impact.

The metric $\rho$ is computed for each feature channel within the activation tensors and can be averaged across all channels and image tokens to provide a comprehensive view of the relative contributions. The value of $\rho$ will be zero when the contributions from the core and skip connections are approximately equal, positive when the core's contribution dominates, and negative when the skip connection contributes more significantly. As illustrated in Figure 15, this metric demonstrates a relatively stable channel-wise semantic content across most diffusion sampling timesteps, with a notable decline only in the final steps. This distinct pattern of stability followed by a drop-off contrasts sharply with the L2 norm, which shows a continuous monotonic decline in energy, misleadingly suggesting a uniform reduction in semantic content throughout the diffusion process.

# E  PATCH REGULARIZATION

During training, the output activations of the UnPatch module in the MDiT core exhibited a checkerboarding pattern, the impact of which on model performance remains unclear. This pattern may result from the reduced constraints on gradient signals, given the output's location N blocks from the latent head, or the output blocks' capacity to mitigate inherent noise in network activations. To explore and potentially address this issue, we devised a patch regularization method. This method computes a 2D-FFT on the UnPatch output activations, calculates the mean spectral frequencies across batches and channels, and then determines the difference between this mean and the frequencies associated with checkerboarding. A ReLU activation is applied to ensure that only the excess spectral power at the checkerboarding frequencies is penalized. The specific steps of this process are detailed in the pseudocode provided in Listing 1.

Listing 1: Patch Regularization Implementation

```python
def patch_loss(x):
    # Extract the activation shape
    B, H, W, _ = x.shape

    # We have (B,H,W,C) and want to perform the FFT on
    # dims -3 and -2 (i.e. H and W)
    # We also want to perform the spatial reduction twice
    # on -3 and -2 for W and then H
    # The output will then be of shape (B,)
    # Note that in this case, the mask will be H,W,1
    # (the 1 can broadcast to C)

    # 1) Create a mask to select the checker frequencies
    mask = torch.ones((H, W, 1), dtype=x.dtype, device=x.device)
    mask[0    ,W//2] = 0 # w-patch
    mask[H//2,    0] = 0 # h-patch
    mask[H//2,W//2] = 0 # hw-coupled patch

    # 2) Apply the fft - have to convert to float
    # because fp16 is not supported
    xf = torch.fft.fft2(x.float(), dim=(-3, -2)).abs()

    # 3) Compute the baseline mean over H and W, excluding the mask
    m = (xf*mask).mean(dim=(-2,-3))

    # 4) Construct the goal tensor by replacing the patch frequencies
    xm = xf.clone()
    xm[:,    0,w//2] = m # w-patch
    xm[:,h//2,    0] = m # h-patch
    xm[:,h//2,w//2] = m # hw-coupled patch

    # 5) compute the difference loss
    # We are using ReLU here to prevent penalization if the patch
    # frequencies are below the mean
    # It's okay if the model learns to ignore them, but it's
    # not okay if the model emphasizes them
    # Note the stop grad, which prevents back prop through the mean
    delta = torch.nn.functional.relu(xf - xm.detach())

    # 6) Reduce the H, W, and C dims out, resulting in shape (B,)
    return delta.mean(dim=(-1,-2,-3))
```

While the patch regularization effectively mitigated the checkerboarding behavior, two issues were observed. First, loss spikes associated with the patch regularization emerged after 100,000 training steps when training MDiT-B on ImageNet-256, although these spikes did not significantly impact the MSE training loss. Second, there was an in-excess of spectral power at frequencies corresponding to the checkerboarding, indicating potential deficiencies in those frequencies which might negatively impact model performance. Consequently, we adjusted the regularization schedule to discontinue after 100,000 steps. The results of three experiments, along with their Fréchet Inception Distances (FID) at 300,000 training steps, are depicted in Figure 16.

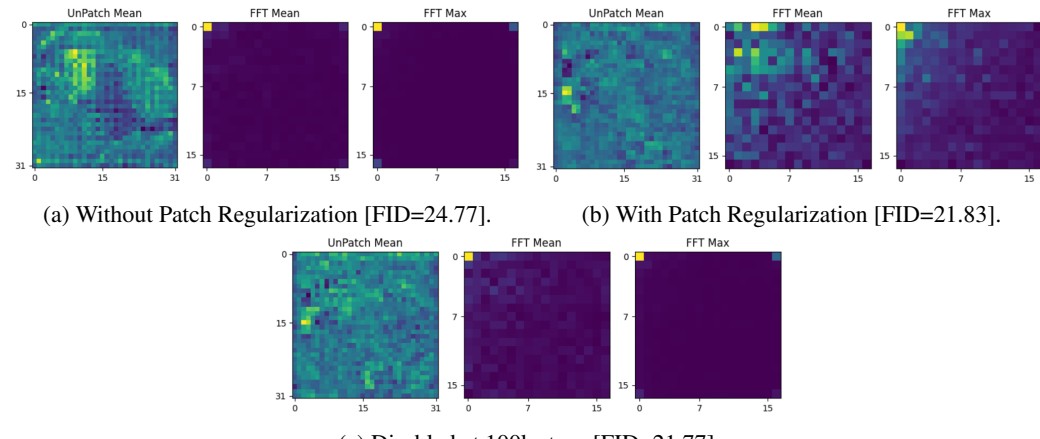

(a) Without Patch Regularization [FID=24.77].  (b) With Patch Regularization [FID=21.83].

(c) Disabled at 100k steps [FID=21.77].

Figure 16: Comparing UnPatch outputs at step 12/25 (maximum core contribution) of MDiT-B models trained to 300k steps on ImageNet-256. Each subplot shows a typical image sample output activation of the MDiT core UnPatch block, showing channel-wise mean, FFT channel mean, and FFT channel max. Comparing cases of: (a) no patch regularization; (b) always enabled patch regularization; and (c) scheduled patch regularization which is disabled at training step 100k.

Disabling the patch regularization at 100,000 steps led to a modest improvement, as indicated by a 0.06 point reduction in the Fréchet Inception Distance (FID) score. Although some undesirable spectral power reappeared, checkerboarding was no longer observed in the mean activations. This supports the utility of patch regularization in the early stages of training to help establish favorable model behavior. Additionally, the reappearance of this spectral power suggests that its complete removal might obscure crucial information necessary for the output blocks. Notably, patch regularization was not applied to the aggregation blocks, which also display a checkerboarding pattern, albeit to a much lesser extent. This strategy's benefits align with improvements observed when more output blocks are used, indicating that a deeper network can better manage aberrant signals.

## F ADDITIONAL ABLATIONS

In this section, we present a comprehensive set of ablation experiments aimed at understanding the impact of individual architectural choices on model performance. The results, summarized in Table 8, systematically evaluate modifications to the feedforward network (FFN) layers, attention mechanisms, and conditioning schemes. These experiments isolate the individual contributions of each change while also exploring potential interdependencies among them.

The ablations follow a structured approach. We first assess the impact of modifying the base transformer blocks by comparing the original DiT blocks to LLaMA-style blocks. Next, we investigate the effects of augmenting these blocks for MDiT by incorporating elements such as cross-attention mechanisms and removing conditioning gates. We then evaluate the inclusion of scale and aspect conditioning (Aux Cond.), the addition of rotary position embeddings (RoPE), the replacement of adaptive norm conditioning with a time token, and the individual impacts of the two multi-scale architectural contributions: the outer blocks and the aggregate blocks. Furthermore, we explore the effect of adding the multi-scale components directly to the base LLaMA architecture to isolate any dependencies with the specific MDiT adjustments.

**Overview of Results:** Our ablation experiments reveal both independent and compounding contributions of architectural modifications to performance. Transitioning from DiT to LLaMA-style blocks increases FID slightly, likely due to a destructive interaction between DiT conditioning gates and the GeGLU Feed Forward Network (FFN). Removing the gates resolves this issue, enabling GeGLU to improve performance in line with prior works. Beyond block structure, the addition of multi-scale components (outer and aggregate blocks) enhances performance, delivering gains comparable to or exceeding those from GeGLU. Importantly, these improvements are additive – the benefits of the multi-scale components are independent of the initial block configuration.

Table 8: Detailed ablation results for MDiT-B on ImageNet-256. Showing class conditioning method: Adaptive-Norm (AN), Gate (G), Cross-Attention (CA), and Time Token (T). Also showing position embedding type: Sinusoidal (Sin) or RoPE. Further showing FID, D-FID, Param. count, and FLOPs.

| Configuration | | Cond. | PE. | FID ↓ | D-FID ↓ | Params | FLOPs |
|---|---|---|---|---|---|---|---|
| **Baseline** | | | | | | | |
| A | DiT B/2 (Peebles & Xie, 2022) | AN+G | Sin | 39.78 | 770 | 130M | 23.0G |
| **LLaMA Blocks** | | | | | | | |
| B | A - Bias; LN→RN; +Attn Norm. | AN+G | Sin | 38.62 | 784 | 120M | 23.1G |
| C | B + GeGLU (Shazeer, 2020) | AN+G | Sin | 39.51 | 791 | 120M | 23.1G |
| **MDiT Blocks** | | | | | | | |
| D | B - Gates | AN | Sin | 50.34 | 929 | 105M | 23.1G |
| E1 | D + GeGLU | AN | Sin | 31.27 | 682 | 105M | 23.1G |
| E2 | D + Cross-Attn. | AN+CA | Sin | 39.07 | 782 | 139M | 26.8G |
| F | C - Gates; +Cross-Attn. | AN+CA | Sin | 30.15 | 636 | 139M | 26.8G |
| **MDiT Mulit-scale** | | | | | | | |
| G | F + Aux Cond | AN+CA | Sin | 28.05 | 632 | 141M | 26.8G |
| H | G + RoPE | AN+CA | RoPE | 28.05 | 645 | 141M | 26.8G |
| I | H + Time Token | CA+T | RoPE | 27.47 | 626 | 133M | 26.8G |
| J | I + Outer blocks | AN+CA+T | RoPE | 22.85 | 521 | 118M | 31.9G |
| K | J + Aggregate Blocks | AN+CA+T | RoPE | 21.77 | 514 | 137M | 31.7G |
| **LLaMA Multi-scale** | | | | | | | |
| R | B + Outer & Agg. Blocks | AN+G | Sin | 30.77 | 665 | 133M | 29.5G |

Notably, several modifications have a minor impact on the overall performance metrics; however, these changes offer other significant benefits not captured by these metrics:

- **Removing bias terms:** Reduces B-Scale model size by 10 million parameters (7.7%).
- **Scale and aspect conditioning:** Enables zero-shot scale and aspect adjustment.
- **RoPE:** Facilitates zero-shot extrapolation to larger resolutions and arbitrary aspect ratios.
- **Time token:** Eliminates an additional 8 million parameters (5.7%) and simplifies the blocks.

**Architectural Interdependencies:** We observe that many of the architectural changes are largely independent, allowing their effects to compound. However, there is an interdependency between the conditioning gates, cross-attention mechanisms, and GeGLU-based FFN layers. This relationship appears to stem from a conditional feature suppression mechanism that may be required by diffusion transformers. The base DiT blocks use a gating mechanism for conditioning, which modulates the output of the FFN and self-attention layers. Removing the gates disrupts this suppression, degrading performance. Replacing the FFN with a GeGLU layer reintroduces a similar mechanism, as does adding cross-attention, albeit through indirect means. While GeGLU compensates for the removal of gates, combining GeGLU with gates appears to introduce interference, resulting in degraded performance compared to using either mechanism independently. This suggests a potential redundancy or conflict in how these mechanisms apply conditional information. Notably, the absence of gates paired with GeGLU consistently outperforms cross-attention alone, indicating that the degradation is likely related to timestep conditioning rather than difficulties in integrating class conditioning.

## F.1 PARAMETERIZED MULTI-SCALE CONFIGURATION

Building upon the previous subsection, we discuss additional architectural ablations not present in the main paper. Table 9 lists the results shown in Section 5.3, in addition to studying the impact of using a time token in the cross-attention compared with the typical adaptive modulation scheme. The results indicate a reduction in parameter count (due to fewer modulated scale activations), and a reduction in both FID and D-FID when using a time token. The configurations with $M, N, K \neq 0$ retain the modulation scheme in blocks without cross-attention as adding a cross-attention layer would have traded a marginal improvement for additional FLOPS and parameters. However, in the cases which already used cross-attention, utilizing this method impacted all metrics favorably.

Table 9: Structural Ablations for the MDiT Base model on ImageNet-256. Showing Giga-FLOPS, parameters count (Millions), time condition (Modulation, Token, or Hybrid), FID, and DINO FID at 300k training steps, using 50 DDIM sampling steps without CFG. Showing **best** and chosen.

| Configuration | FLOPS | Params | M | N | K | L | Time | FID $\downarrow$ | D-FID $\downarrow$ |
|---|---|---|---|---|---|---|---|---|---|
| DiT-B/2 equivalent | 26.7G | 141M | 0 | 0 | 0 | 12 | Mod | 28.05 | 645 |
| MDiT-B/2 + time token | 26.7G | 133M | 0 | 0 | 0 | 12 | Token | 27.47 | 626 |
| MDiT balanced Enc-Dec | 31.9G | 118M | 3 | 3 | 0 | 9 | Hyb | 22.94 | 528 |
| MDiT big Enc, small Dec | 31.9G | 118M | 4 | 2 | 0 | 9 | Hyb | 23.35 | 543 |
| MDiT small Enc, big Dec | 31.9G | 118M | 2 | 4 | 0 | 9 | Hyb | 22.85 | 521 |
| MDiT full aggregate | 31.5G | 156M | 2 | 4 | 8 | 1 | Hyb | 22.47 | 531 |
| MDiT half aggregate | 31.7G | 137M | 2 | 4 | 4 | 5 | Hyb | **21.77** | **514** |

Table 10: Ablations on position embeddings and MDiT structure for the MDiT Base model on ImageNet-256. Comparing with FID, and DINO FID at 300k training steps, using 50 DDIM sampling steps without CFG. Showing **best** and chosen.

| Configuration | L | MDiT Block Type | Pos. Method | RoPE Freq. | FID $\downarrow$ | D-FID $\downarrow$ |
|---|---|---|---|---|---|---|
| Serial Baseline | 12 | Serial | Sinusoidal | N/A | 28.05 | 632 |
| Serial RoPE | 12 | Serial | RoPE | 16 | 28.05 | 645 |
| Serial RoPE | 12 | Serial | RoPE | 32 | **27.33** | **629** |
| Parallel RoPE | 12 | Parallel | RoPE | 16 | 36.39 | 772 |

Table 10 considers the impact of position embeddings, and the difference between serial and parallel transformer blocks as proposed by Wang & Komatsuzaki (2021); Dehghani et al. (2023). Our results showed little impact on metrics when switching from sinusoidal position embeddings to RoPE, with a slight improvement when increasing the RoPE frequency. This improvement may be due to an increased semantic head capacity, which could alternatively be achieved by decreasing the $r_{dim}$ channel threshold. However, we did not explore this further due to computational constraints, and decided to use a frequency of 16 throughout our experiments as it provided higher phase resolution for fine detail.

When considering parallel vs. serial transformer blocks, we observed a 3% speedup in wall-time with the parallel case at the cost of a significant performance degradation. This discrepancy does not align with the results in Dehghani et al. (2023), which may be due to the model scale, where Dehghani et al. considered ViTs up to 22 Billion parameters (MDiT-B used 133 Million). Alternatively, the performance degradation could be due to the task, where image classification is less sensitive to feature shifts than generative image models.

## F.2 TIMESTEP DEPENDENT PROBE BEHAVIOR

The utilization of $x_0$ prediction facilitates a probe analysis at $t = 0$, capitalizing on the model's training towards reconstructing the original clean images. This contrasts with the approach of Tang et al. (2023), who conducted a correspondence analysis on Stable Diffusion (Rombach et al., 2021) using noised images ($t > 0$) due to its noise ($\epsilon$) objective. Nevertheless, similar patterns may be discernible with MDiT.

We revisited the probe analysis detailed in Section 4.2, applying it to the configurations $\{M, N, K, L\} = \{0, 0, 0, 12\}$ and $\{2, 4, 5, 4\}$ across various diffusion timesteps. The results are visualized as a heat map of probe accuracy vs. normalized network depth and diffusion timestep in Figure 17, accompanied by the maximum probe values for each timestep. Notably, the maximum probe behavior echoes the findings in Tang et al. (2023), suggesting a parallel with models trained under the $\epsilon$ objective. Additionally, the initial discrepancy between $M, N = 0$ and $M, N \neq 0$ observed in Section 4 persists across all measured timesteps.

To further examine the probe behavior as a function of the heterogeneous configuration, the analysis from Section 5.3 was reproduced at diffusion timestep $t = 110$, where maximum probe accuracy is

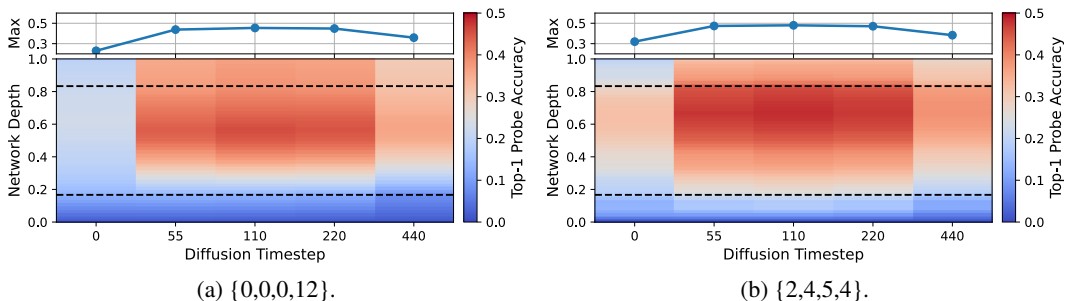

(a) {0,0,0,12}.        (b) {2,4,5,4}.

Figure 17: Comparison of MLP probe accuracy as a function of diffusion timestep for different values of {M,N,K,L} vs. normalized network depth. The MDiT core region is marked by horizontal dashed lines. Maximum probe value for each timestep plotted vs. timestep above heatmaps. Probe accuracies from MDiT-B models at 300k training steps on ImageNet-256.

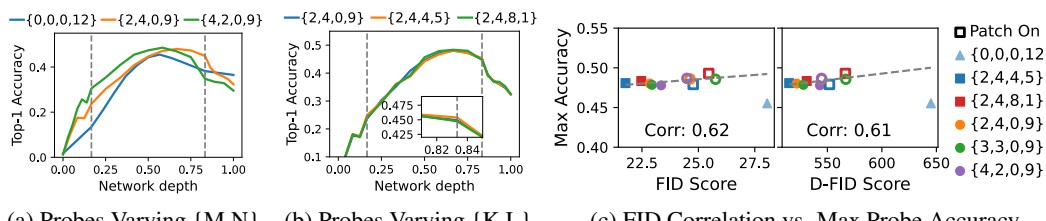

(a) Probes Varying {M,N}.    (b) Probes Varying {K,L}.    (c) FID Correlation vs. Max Probe Accuracy.

Figure 18: (a-b) Comparison of MLP probe accuracy for different values of {M,N,K,L} vs. normalized network depth for $t = 110$. The MDiT core region is marked by vertical dashed lines. (c) Correlation plots of maximum probe accuracy vs. FID and D-FID scores at 300k training steps on ImageNet-256. Open shapes are the patch-on set (see Appendix E).

observed as shown in Figure 17. This analysis revealed accuracy patterns similar to those presented in Figure 7, but with nearly identical vertical scaling for $M, N \neq 0$. Additionally, the analysis of the correlation between maximum probe accuracy and both Fréchet Inception Distance (FID) and DINO-FID (D-FID) showed a shift to nearly horizontal, suggesting that higher probe accuracy does not correlate strongly with these FID metrics. This observation suggests that the near-zero correlation between maximum probe accuracy and FID metrics at noisy input stages indicates strong semantic similarities across different architecture configurations. As $t \rightarrow 0$, despite a decrease in maximum probe accuracy (implying a reduction in semantic power) there is a significant differentiation among model architectures, which strongly correlates with overall model performance. This trade-off underscores that at lower timesteps, distinct architectural features become more pronounced when training under $x_0$, though at the expense of overall semantic accuracy.

## G  IMPLEMENTATION DETAILS AND HYPER-PARAMETERS

We implemented all models using PyTorch and utilized PyTorch Lightning to handle distributed training and mixed precision operations. The training was performed on a mix of NVIDIA A6000 and 80GB A100 GPUs, using Distributed Data Parallel (DDP) and gradient accumulation to optimize computational resources. Evaluations were conducted on NVIDIA A6000 GPUs. The NVIDIA Apex library was employed for fused RMS normalization operations to enhance computational efficiency. For attention mechanisms, we used the xFormers library (Lefaudeux et al., 2022) for standard attention and the NATTEN library (Hassani et al., 2023) for neighborhood attention.

Our training regimen incorporated mixed precision techniques, specifically using bfloat16 for most operations while maintaining float32 precision for all attention-related computations. Initial tests with bfloat16 for attention operations indicated stable training; however, the loss trajectory exhibited more variability compared to using float32. No significant differences were noted when using bfloat16 instead of float32 for other model components. A gradient clip of 1.0 was set as a safeguard, though it was not triggered during training.

For data handling, we converted all images into WebDataset shards. FFHQ images were rescaled to 256x256 pixels for FFHQ-256 training. For ImageNet, the shortest side was rescaled to 256 pixels and then center cropped, in alignment with standard practices for ImageNet-256 training. All image latents were precomputed using the Stable Diffusion VAE[3] and included in the shards, which significantly optimized data throughput by 2x. Horizontal flips were also precomputed and stored within the latent tensors as a flip dimension, selected randomly during training with a 50% probability. We note that storing the both flipped and un-flipped versions was necessary, rather than flipping the tensor during loading, as the VAE encodes a directional bias with asymmetric convolution kernels.

Similar to FFHQ and ImageNet, we precompute the latent images for CC3M, however, we do not use horizontal flips for this dataset as some images contain directionality (e.g. text). Given the text-conditioned nature, we store the precomputed text embeddings along side the latents, which are truncated to 16 tokens, of which 70% of all captions are shorter than. Finally, we choose not to train this model with variance matching as only 0.08% of the re-scaled images fall bellow a scale of 1.0 (i.e. are less than 256 pixels on a side), preventing the negative prompt trick discussed in appendix H.1 from being applicable. While dynamic threshold remains an option, we choose to avoid it, as we consider it a sampling trick to improve overall image quality.

## G.1 Augmenting the Diffusion Transformer Blocks

In developing the multi-scale diffusion transformer blocks (MDiT blocks), we integrate elements that reflect recent advancements in vision transformers (ViTs) and large language models (LLMs). These integrations are specifically chosen to enhance computational efficiency and training stability. Each MDiT block (see Fig.2) follows the structured sequence: Multi-Head Self-Attention (MHSA), optional Multi-Head Cross-Attention, and Feed Forward Network, with the following optimizations:

**Bias Removal from Matrices:** Aligning with current best practices in LLMs, we remove biases from all matrices, reducing excess parameter count (Touvron et al., 2023).

**Output Gate Removal:** Following HDiT (Crowson et al., 2024), we omit output gates from each feed-forward and attention layer, reducing parameter count and increasing explainability by enforcing layer participation in the residual stream.

**RMS Normalization:** We adopt Root Mean Square (RMS) normalization for input normalization, which is computationally less demanding than Layer Norm while providing similar benefits (Zhang & Sennrich, 2019).

**GeGLU FFN:** We incorporate Gated Linear Units (GeGLU) (Shazeer, 2020) into our FFNs. As the de facto standard (Rombach et al., 2021; Crowson et al., 2024; Touvron et al., 2023), GLU's enhance control of information flow through their gating mechanism.

**Normalization on Q and K Vectors:** We apply a layer normalization without affine scaling to the Q an K vectors in all attention layers, improving training stability particularly in vision-related tasks (Esser et al., 2024; Dehghani et al., 2023). Layer norm was chosen over RMS norm to enforce a zero mean[4].

**Partial Head Axial-RoPE:** Inspired by GPT-J (Wang & Komatsuzaki, 2021), we implement partial head Rotary Positional Embeddings (RoPE) (Su et al., 2022) to enforce 2D translation invariance, selectively applying positional embeddings to a subset of self-attention head channels. Further discussion follows in Section 4.

A comparison matrix between can be seen in Table 11.

---

[3]We use the `ft-mse-840000-ema` VAE from https://huggingface.co/stabilityai/sd-vae-ft-mse
[4]Layer norm exhibits less performance degradation when applied to the smaller attention head dim.

Table 11: Comparison matrix indicating the impact of each change with '+', '-', '0' representing improvement, decline, and no change, respectively, in terms of Evaluation/Performance (value), Parameters (value), FLOPS (value), and Explainability (effect). A perfect method would be +,-,-,+. $\alpha$ From ablations in Crowson et al. (2024). $\beta$ From ablations Appendix F.

| Change | Evaluation Performance | Parameters | FLOPS | Explainability |
|---|---|---|---|---|
| Bias Removal | - $^{\alpha},^{\beta}$ | - | - | 0 |
| Gate Removal | - $^{\alpha},^{\beta}$ | - | - | + |
| RMS Normalization | - $^{\alpha},^{\beta}$ | - | - | - |
| GeGLU FFN | + $^{\alpha},^{\beta}$ | 0 | 0 | 0 |
| Normalized Q/K Vectors | 0 $^{\beta}$ | 0 | - | + |
| Full Axial RoPE | - $^{\beta}$ | 0 | + | + |
| Partial Axial RoPE | 0 $^{\beta}$ | 0 | 0 | + |

## G.2 INTEGRATING A HYBRID CONDITIONING SCHEME

Most diffusion transformer models utilize adaptive normalization and gating mechanisms for modulating class and timestep information, a method effective yet complex when extending to other conditioning types like text-based inputs (Chen et al., 2024). To simplify this and enhance flexibility, our MDiT architecture incorporates a hybrid conditioning scheme that utilizes cross-attention for class conditioning and a combination of modulation and cross-attention for time. This configuration simplifies the integration of class-specific information and sets a common foundation for text-based conditioning.

Cross-attention conditioning is exclusively applied to the core MDiT blocks due to its computational intensity, which scales with $\mathcal{O}(HW)$. This selective use concentrates semantically rich information within the core, optimizing processing capacity and avoiding semantic dilution across the network. Within these layers, the time-step embedding, class conditioning token, and a null token are concatenated prior to the attention computation. This configuration allows the model to selectively focus on temporal and class information or to ignore both on a per-token basis as proposed by eDiff-I (Balaji et al., 2023).

$$
\begin{aligned}
\mathbf{K} &= \mathrm{LN}\left(\mathbf{n}_K \oplus [\mathbf{c} \cdot \mathbf{W}_{Kc}] \oplus [\mathcal{G}\left(\mathbf{t}\right) \cdot \mathbf{W}_{Kt}]\right) \\
\mathbf{V} &= \mathbf{n}_V \oplus [\mathbf{c} \cdot \mathbf{W}_{Vc}] \oplus [\mathcal{G}\left(\mathbf{t}\right) \cdot \mathbf{W}_{Vt}]
\end{aligned}
\tag{5}
$$

Here, $\mathbf{c}$ represents the class token embedding, $\mathbf{t}$ the time condition token, and $\mathbf{n}$ is the null token. The function $\mathcal{G}(\cdot)$ denotes a GELU activation function, and $\oplus$ symbolizes concatenation in the sequence dimension. In blocks without cross-attention, we maintain a modulated pre-layer RMS norm, aligning with previous implementations (Peebles & Xie, 2022; Esser et al., 2024; Crowson et al., 2024). This modulation is defined by the following equation:

$$
\tilde{\mathbf{x}} = \mathrm{RN}\left(\mathbf{x}\right) \odot [1 + \mathcal{G}\left(\mathbf{t}\right) \cdot \mathbf{W}_t]
\tag{6}
$$

Where $\odot$ is the Hadamard product, $\mathbf{x}$ and $\tilde{\mathbf{x}}$ are the residual activation and layer input, respectively. Additionally, for ImageNet, we enhance the timestep embedding by including normalized aspect ratio and scale data, as proposed by the SDXL (Podell et al., 2024). This modification helps in generating images with a centered subject and conditions against the undersized images prevalent in the dataset. The auxiliary conditioning is integrated globally into the timestep embedding through a shared mapping network.

## G.3 INITIALIZATION

We follow a similar initialization procedure to DiT (Peebles & Xie, 2022), with some changes. All matrices were initialized using a truncated normal distribution with a zero mean and a standard deviation of 0.02. This initialization was applied to the input embedding layers as well, whereas all layer output weights were initialized to zero. The RMS modulation projections $\mathbf{W}_t$ were likewise initialized to zero - similar to Ada-LN-Zero used in DiT.

### G.4 Auxiliary Conditioning

Following the approach in SDXL (Podell et al., 2024), we incorporated auxiliary conditioning in our ImageNet experiments, which leverages statistics from the data preprocessing. Unlike SDXL, which uses image dimensions for conditioning, we opted for aspect ratio and image scale, given our specific preprocessing steps of rescaling and center cropping. Moreover, scale information was utilized only for images that underwent upscaling during preprocessing, with the maximum scale value clamped at 1.0. Following standard practices, the auxiliary conditioning was dropped during training with a probability of 0.1 to facilitate classifier free guidance.

In the model, the scale parameter was encoded using 256 sinusoidal feature channels, with a maximum frequency of 1000, while the aspect ratio was encoded using an identical number of channels but within a frequency range of 500 to 2000. These auxiliary conditions were concatenated with the sinusoidal timestep embedding, which uses 384 channels, resulting in a conditional input size of 896. This combined condition was processed through a 2-layer MLP mapping network featuring a GeGLU activation function. The output of this network serves as a global condition that integrates the diffusion timestep, aspect ratio, and scale parameters. Due to an increased learning rate, we implemented a gradient flow reduction strategy to modulate the learning pace of the mapping network, linearly interpolating between the active and detached (stop grad) outputs, rather than utilizing parameter groups. A similar strategy was applied to the embedding vectors used for class conditioning.

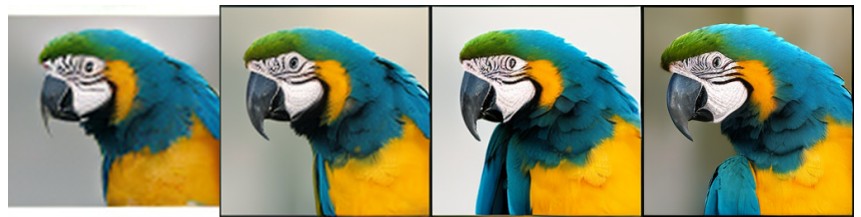

(a) Varying Scale Condition.

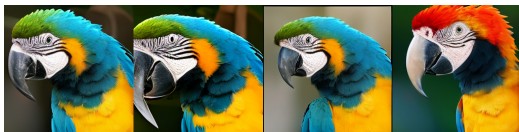

(b) Varying Aspect Ratio Condition.

Figure 19: Comparison of varying the scale and aspect ratio conditions on image generations. Using 100 DDIM sample steps with $\eta = 1.0$ and cfg=3.0 with the MDiT-B model trained for 300k steps on ImageNet-256 without variance matching. (a) shows the effect of varying the scale, with values (left to right) of 0.3, 0.5, 0.75, and 1.0; (b) shows the effect of varying aspect ratio with values (left to right) of 0.0, 1.5, 1.0, and 0.66. The case of 0.0 indicates dropout of both conditions.

Figure 19 illustrates the impact of the scale and aspect ratio conditions on image quality and composition. As depicted in figure 19a, reducing the scale tends to result in blurrier images. Conversely, modifications to the aspect ratio (Fig. 19b) influence the framing of the subject: increasing the aspect ratio leads to horizontal clipping of the subject, whereas decreasing it results in vertical clipping. Maintaining an aspect ratio of 1.0 generally produces images with subjects that are well-centered and more effectively framed compared to those generated when the aspect ratio condition is omitted.

## G.5 AGGREGATE BLOCKS

The Aggregate Blocks, integral to our MDiT architecture as described in Section 3.3 and depicted in Figure 2, are designed to effectively capture and process medium-scale spatial features. Mirroring the down-sampling and up-sampling dynamics found in U-Net architectures, these blocks adapt this concepts for individual transformer blocks, enabling efficient attention processing at varied scales.

Listing 2: Aggregate Block Implementation

```python
def aggregate_block(x, temb, pos_emb_coefs):
    # x input is of shape B,H,W,C

    # 1) Compute modulation scale for inputs to encode timestep
    scale_msa, scale_ffn = adaLN_modulation(temb).chunk(2, dim=-1)

    #2) pixel shuffle the ada_norm output
    #    - downsamle x by 2
    x_down = rearrange(msa_norm(x)*scale_msa,
        'b (h p) (w q) c -> b h w (c p q)', p=2, q=2)

    # 3) apply multi-head self-attention
    #    - applied to downsampled x (i.e. x_down)
    h_msa = mhsa(x_down, pos_emb_coefs=pos_emb_coefs)

    # 4) pixel unshuffle the mhsa output and residual add
    #    - h_msa upsample by 2
    x = x + rearrange(h_msa,
        'b h w (c p q) -> b (h p) (w q) c', p=2, q=2)

    # 4) apply the feed-forward to the original stream
    x = x + ffn(ff_norm(x)*scale_ffn)
    return x
```

The functional details and operational specifics of the Aggregate Blocks are further elaborated in the pseudocode provided in Listing 2. Key design choices for the aggregate blocks and their performance implications are as follows:

**Reduced Computational Complexity:** Utilizing pixel shuffle operations for down-sampling and pixel unshuffle for up-sampling within the self-attention layers effectively reduces the complexity of attention computations from $\mathcal{O}(H^2W^2)$ to $\mathcal{O}(\frac{1}{16}H^2W^2)$. By avoiding down-sampling in the feed-forward layers, we also prevent the additional computational complexity and memory I/O typically associated with larger weight matrices. Despite the added steps of down-sampling and up-sampling, the overall FLOPS required for an aggregate block remain comparable to those of a standard transformer block, as evidenced by the metrics in MDiT-L (3.78G for aggregate vs 3.92G for standard blocks).

**Parameter Efficiency:** By maintaining the original scale in the feed-forward layer while downsampling in the self-attention layers, the parameter count is effectively reduced from $\mathcal{O}(\tilde{D}^2)$ to $\mathcal{O}(\frac{1}{4}\tilde{D}^2)$, where $\tilde{D} = 2D$ following typical downsampling scaling. This reduction is partially offset by the need for non-square Q, K, V, and O matrices, which increase the total parameter count for each block. To minimize this impact, the inner self-attention dimension is scaled by a factor of $1.5\times$, such that $h_A \cdot d_k = 1.5 \cdot d$. However, the resulting Q, K, V, and O matrices are sized at $(4 \times 1.5) \cdot d$, leading to approximately 2.6 times more parameters than would typically be expected for an equivalent capacity increase of $1.5 \cdot d$. These additional parameters function similarly to convolutional weights used in the up/downsampling processes of a U-Net. Specifically, the non-square matrices serve as individual downsampling kernels for Q, K, and V, with a common upsampling kernel for O. Consequently, the extra parameters in these matrices primarily contribute to dimensional transformations rather than adding to computational capacity through increased non-linear processing.

**Enhanced Conditional Focus:** Preliminary experiments incorporating a third U-Net downsampling layer demonstrated limited conditional contribution from the inner layers, particularly in scenarios involving text conditioning. These findings align with those reported in Kim et al. (2023), where the removal of the entire middle layer of the Stable Diffusion U-Net (Rombach et al., 2021) had minimal impact on image quality. By selectively downsampling only the attention layers and strategi-

cally interleaving these blocks within the MDiT core, we effectively add the semantic processing capabilities typical of an additional downsampling layer without incurring the computational overhead typically associated with processing conditioning that would otherwise be underutilized. This approach optimizes the use of computational resources, enhancing the model's ability to focus on relevant conditional information where it contributes most effectively.

### G.6 GUIDELINES FOR PARAMETER SELECTION IN MDiT

To address potential complexity in configuring the MDiT architecture, we provide practical guidelines for parameterization using {M,N,K,L}. These guidelines aim to streamline the design process and ensure computational efficiency while maintaining flexibility.

**Parameterization with {M, N, K, L}**: The architecture's heterogeneity is defined by the set {M, N, K, L}, which controls the distribution of computational resources across the outer and core levels:

- **Constraints on K:** $K$ is restricted to even values, aligning with the repeated block pattern.
- **Balancing Outer and Core Contributions:** The sum $M + N$ (outer levels) is scaled inversely with $K + L$ (core levels) to balance computational load. For Each increment of 2 in $M + N$, $K + L$ is reduced by 1.
- **Balancing Prior and Post Outer Blocks:** We find that the outer blocks following the core are more important for image fidelity, while fewer blocks before the core are necessary to absorb noise and extract low-level features. Empirical tests suggest that a ratio of $N = 2 \cdot M$ works well for the $x_0$, $\epsilon$, and rf objectives.

**Hidden Dimension Scaling:** Hidden dimension scaling follows conventions adapted from U-Nets:

- **Inner to Outer Scaling:** For downsampling layers by $2\times$, the inner dimensions ($d_{\text{inner}}$) are scaled as $d_{\text{inner}} = 2 \cdot d_{\text{outer}}$.
- **Aggregate Block Dimensions:** Unlike typical scaling, we find that a weaker scaling of $d_{\text{agg}} = 1.5 \cdot d_{\text{inner}}$ produces adequate results without incurring excessive computational and parameter overhead. Notably, this is equivalent to scaling the number of aggregate attention heads $d_k$ by a factor of $1.5$.

## G.7 HYPER-PARAMETERS

Table 12: **Details of Training Hyper-parameters.** $^{*}$Using a condition length of 16 text tokens without CFG. $^{\dagger}$ Configurations for noise ($\epsilon$) and Rectified linear Flows (rf). See Appendix C for CC3M-L.

| Parameter | FFHQ | ImageNet-B | ImageNet-L | CC3M-L | ImageNet-XL |
|---|---|---|---|---|---|
| Resolution | 256x256 | 256x256 | 256x256 | 256x256 | 256x256 |
| Parameters | 111M | 137M | 455M | 454M | 572M |
| Fwd. FLOPS | 29.52G | 31.66G | 110.5G | 111.0G$^{*}$ | 148.4G |
| Training Steps | 100k | 400k | 600k | 200k | 1000k |
| Batch Size | 256 | 256 | 256 | 256 | 256 |
| Grad. Accum. Steps | 1 | 1 | 1 | 4 | 1 |
| Grad. Checkpointing | False | False | False | False | False |
| Precision | bfloat16 | bfloat16 | bfloat16 | bfloat16 | bfloat16 |
| Attn. Precision | float32 | float32 | float32 | float32 | float32 |
| Training Hardware | 2xA6000 | 2xA100 | 4xA100 | 2xA6000 | 4xA100 |
| Training Time | 30 Hours | 67 Hours | 185 Hours | 228 Hours | 336 Hours |
| Config. {M,N,K,L} | {2,4,4,5} | {2,4,4,5} | {4,8,8,10} | {4,8,8,10} | {4,9,8,12} |
| Hidden Dim | [384,768] | [384,768] | [512,1024] | [512,1024] | [576,1152] |
| Neighborhood Kernel | [7, -] | [7, -] | [7, -] | [7, -] | [7, -] |
| Attention Heads | [6,12] | [6,12] | [8,16] | [8,16] | [9,18] |
| Aggregate Heads | [-, 18] | [-, 18] | [-, 24] | [-, 24] | [-, 26] |
| Attention Head Dim | 64 | 64 | 64 | 64 | 64 |
| RoPE Dim ($r_{dim}$) | 16 | 16 | 16 | 16 | 16 |
| RoPE Freqency | 16 | 16 | 16 | 16 | 16 |
| FFN Ratio | 2.66 | 2.66 | 2.66 | 2.66 | 2.66 |
| Condition Type | None | Class | Class | T5-FLAN-L | Class |
| Condition Dim | – | 768 | 1024 | 1024 | 1152 |
| Timestep Dim | 384 | 384 | 384 | 384 | 384 |
| Aux Condition Dim | – | 2x256 | 2x256 | 2x256 | 2x256 |
| Global Condition Dim | 512 | 768 | 768 | 768 | 768 |
| Mapping Layers | 2 | 2 | 2 | 2 | 2 |
| Mapping Ratio | 2.66 | 2.66 | 2.66 | 2.66 | 2.66 |
| Mapping Gradient | 0.25 | 0.25 | 0.25 | 0.25 | 0.25 |
| Embedding Gradient | – | 0.25 | 0.25 | – | 0.25 |
| Training Objective | $x_0$ | $x_0$ | $x_0$ | $x_0$ | ($\epsilon, \Sigma$) / rf$^{\dagger}$ |
| Noise Schedule | Cosine | Cosine | Cosine | Cosine | Linear |
| Num Timesteps ($t_{max}$) | 1000 | 1000 | 1000 | 1000 | 1000 / –$^{\dagger}$ |
| Min-SNR-$\gamma$ | 5 | 5 | 5 | 5 | – |
| FFN Dropout Rate | 0.1 | 0.0 | 0.0 | 0.0 | 0.0 |
| Aux Cond. Dropout | – | 0.1 | 0.1 | 0.1 | 0.1 |
| Optimizer | AdamW | AdamW | AdamW | AdamW | AdamW |
| Learning Rate | 4e-4 | 4e-4 | 4e-4 | 4e-4 | 4e-4 |
| Betas | [0.9, 0.95] | [0.9, 0.95] | [0.9, 0.95] | [0.9, 0.95] | [0.9, 0.95] |
| Eps | 1e-8 | 1e-8 | 1e-8 | 1e-8 | 1e-8 |
| Weight Decay | 1e-2 | 1e-2 | 1e-2 | 1e-2 | 1e-2 |
| EMA Decay | 0.9999 | 0.9999 | 0.9999 | 0.9999 | 0.9999 |
| Gradient Clip | 1.0 | 1.0 | 1.0 | 1.0 | 1.0 |
| $\lambda_{\text{VAR}}$ | 0.02 | 0.05 | 0.05 | 0.0 | 0.0 |
| $\lambda_{\text{Patch}}$ | 0.5 | 0.5 | 0.5 | 0.5 | 0.5 |
| Patch Start Step | 100 | 100 | 100 | 100 | 100 |
| Patch End Step | 20k | 100k | 50k | 50k | 50k / 10k$^{\dagger}$ |

Table 13: Details of Probe Hyper-parameters.

| Parameter | Probes ImageNet-256 |
|---|---|
| Training Steps | 50k |
| Batch Size | 128 |
| Precision | float32 |
| Training Hardware | 1xA6000 |
| Training Time | 10 Hours |
| Frozen Backbone | MDiT-B-EMA |
| Pooling | Mean |
| Input Norm | Layer Norm |
| MLP Layers | 2 |
| MLP Ratio | 2 |
| MLP Activation | GELU |
| MLP Bias | True |
| Loss | Cross-Entropy |
| Optimizer | Adam |
| Learning Rate | 2e-3 |
| Betas | [0.9, 0.999] |
| Eps | 1e-8 |
| Weight Decay | 0.0 |
| EMA Decay | N/A |
| Gradient Clip | N/A |
| Test Images | 50k |

## H  VARIANCE MATCHING

In this section, we explore the variance matching regularization technique further by comparing the variance distributions of the FFHQ and ImageNet datasets in Figure 20. We compute the per sample sample variance of each latent channel post-VAE encoding, and aggregate the variance distributions into histograms. This process is repeated for the ground-truth images (validation set for ImageNet), the images generated without variance matching, and the images generated with variance matching.

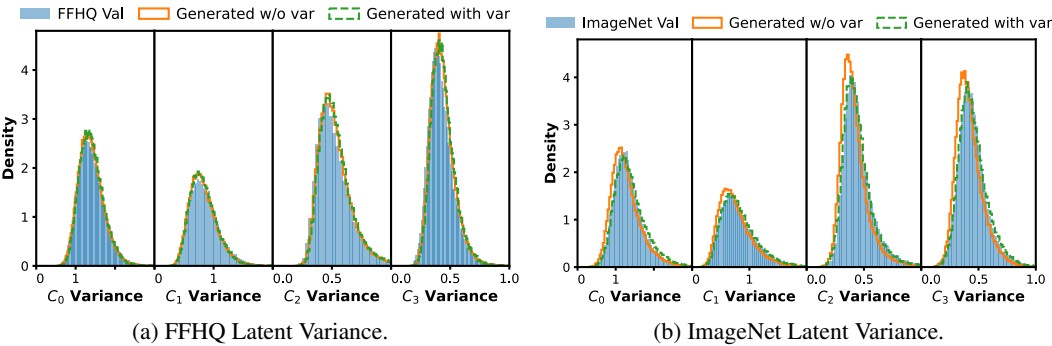

(a) FFHQ Latent Variance.  (b) ImageNet Latent Variance.

Figure 20: Channel variance histogram of FFHQ and ImageNet validation set compared with generated images using only MSE loss and generated with MSE loss + variance matching.

Notably, we observe a distribution shift between the generated samples without variance matching and the true data distribution for both datasets, with a starker deviation on ImageNet. This deviation is reduced when training with variance matching on ImageNet, where the generated distributions much more closely follow the ground-truth image distributions. However, the shift is less obvious with FFHQ, where the largest contribution appears to be a reduction in distribution peak, thereby coming closer to the ground-truth distributions.

## H.1 Fixing the Variance Matching Blur

When comparing conditional samples generated with and without variance matching, a noticeable artifact appears where the images can become slightly blurry, thereby reducing their quality (see Fig.21a). This effect is amplified by larger CFG scales, which we speculate may be related to a know issue with diffusion models trained on the $x_0$ objective, as demonstrated by Saharia et al. (2022). Saharia et al. introduced the Dynamic Thresholding technique to counter an undesirable behavior where high CFG scales can lead generated pixel values to go out of range during the sampling process. Here, the generated pixels are clipped based on the $p$ quantile, and re-scaled so that they remain bounded within [-1,1]. We further extend this method by applying a post-clip scale, so that our pixels (latents) are bounded by [-s, s]. Doing so alleviates the blurring issue, and can bring out more intricate details in the images as can be seen in Figure 21b.

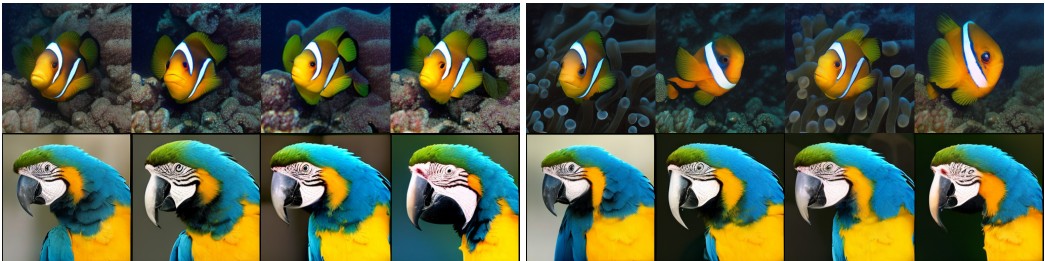

| (a) Baseline Variance Matching. | (b) Variance matching with Dynamic Tresholding. |

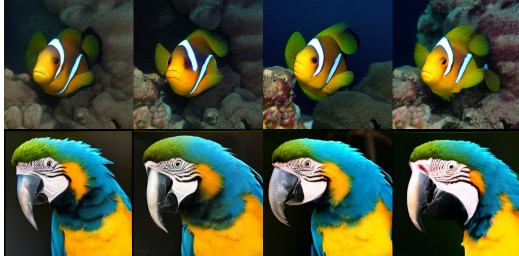

(c) Variance matching with Negative Conditioning.

Figure 21: Comparing image quality for ImageNet-256 on MDiT-B using Dynamic Thresholding and Negative Conditioning to remove the blur caused by variance matching with "high" CFG. Showing impact as a function of $\lambda_{\text{VAR}} = 0.0, 0.02, 0.05, 0.1$ using 100 DDIM steps with $\eta = 1.0$ and cfg=3.0. Samples generated with models trained for 300k steps. Negative conditioning is set at size=75%, and Dynamic Thresholding is set at $p = 0.9, s = 1.4$. Best viewed zoomed in.

As an alternative to Dynamic Thresholding, we considered negative guidance, a popular technique in text-to-image models. This method utilizes the "unconditional" outputs as a target for "what to remove", "blurry-ness" in our case, represented by an image scale below 1.0. Setting this scale too low, such as at 0.5, can induce high-frequency artifacts; however, a less aggressive scale is more effective, as demonstrated in Figure 21c. We implemented this negative guidance across all conditional examples in this paper but did not apply it in our image quality statistics calculations.

## H.2 Application to Rectified Flows

We explored the adaptation of rectified linear flows (RF) using the method proposed by Esser et al. (2024) for DiT-B/2 and MDiT-B, incorporating the recommended importance sampling method. The DiT-B/2 model was trained for 100k steps, while the MDiT-B was trained for 150k steps with and without variance matching regularization. These durations were selected to verify the initial convergence trajectories with those observed under the baseline $x_0$ training method, as demonstrated in Figure 22a, and were evaluated using 50 Euler sampling steps.

The application of RF significantly accelerated the training process, with both MDiT-B and DiT-B/2 achieving approximately $1.5\times$ speedup in convergence compared to the $x_0$ model under Min-SNR. The incorporation of variance matching in MDiT-B further enhanced this effect, yielding a speedup

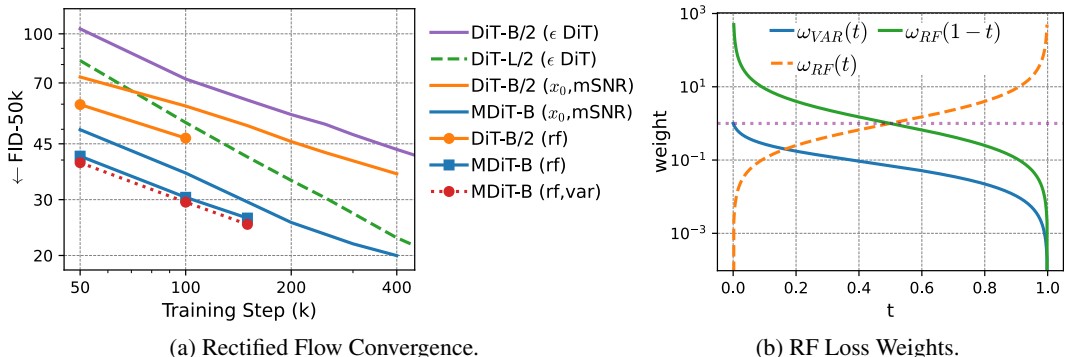

Figure 22: (a) Log-Log FID-50K convergence plots for ImageNet-256. Showing MDiT, DiT baseline with $x_0$ prediction and Min-SNR (mSNR), DiT with $\epsilon$ prediction from Peebles & Xie (2022), and both with Rectified Flows (rf). Evaluated using 50 Euler steps (rf), 50 DDIM steps ($x_0$). (b) Rectified Flow loss weights for variance (VAR) and Rectified Flows (RF). Horizontal purple dotted line marks weight=1.0.

of about $1.6\times$. This improvement underscores the potential of RF, especially when combined with variance matching techniques, to enhance training efficiency and model performance.

In implementing variance matching for the RF framework, where $v_\Theta(y_t, t)$ predicts the velocity to solve the ordinary differential equation $dy_t = v_\Theta(y_t, t)dt$, a direct application of variance matching to the model output is not feasible due to the nature of the predictions. Instead, variance matching is executed indirectly by performing an Euler step to approximate $y_0$ from $y_t$ using $y_0 = v_\Theta(y_t, t)\Delta t$. This step provides a base for applying variance matching directly to $y_0$.

To address the increased Euler error associated with larger $\Delta t$ values and align with the Min-SNR strategy for loss weighting, we incorporate a variance-specific weighting function $\omega_{\text{VAR}}(t)$. Given the complexities introduced by the importance sampling in RF, the weighting function was empirically selected as follows:

$$\omega_{\text{VAR}}(t) = \sqrt{\epsilon}\frac{1-t}{\sqrt{t+\epsilon}} \tag{7}$$

where $\epsilon = 0.01$, providing a bounded and smooth transition similar to the time-reversed loss weighting function $\omega_{\text{RF}}(t) = \frac{1-t}{t}$ suggested by Esser et al. (2024), but with limits $\omega_{\text{VAR}}(0) = 1$ and $\omega_{\text{VAR}}(1) = 0$. This design ensures that the variance matching is more heavily weighted to less noisy images where the Euler step error is smaller. Figure 22b compares the forward and reverse versions of $\omega_{\text{RF}}(t)$ along with the weighting function $\omega_{\text{VAR}}(t)$. We further note that the adjustment of the variance matching loss weight similarly requires a higher regularization weight of $\lambda_{\text{VAR}} = 0.1$, as used in Figure 22a, but was not ablated for an optimal value.

# I  EXTRAPOLATING ASPECT RATIO WITH RoPE

A natural question when using RoPE position embeddings is whether or not the model is capable of extrapolating beyond the training sequence length. We find the answer to this question is: yes, with several caveats. Namely, FFHQ is unable to extrapolate. ImageNet can, however, extrapolating beyond a certain point leads to image degradation. The image degradation is likely due to a self-attention logit scale discrepancy as proposed by Crowson et al. (2024), and supported by quality improvements when switching to neighborhood attention. It should be noted that this section does not consider theta re-scaling as proposed with LLM, and only considers out of distribution extrapolation.

## I.1  IMAGENET UNIFORM EXTRAPOLATION

We evaluate the model's capability for uniform extrapolation on square images scaled beyond the nominal training resolution, testing both standard full self-attention and configurations where

traditional self-attention blocks in the MDiT core are replaced with neighborhood self-attention (NATTEN). The kernel size for neighborhood attention is set to k=15, closely aligning with the 16x16 image tokens used during training, facilitating this as a *drop-in* solution *without* necessitating fine-tuning. Comparisons at the original resolution of 256x256 between standard attention (Figure 23a) and neighborhood attention (Figure 23b) demonstrate minimal visible quality loss, confirming the effectiveness of neighborhood attention in maintaining image quality at trained resolutions.

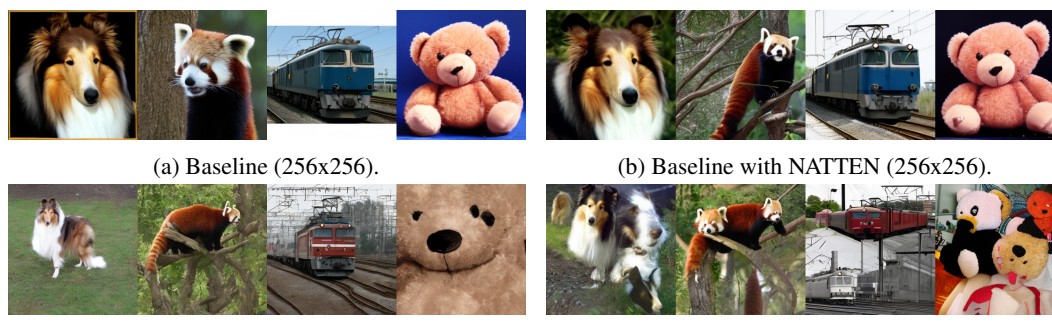

(a) Baseline (256x256).             (b) Baseline with NATTEN (256x256).

(c) Baseline bigger (384x384).             (d) Baseline bigger with NATTEN (384x384).

Figure 23: Comparison of extrapolating samples on the ImageNet MDiT-B model, trained for 400k steps without variance matching. (a) showing the baseline samples at the training resolution, (b) showing no degradation when replacing the MDiT core self-attention layers with neighborhood attention, (c) showing gamut quality degradation when scaling up, (d) showing no gamut quality degradation when scaling up with neighborhood attention, but twinning occurs.

As images are uniformly scaled to 384x384, we begin to observe significant differences. Figures 23c (standard attention) and 23d (neighborhood attention) illustrate the effects of this scaling. Notably, "twinning" artifacts appear under neighborhood attention due to the reduced attention window size, echoing challenges noted in convolutional diffusion models like Stable Diffusion. Additionally, standard self-attention exhibits a noticeable reduction in color vibrancy at this enlarged scale, likely due to logit scaling issues as the model adjusts to a greater number of tokens (576 instead of 256). This scaling challenge, articulated by Crowson et al. (2024), suggests that models originally trained with a certain token count face difficulties when adapting to significantly different scales. Conversely, neighborhood attention, by adhering more closely to the original training token count (225), appears to better manage these challenges. This observation leads us to hypothesize that non-uniform scaling - adjusting images to aspect ratios like 3:2 or 2:3 - might result in less quality degradation compared to uniform scaling, as the effective token count could align more closely with training conditions.

Furthermore, the model displays an ability to maintain structure and composition at enlarged resolutions when using full self-attention. This observation is particularly significant given that it was trained on 256x256 center-cropped ImageNet images. Although these crops generally center the subject, they often truncate peripheral details, leaving out information about the edges. The model's capability to "fill in" these missing areas is likely enabled by the use of Axial RoPE, which enforces translation invariance. Translation invariance allows the model to learn and utilize relative positional information of the subjects, which varies due to the different orientations and positioning within the training samples. This mechanism mirrors the reconstructive capabilities seen in Wang et al.'s work on Patch Diffusion (Wang et al., 2023), where small random crops were used to reconstruct larger images during inference. Similarly, our model treats center crops as partial views of a larger context, thus demonstrating a comparable ability to reconstruct beyond the trained image bounds, leveraging the relative positional cues encoded by Axial RoPE - a task that would pose significant challenges with traditional learned embeddings.

## I.2 IMAGENET EXTREME EXTRAPOLATION WITH NATTEN

Building on the insights from the previous section, we investigate whether neighborhood attention, despite its associated "twinning" artifacts, can effectively handle extreme resolutions while preserving image quality. We specifically examine how the model performs at three distinct resolutions: 1024x256, 256x1024, and 1024x1024. The results demonstrate that neighborhood attention can

effectively manage large scale extrapolations, particularly in scenarios where consistent visual patterns or elements are present. The model effectively extends natural gradients and maintains global consistency, employing an "auto-complete" behavior that uses local cues to generate plausible scene continuations.

However, at the largest tested resolution, complex scenes such as those involving multiple interacting elements, show the inability to maintain spatial coherence, as some elements might merge unnaturally while individual subjects remain distinct. This behavior is linked to the previously observed "twinning", in which the local window remains plausible, but the global context is not taken into consideration.

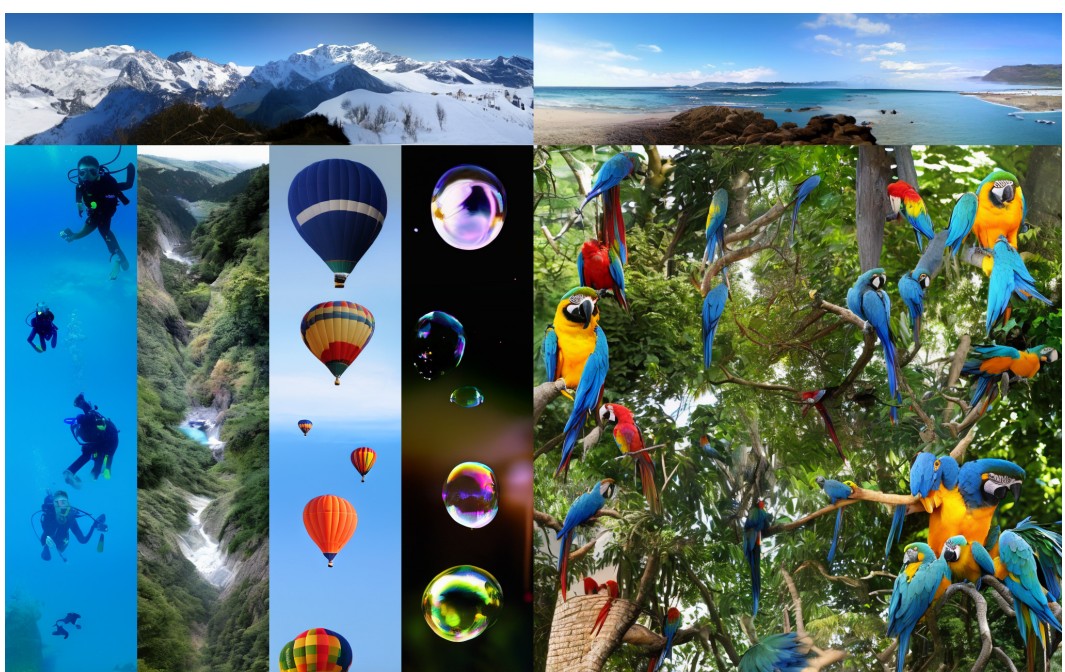

Figure 24: Extreme extrapolation samples on MDiT-B using neighborhood attention in the MDiT core. Showing horizontal images (1024x256), vertical images (256x1024), and a square image (1024x1024). Images were generated without finetuning or training beyond the original 400k steps at 256x256 resolution.

### I.3 IMAGENET NON-UNIFORM EXTRAPOLATION

We explore the effects of non-uniform image scaling by initially examining a scale increase from 256x256 to 320x320, which raises the MDiT core token count from 256 to 400. These images, as demonstrated in Figure 25, avoid the vibrancy loss seen in previous larger scale experiments (e.g., to 384x384) and retain better overall subject composition, benefiting from additional surrounding pixels that provide more contextual information.

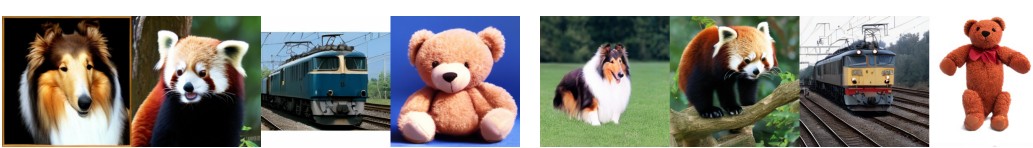

(a) Baseline (256x256).  (b) Square Scaling (320x320).

Figure 25: Comparison with square scaling to lower token count (320x320 = 400) on the MDiT-B model. Generated using 100 DDIM steps, $\eta = 1.0$, cfg=4.0, full MHSA in the MDiT core.

Further experiments investigate changes in aspect ratios, analyzing how varying the aspect ratio condition parameter affects image composition. Figure 26 for wider images (384x256 = 384 tokens)

shows that a higher aspect ratio condition results in wider shots, suggesting an expansion in scene context, while a lower condition emphasizes more closeup shots. For taller images, as shown in Figure 27 (256x384 = 384 tokens), the model is more prone to image distortion and unnatural cropping, particularly as the aspect ratio condition deviates further away from unity. Notably, the impact of scaling is class and seed dependent; for instance, "red panda" shows little variation with changes in scale and aspect conditioning, whereas "teddy bear" fails to form coherent images under any condition, highlighting the variability in scaling effectiveness across different subjects.

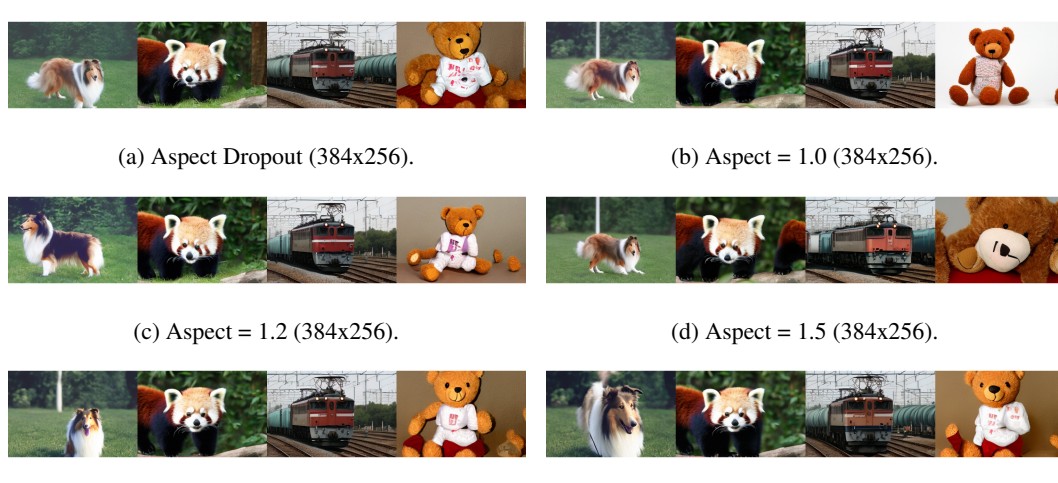

(a) Aspect Dropout (384x256).  (b) Aspect = 1.0 (384x256).

(c) Aspect = 1.2 (384x256).  (d) Aspect = 1.5 (384x256).

(e) Aspect = 0.8 (384x256).  (f) Aspect = 0.6 (384x256).

Figure 26: Comparison of a wider physical aspect ratio (384x256 = 384 tokens) as a function of aspect ratio condition for the MDiT-B model. Generated using 100 DDIM steps, $\eta = 1.0$, cfg=4.0, full MHSA in the MDiT core.

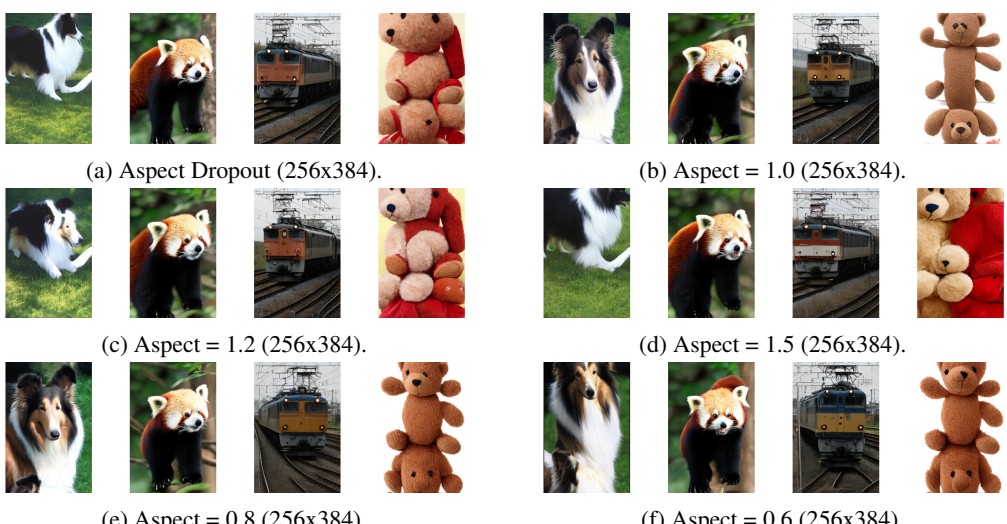

(a) Aspect Dropout (256x384).  (b) Aspect = 1.0 (256x384).

(c) Aspect = 1.2 (256x384).  (d) Aspect = 1.5 (256x384).

(e) Aspect = 0.8 (256x384).  (f) Aspect = 0.6 (256x384).

Figure 27: Comparison of a taller physical aspect ratio (256x384 = 384 tokens) as a function of aspect ratio condition for the MDiT-B model. Generated using 100 DDIM steps, $\eta = 1.0$, cfg=4.0, full MHSA in the MDiT core.

### I.4 FFHQ'S FAILURE TO EXTRAPOLATE

Exploring the extrapolation capabilities with the FFHQ dataset yields distinctly different results compared to ImageNet. Applying neighborhood attention to the FFHQ MDiT-B model, as shown

in Figure 28b, leads to significant visual changes; some images maintain coherence, while others exhibit pronounced artifacts. This outcome suggests a reliance on edge tokens for storing essential scene-specific information, which is compromised when the kernel size is reduced to 15, smaller than the original training size of 16.

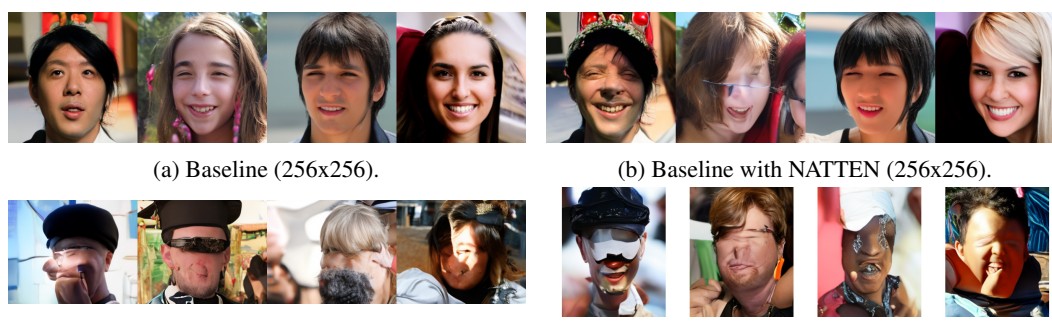

(a) Baseline (256x256).  (b) Baseline with NATTEN (256x256).

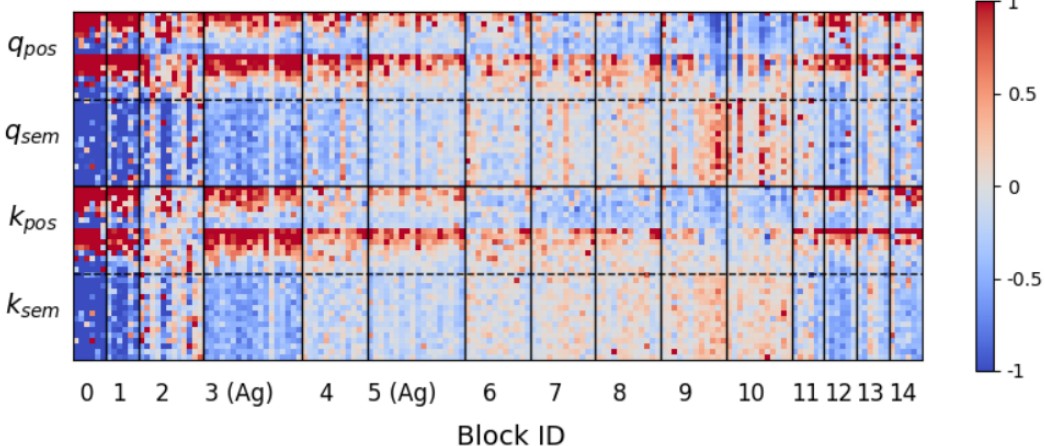

(c) Baseline wider (320x256).  (d) Baseline taller (256x320).

Figure 28: Comparison of extrapolating samples on the FFHQ MDiT-B model, trained for 100k steps. (a) showing the baseline samples at the training resolution, (b) showing degredation when replacing the MDiT core self-attention layers with neighborhood attention, (c) showing failure to extrapolate wider images, (d) showing failure to extrapolate taller images.

Further attempts to generate taller and wider images using the original self-attention mechanism, without switching to NATTEN, consistently result in distorted images. These outputs, particularly seen in Figures 28c and 28d, are notably marred by artifacts concentrated on facial features. Interestingly, details such as hair, headwear, and clothing are less affected. This pattern, demonstrated across the examples in Figure 28, suggests that the model has learned a strong bias towards absolute positions as well as an anisotropic bias, which influences how extrapolation is handled based on the image dimensions and content orientation.

Figure 29: Complex magnitude ($|| \cdot ||^2 - 2$) of Q and K vectors of the MHSA heads for the MDiT-B FFHQ model. Red and Blue indicates strong and weak activation, respectively. The aggregate blocks are marked with "(Ag)", which visually have more attention heads (wider) than the other blocks. Similarly blocks 0,1 are input blocks (M=2), and blocks 11,12,13,14 are output blocks (N=4), both sets having half as many heads as the core blocks. Per vector channels are (from top to bottom): x-position, y-position, and semantic features.

Further insights into the anisotropic behavior of the model are substantiated by an analysis detailed in Section 4.1, which examines the model's focus across self-attention heads and channels. This analysis, visualized in Figure 29, clearly shows the model's differential focus on positional versus semantic information and reveals a distinct emphasis on the y-axis over the x-axis. Such a focus

pattern aligns with the observation that facial features like hair, headwear, and clothing - which exhibit less variability along the x-axis and are localized in specific regions along the y-axis - are less affected by distortion. This behavior likely stems from an over-reliance on the regular positioning of features such as eyes, mouth, and nose in the FFHQ dataset, a pattern less prevalent in more varied datasets like ImageNet. The distinct spatial focus not only explains the model's handling of extrapolation but also highlights intrinsic dataset characteristics that shape learning outcomes.

## I.5 Visualizing Axial RoPE Focal Patterns

An alternative method for understanding the anisotropy present in the complex magnitude analysis is to directly visualize the Axial RoPE focus patterns. This approach is similar to visualization of learned position embeddings, where instead of learning direct bias shifts, Axial RoPE effectively learns Fourier amplitudes for a 2-D harmonic series. These amplitudes are directly linked to the complex magnitudes, and can then be used to reassemble the 2-D series by summing the contributions in the frequency space and then taking a Fourier transform back into image (token) space. The resultant FFHQ and ImageNet focal patterns for the two MDiT-B models are illustrated in figure 30.

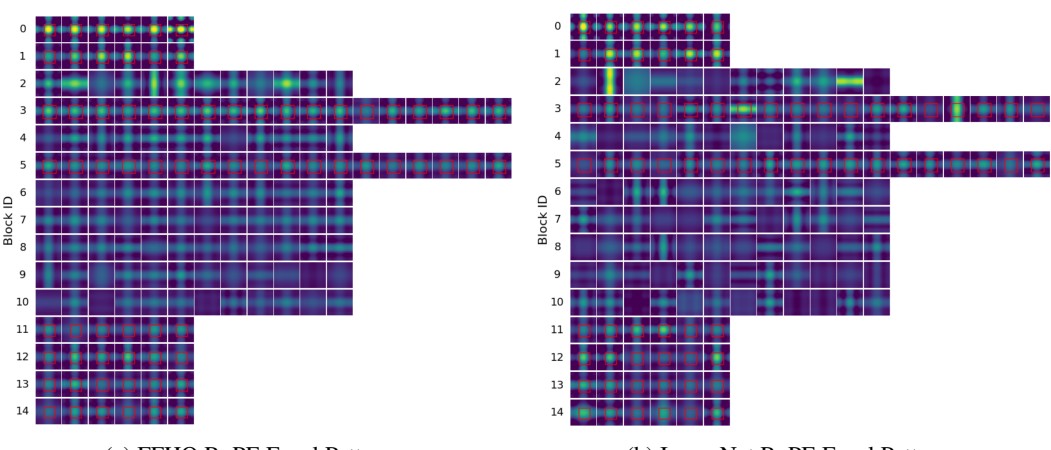

(a) FFHQ RoPE Focal Patterns.       (b) ImageNet RoPE Focal Patterns.

Figure 30: Comparing Axial RoPE query vector focus patterns between (a) MDiT-B trained on FFHQ, and (b) MDiT-B trained on ImageNet. The attention heads are arranged horizontally, plotting the focal pattern for a centered image token. All patterns are plotted for 16x16 tokens with red squares representing the attention windows for neighborhood attention in the outer blocks (ID=0,1,11,12,13,14) and the aggregate blocks (ID=3,5). Plots are normalized to the amplitude range of [0, 2.0].

Similar to the previous subsection, the anisotropy in the FFHQ model (Fig. 30a) can be clearly seen by a strong representation in the vertical direction (horizontal bars). Conversely, the ImageNet model (Fig. 30b) has a more even distribution of focal patterns, balancing horizontal and vertical focus, along with isotropic focus (as seen by plus-shaped patterns). Furthermore, the shift from spatial to feature and hybrid focus can be observed in the magnitude of the focal patterns for each head, stronger at the inputs and becoming weaker deeper in the MDiT core.

## J Efficient Finetuning for Larger Resolutions

### J.1 Leveraging the Aggregate Blocks

Building on our model's demonstrated ability to extrapolate to larger image sizes using neighborhood attention, we explored a targeted finetuning strategy to further improve image coherence. This approach is focused on leveraging NATTEN for its efficiency in the MDiT core's self attention blocks while relying predominantly on the aggregation blocks to establish and maintain the global structure of the images. Our finetuning procedure resumes from the 400k training step checkpoint of our MDiT-B model, which was initially trained without variance matching. We then replace the multi-head self-attention (MHSA) layers of the MDiT core with neighborhood attention, using a

kernel size of k=15, and freeze *all* parameters *except* for the aggregation blocks, of which there are *two*. Training is then continued at a resolution of 384x384, up from the original 256x256.

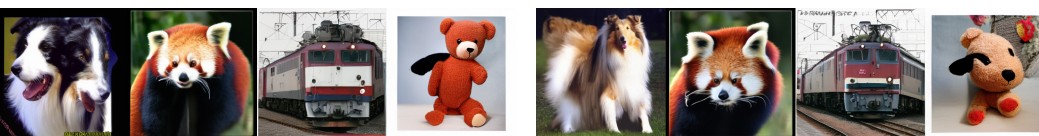

(a) 384x384 with 5k finetune steps.      (b) 384x384 with 30k finetune steps.

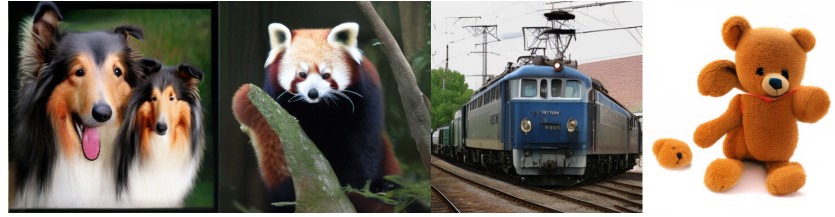

(c) 448x448 with 30k finetune steps.

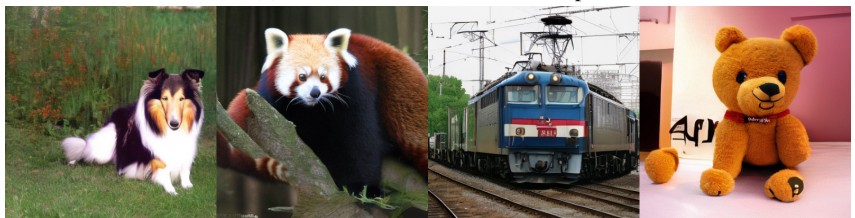

(d) 448x448 + aspect condition, with 30k finetune steps.

Figure 31: Comparing generated results after only finetuning MDiT-B's aggregate blocks on larger resolutions. (a) and (b) show the results at 384x384 after 5k and 30k finetune steps, respectively. These should be compared with Figure 23d. (c) and (d) show the results at 448x448 pixels after 30k steps, where (c) uses an aspect condition of 1.0, and (d) uses 1.5. All samples are generated with 100 DDIM steps, $\eta = 1.0$, and cfg=4.0.

This finetuning process was conducted over 30,000 steps but was stopped early due to computational constraints. Remarkably, significant improvements in global consistency were observed as early as 5,000 training steps, equivalent to one epoch, as illustrated in Figure 33. By this early stage, the images already demonstrated enhanced structural coherence and a notable reduction in common artifacts such as twinning, which had been more prevalent in the baseline model shown in Figure 23d. We further demonstrate that these improvements extend to the larger 448x448 resolution. However, despite the advancements, some samples at 30k steps still exhibit global inconsistencies. These can be partially mitigated by applying a wider aspect ratio condition during sampling, as seen in Figure 31d, which helps further enhance the structural integrity of the images.

In further exploring the impact of finetuning on different aspect ratios, as documented in Figure 32, we observe that finetuning leads to improved support for more extreme aspect ratios (2.0 and 0.5). However, some notable inconsistencies remain, such as the occasional appearance of duplicated elements within a single frame. Despite these issues, the outcomes for less extreme aspect ratios closely align with those exhibited in Appendix I.3, but at an increased resolution of 1.25x, demonstrating that the model can handle larger resolutions with enhanced consistency compared to previous capabilities. While not flawless, these outcomes show significant promise given the constraints of the model size and the relatively few training steps undertaken. These findings suggest that a larger model equipped with more than two aggregation blocks would likely yield better performance, particularly with extended finetuning. Such enhancements could further improve the model's ability to accurately handle varying image dimensions, reinforcing the potential of our architecture for scalable, high-resolution image processing tasks.

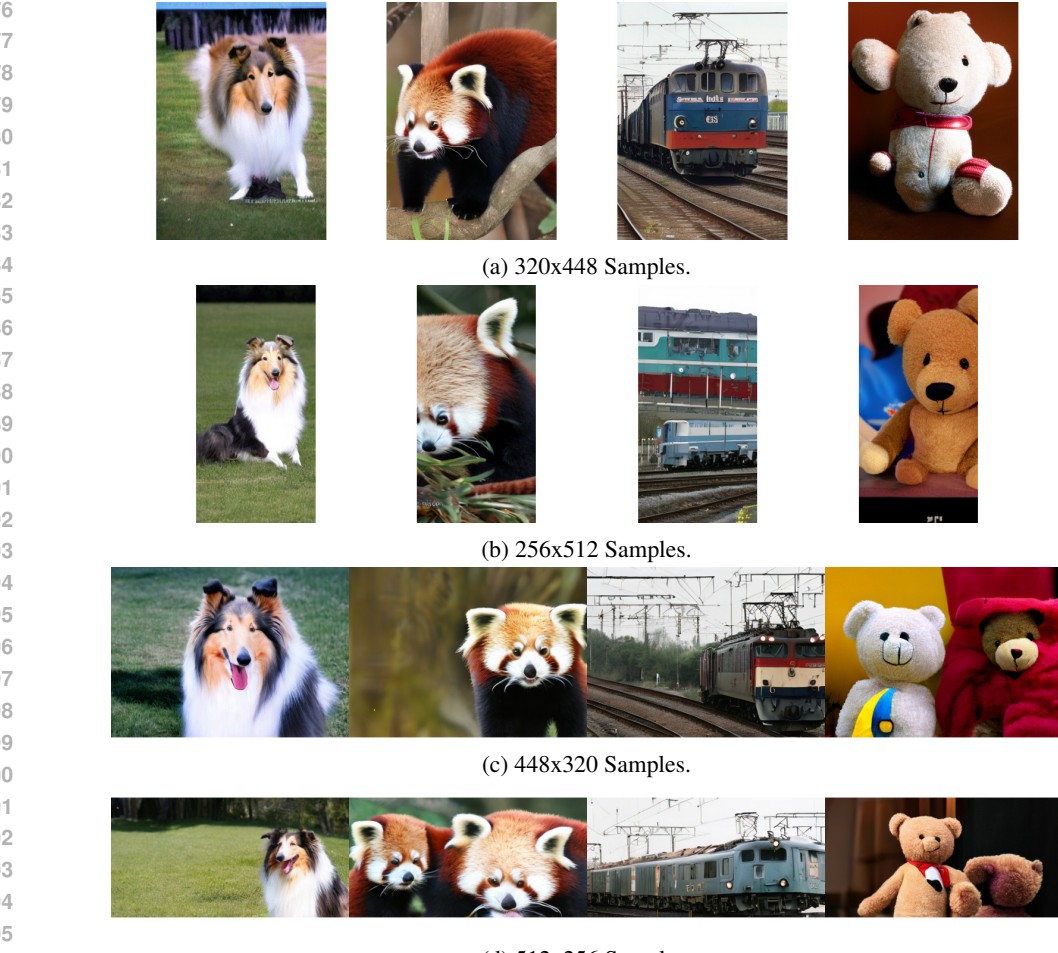

(a) 320x448 Samples.

(b) 256x512 Samples.

(c) 448x320 Samples.

(d) 512x256 Samples.

Figure 32: Comparing generated results after only finetuning MDiT-B's aggregate blocks on larger resolutions for 30k steps. (a) and (b) show taller aspect ratios at 320x448 and 256x512, respectively. (c) and (d) show wider aspect ratios at 448x320 and 512x256, respectively. All samples are generated with 100 DDIM steps, $\eta = 1.0$, and cfg=4.0.

## J.2 ADAPTING THE CORE PATCH LAYERS

We further explore the extension of the MDiT to larger resolutions, building on the successful outcomes demonstrated in earlier sections. Given MDiT's U-Net-like structure, which shares similarities with models such as Stable Diffusion (Rombach et al., 2021) and SDXL (Podell et al., 2024), we investigate the applicability of the HiDiffusion technique (Zhang et al., 2024) to our architecture. HiDiffusion enables the generation of high-resolution images by adapting U-Net down/upsampling according to a resolution schedule, thereby remaining remaining at the final resolution for all inference steps. In the initial phase, covering the first $p \cdot T$ of the total $T$ timesteps, the technique adapts the first down/up sampling layers to employ a $4\times$ resolution change, allowing the inner U-Net layers to function at their native training resolution, thus facilitating large-scale structural and semantic development early on. For the remaining $(1 - p) \cdot T$ timesteps, it reverts to the original $2\times$ resolution change, focusing primarily on fine-detail refinement. This approach allows the model to efficiently achieve high-resolution inference at much higher resolutions than the model was originally trained on, without additional finetuning.

Following the approach in HiDiffusion, we adapted the patch embed/decode layers of the MDiT-core using average pooling and bi-linear interpolation to facilitate the $4\times$ resolution change. Similar to the previous sections, we applied NATTEN to all of the self-attention layers, enabling full-scale

detail refinement at the final inference steps. While this approach did improve global structural coherence compared to using NATTEN alone, it resulted in ghosting behavior and occasional subject duplication, as illustrated in Figure 33a. Notably, we were unable to find a combination of pooling, interpolation, and resolution schedule $p$ that prevented such artifacts. Consequently, we explored the idea of patch replication without the resolution schedule, which behaves similarly to average pooling and nearest neighbor upsampling. This adjustment allowed the model to generate images with a $4\times$ resolution change when entering and exiting the MDiT-core. While this approach corrected the ghosting artifacts, it introduced new ones, as can be seen in Figure 33b.

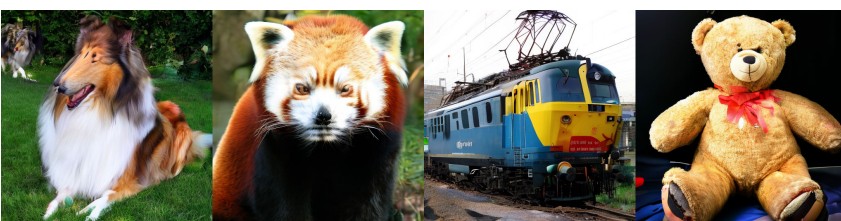

(a) 512x512 with HiDiffusion.

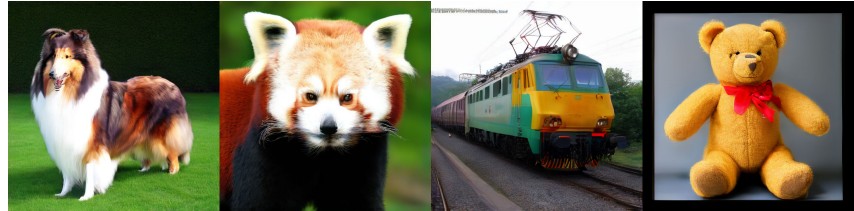

(b) 512x512 with 4x4 Repatching.

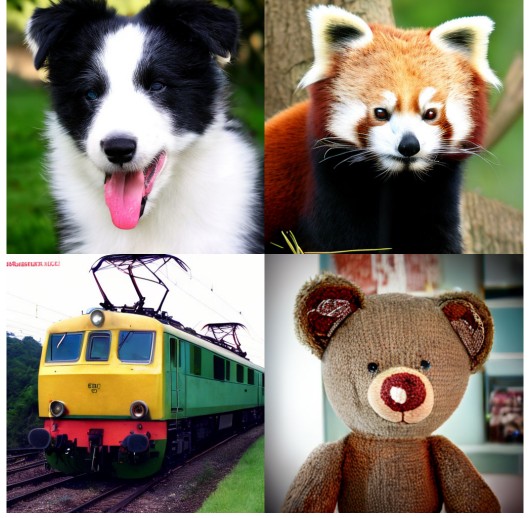

(c) 512x512 with 4x4 Patch Finetune.          (d) 512x512 with Finetuned HiDiffusion.

Figure 33: Comparing different stages in the 512x512 MDiT-L finetune process with HiDiffusion. a) The initial attempt without any finetuning. b) The effect with expanding the MDiT-Core patch embeddings to 4x4. c) Repeating (b) after 1 epoch of finetuning. d) reapplying HiDiffusion after the finetune process. All samples are generated with 100 DDIM steps, $\eta = 1.0$, and cfg=4.0. Best viewed zoomed in.

Given the sub-optimal results from simple patch replication, we pursued a finetuning strategy, where all weights were frozen *except* for the patch embed/decode projection matrices at the entry and exit points of the MDiT-core. This adjustment began from a MDiT-L checkpoint that employed patch replication, with further training conducted at a 512x512 resolution on ImageNet for 5k steps (equivalent to one epoch). Remarkably, this limited finetuning proved sufficient to rectify the artifacts seen in Figure 33b, resulting in images at 512x512 resolution that matched the quality of the original

256x256 outputs, as demonstrated in Figure 33c. To build on this, we implemented HiDiffusion by adjusting the patch factor and projection matrices according to the resolution schedule $p$, post-finetuning. The results, depicted in Figure 33d, indicate significant enhancements in fine-detail rendering compared to earlier attempts, though some image artifacts persisted. Notably, we did not optimize the resolution schedule after finetuning, which might resolve these remaining issues.

The effective application of HiDiffusion and patch duplication techniques depends significantly on the U-Net-like structure of our MDiT architecture with neighborhood attention, differentiating it from homogenous transformers such as DiT (Peebles & Xie, 2022) and SD3 Esser et al. (2024). Contrasting with methods using traditional 4x4 patch embeddings, as seen in DiT, our approach results in minimal perceivable quality loss during 4x4 down/up sampling. This is due to the inclusion of outer layer blocks that enhance the encode/decode capacity of the transformer, as described in Section 3.2. Furthermore, this method is orthogonal to the aggregate block finetuning discussed previously and could potentially be combined with independent, parallel training to achieve resolutions up to 1024x1024 without ever exceeding a training resolution of 512x512. However, exploration of this potential was limited by our training budget, highlighting an area for future research.

## K    ATTENTION PROBING FOR ROPE-BASED LLMS

Building on the analysis introduced in Section 4.1, we apply the complex vector attention probing techniques to the GPT-J-6B model (Wang & Komatsuzaki, 2021), the initial inspiration for the partial-head RoPE mechanism. This model serves as a valuable case study for evaluating the adaptability of our findings from diffusion transformers to large language models (LLMs). The results, illustrated in Figure 34, show that the majority of attention heads in GPT-J-6B focus either on semantic information or a combination of semantic and positional information (hybrid heads). This pattern is especially pronounced in the first layer and the final six layers, indicating a systematic variation in the encoding of information across layers.

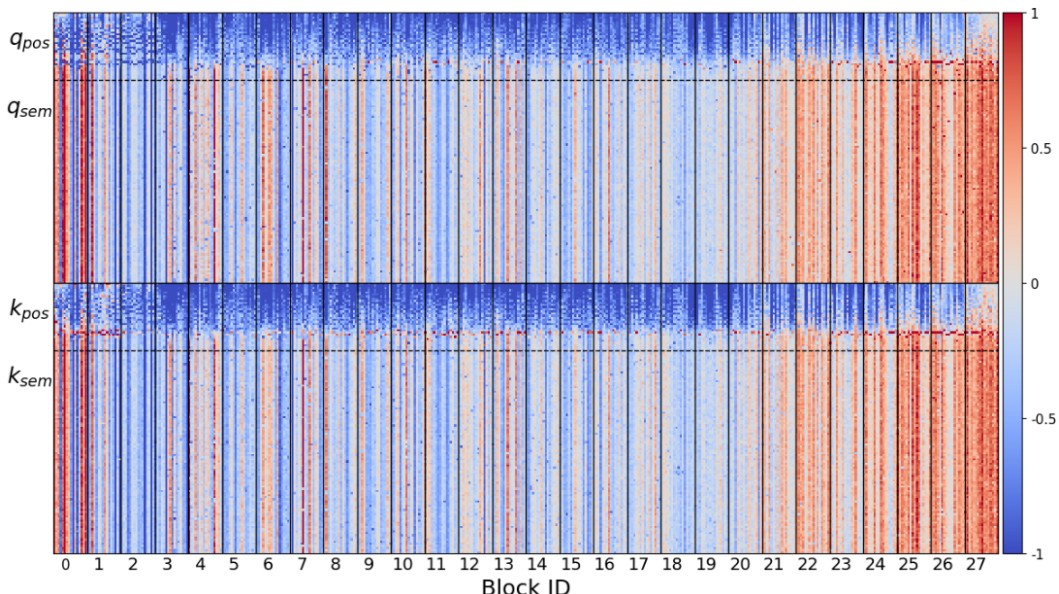

Figure 34: Complex magnitude ($||\cdot||^2 - 2$) of Q and K vectors of the MHSA heads for GPT-J-6B (Wang & Komatsuzaki, 2021). Red and Blue indicates strong and weak activation, respectively. Each attention layer has $d_{head} = 256$ and $r_{dim} = 32$. The RoPE frequencies have a channel cutoff $d = 25$, corresponding to the transition.

Furthermore, we observe a distinct boundary between position and semantic focus below the partial head boundary of $r_{dim} = 32$, likely due to the RoPE frequency cutoff around channel 25. For channels between 25 and 31, there is minimal variation across the model's trained context window of 2048 tokens, suggesting the model allocates these channels to primarily encode semantic information. This phenomenon, also detectable in our MDiT model, is more pronounced in GPT-J due to the lower cutoff channel. These findings provide insight into the behavior of RoPE-based LLMs that does not adopt a partial-head mechanism, where $r_{dim} = d_{head}$.

### K.1    LONG CONTEXT FINE-TUNING EFFECTS

Expanding on the hypothesis from the previous section, which suggests distinct behaviors in RoPE-based LLMs without a partial-head mechanism where $r_{dim} = d_{head}$, we apply these principles to the Llama-3 model (AI@Meta, 2024). If our hypothesis is accurate, we anticipate observing several key phenomena: 1) a smooth transition in activation strength from low to high frequency, akin to the Q/K position region seen in Figure 34; 2) stronger complex magnitudes in higher head channels (longer RoPE frequencies), given the pronounced semantic behavior in GPT-J; and 3) a shift in activation patterns when comparing the base Llama-3-8B model, trained with an 8k token context, to a version fine-tuned with a 1040k token context, with more significant changes in higher frequency channels that become meaningful within the larger context.

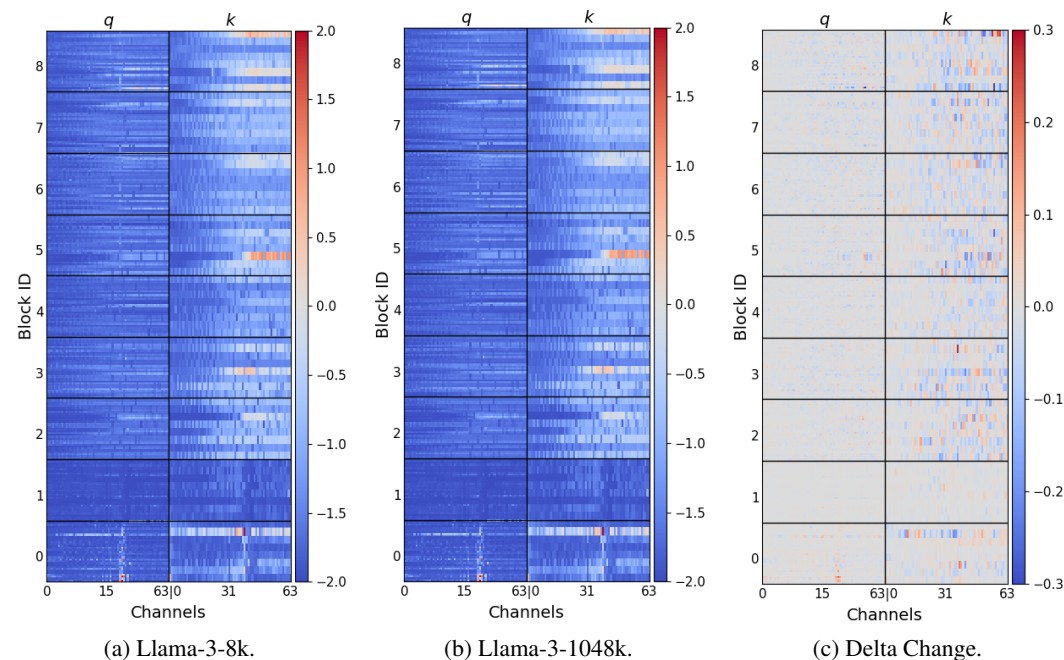

(a) Llama-3-8k.  (b) Llama-3-1048k.  (c) Delta Change.

Figure 35: (a-b) Complex magnitude ($|| \cdot ||^2 - 2$) of Q and K vectors of the first 8 MHSA layers for the Llama-3-8B and Llama-3-8B-1040k models. Red and Blue indicates strong and weak activation, respectively. (c) Comparing the change in complex magnitudes between the two models.

The empirical validation of these predictions is illustrated through a comparative analysis between the 8k[5] and 1040k[6] context configurations in Llama-3. The results for the first eight layers, depicted in Figure 35 and the delta changes in activation strength shown in Figure 35c, confirm our hypotheses. The complex magnitude difference in higher frequency channels are indeed more pronounced in the fine-tuned model, suggesting that these RoPE frequencies, which are adapted to longer distances, become more influential within an expanded context. These findings not only emphasize the utility of this explainability method but also provide a potential explanation for the frequent failures in naive extrapolation of RoPE-based LLMs beyond their training context and the relative success of fine-tuned models.

---

[5]We used the model from https://huggingface.co/meta-llama/Llama-3-8B-Instruct
[6]We used the model from https://huggingface.co/gradientai/Llama-3-8B-Instruct-Gradient-1048k

## L    MORE IMAGE SAMPLES

### L.1    RANDOM FFHQ

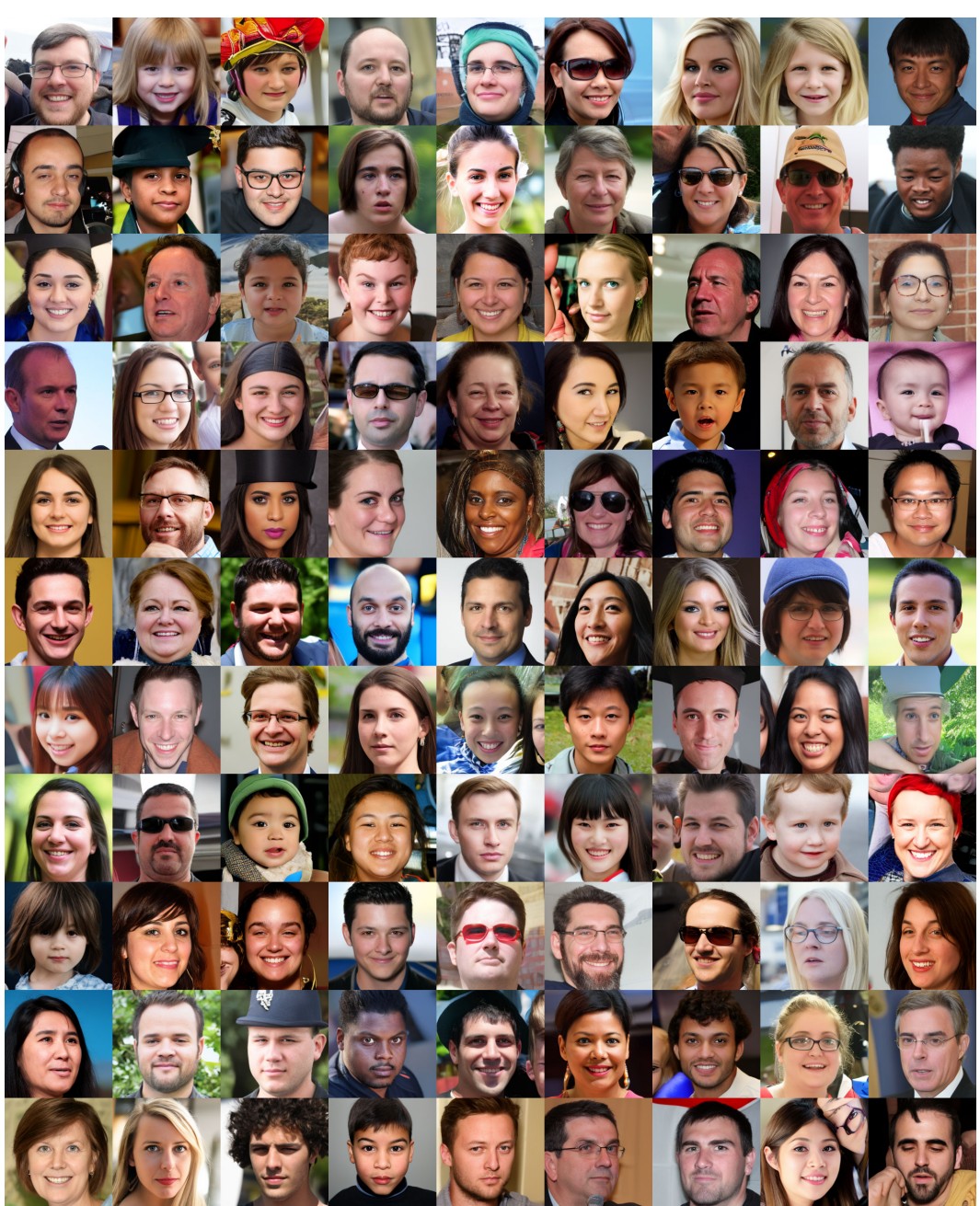

Figure 36: **Uncurated FFHQ-256x256 samples.** Generated with 100 DDIM steps using $\eta = 1.0$.

## L.2 RANDOM IMAGENET

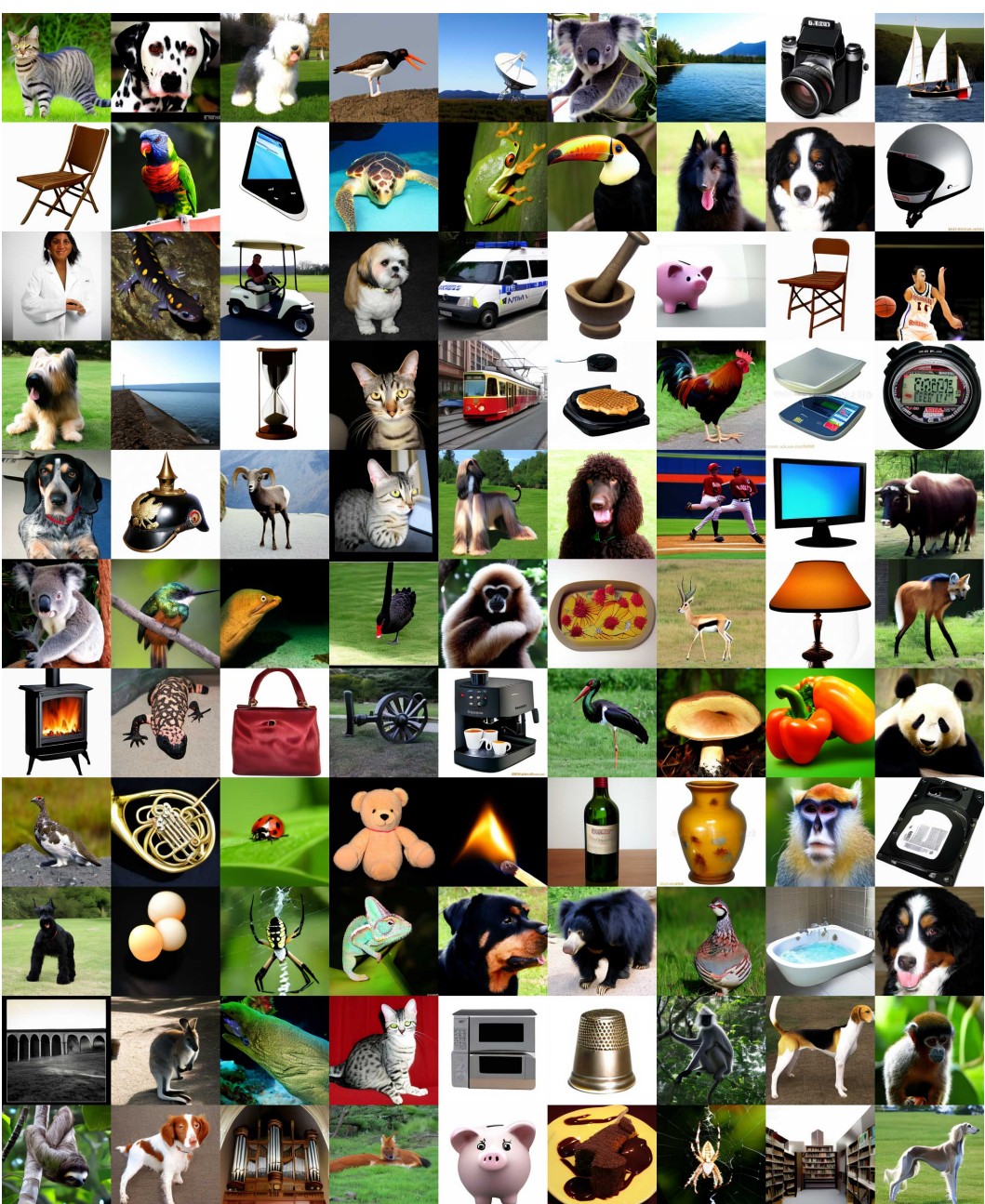

Figure 37: **Uncurated ImageNet-B-256x256 samples with random classes.** Generated using the MDiT-B model at 400k training steps, and 100 DDIM steps using $\eta = 1.0$, cfg=4.0, and negative scale conditioning to negate the variance matching blur.

2754
2755
2756
2757
2758
2759
2760
2761
2762
2763
2764
2765
2766
2767
2768
2769
2770
2771
2772
2773
2774
2775
2776
2777
2778
2779
2780
2781
2782
2783
2784
2785
2786
2787
2788
2789
2790
2791
2792
2793
2794
2795
2796
2797
2798
2799
2800
2801
2802
2803
2804
2805
2806
2807

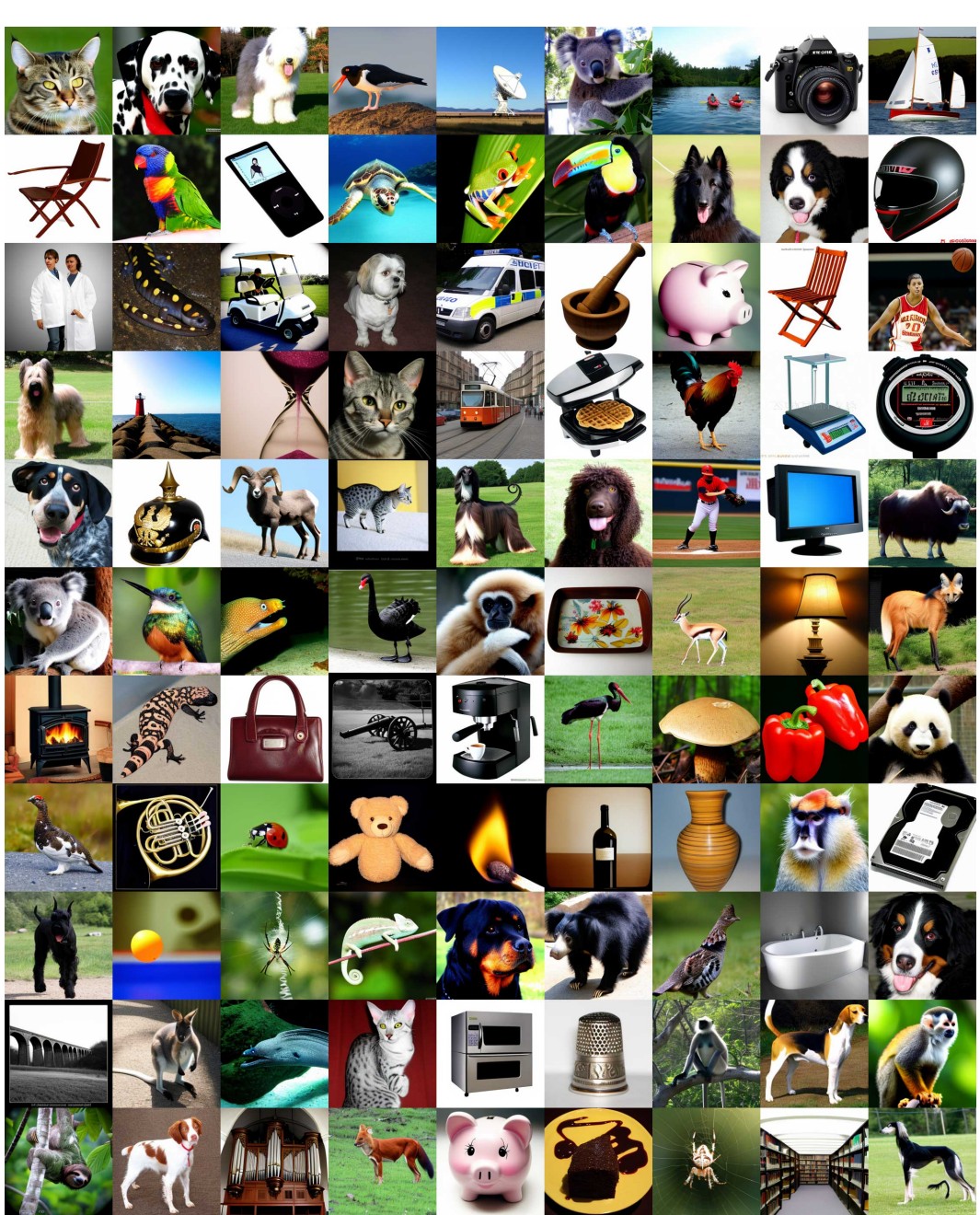

Figure 38: **Uncurated ImageNet-L-256x256 samples with random classes.** Generated using the MDiT-L model at 600k training steps, and 100 DDIM steps using $\eta = 1.0$, cfg=4.0, and negative scale conditioning to negate the variance matching blur.

## L.3 RANDOM CC3M

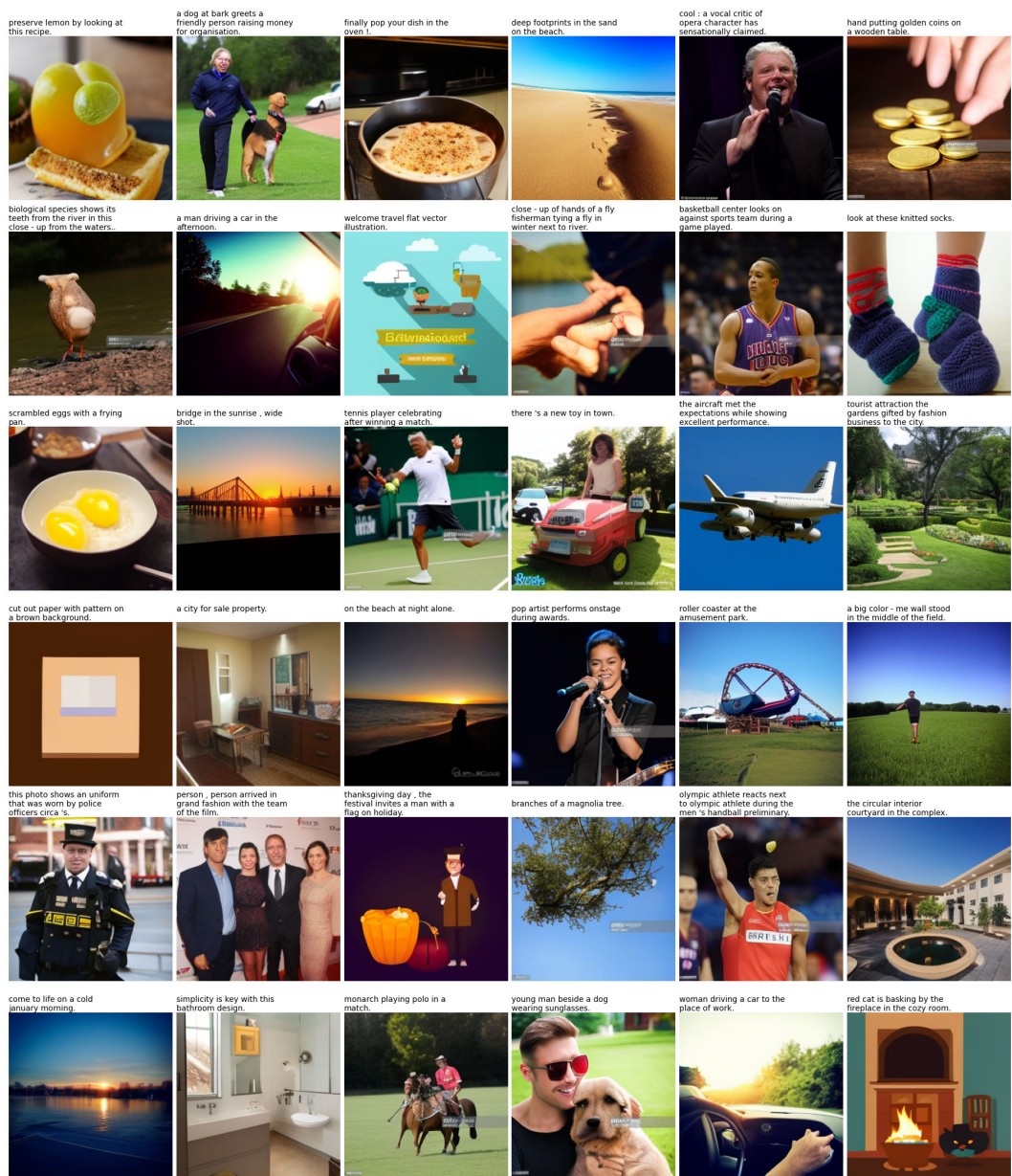

Figure 39: **Uncurated CC3M-L-256x256 samples from CC3M Validation Set.** Generated using the MDiT-L model at 200k training steps, with 50 DDIM steps using $\eta = 1.0$, cfg=4.0.

## L.4 Class Specific ImageNet for MDiT-B

This appendix presents unaltered class-specific images for MDiT-B, with minimal reordering to highlight diverse examples in larger displays. Two versions of MDiT-B are shown at 400k training steps: without variance matching (left) and with variance matching (right).

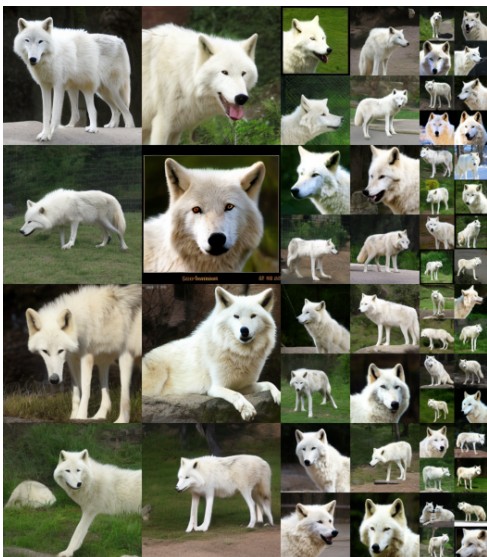

Figure 40: **Uncurated MDiT-B samples.**
100 DDIM steps using $\eta = 1.0$, cfg=4.0
Without Variance Matching.
Class label = "arctic wolf" (270)

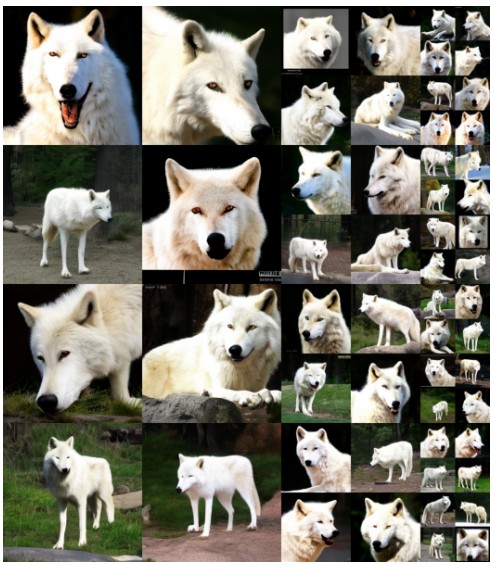

Figure 41: **Uncurated MDiT-B samples.**
100 DDIM steps using $\eta = 1.0$, cfg=4.0
With Variance Matching.
Class label = "arctic wolf" (270)

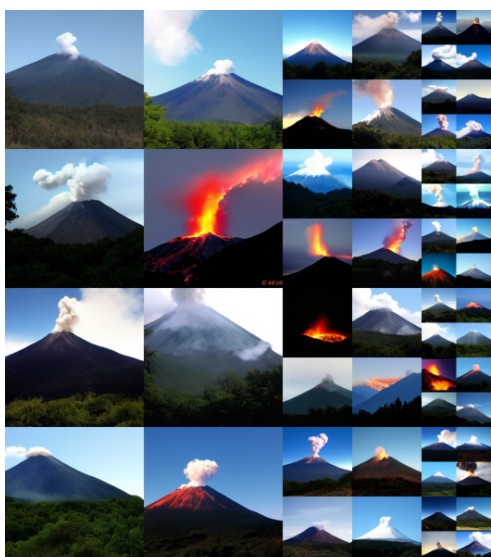

Figure 42: **Uncurated MDiT-B samples.**
100 DDIM steps using $\eta = 1.0$, cfg=4.0
Without Variance Matching.
Class label = "volcano" (980)

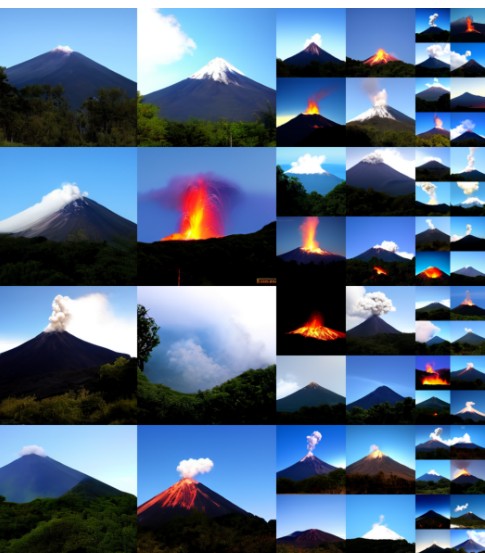

Figure 43: **Uncurated MDiT-B samples.**
100 DDIM steps using $\eta = 1.0$, cfg=4.0
With Variance Matching.
Class label = "volcano" (980)

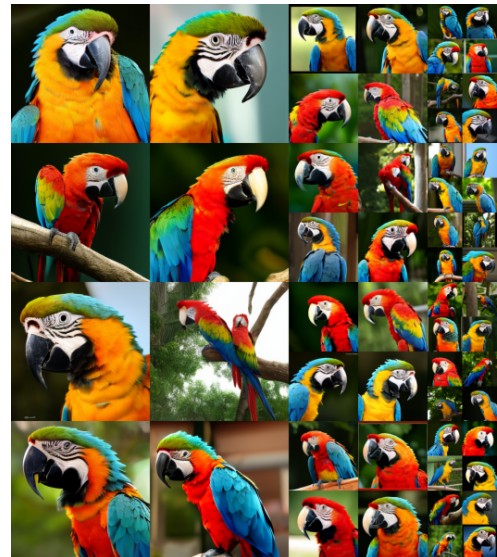

Figure 44: **Uncurated MDiT-B samples.**
100 DDIM steps using $\eta = 1.0$, cfg=4.0
Without Variance Matching.
Class label = "macaw" (88)

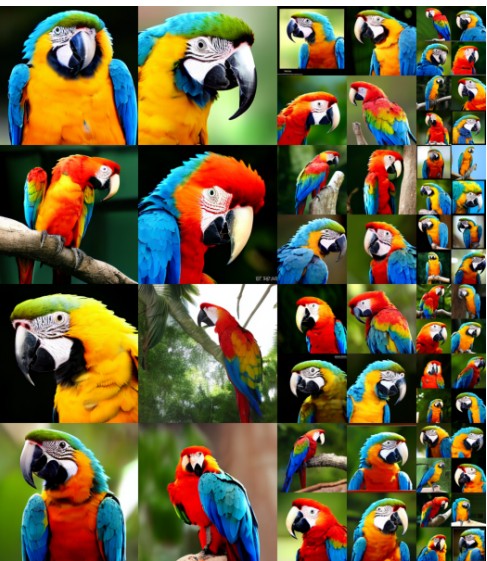

Figure 45: **Uncurated MDiT-B samples.**
100 DDIM steps using $\eta = 1.0$, cfg=4.0
With Variance Matching.
Class label = "macaw" (88)

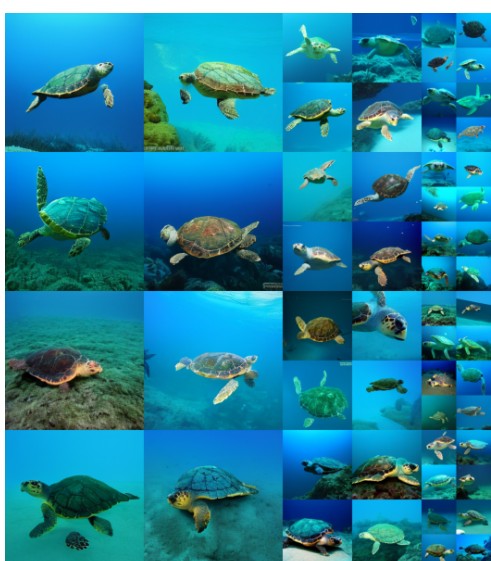

Figure 46: **Uncurated MDiT-B samples.**
100 DDIM steps using $\eta = 1.0$, cfg=4.0
Without Variance Matching.
Class label = "loggerhead sea turtle" (33)

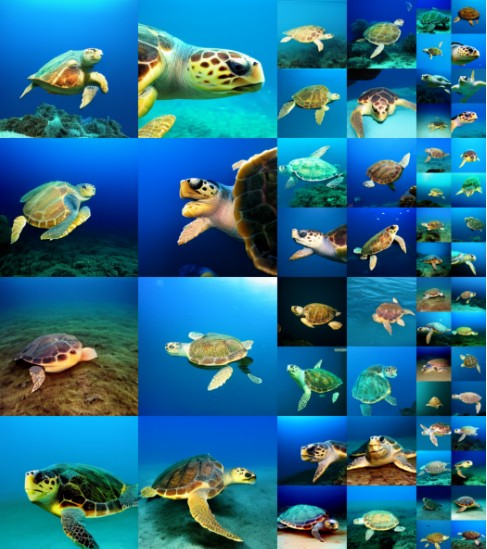

Figure 47: **Uncurated MDiT-B samples.**
100 DDIM steps using $\eta = 1.0$, cfg=4.0
With Variance Matching.
Class label = "loggerhead sea turtle" (33)

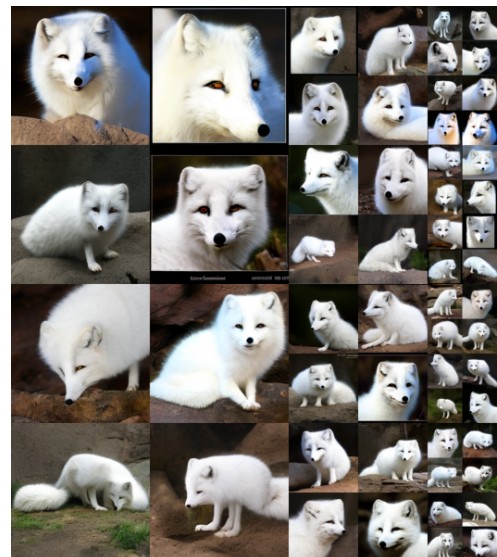

Figure 48: **Uncurated MDiT-B samples.**
100 DDIM steps using $\eta = 1.0$, cfg=4.0
Without Variance Matching.
Class label = "arctic fox" (279)

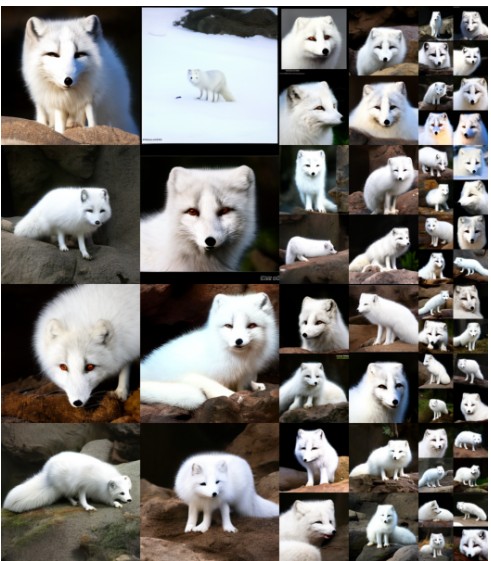

Figure 49: **Uncurated MDiT-B samples.**
100 DDIM steps using $\eta = 1.0$, cfg=4.0
With Variance Matching.
Class label = "arctic fox" (279)

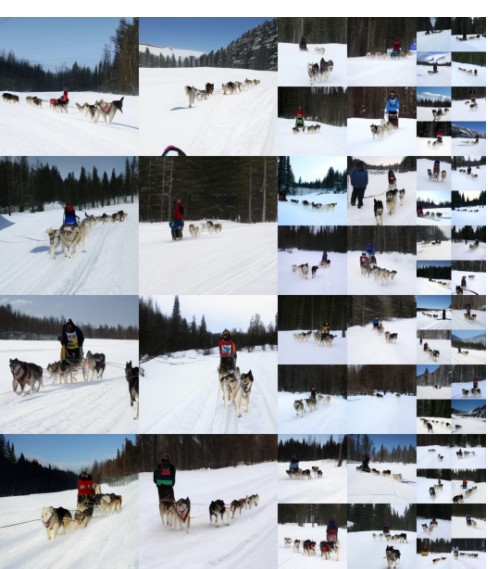

Figure 50: **Uncurated MDiT-B samples.**
100 DDIM steps using $\eta = 1.0$, cfg=4.0
Without Variance Matching.
Class label = "dog sled" (537)

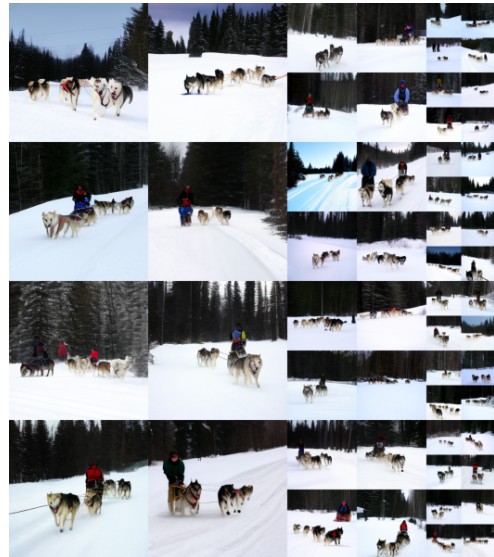

Figure 51: **Uncurated MDiT-B samples.**
100 DDIM steps using $\eta = 1.0$, cfg=4.0
With Variance Matching.
Class label = "dog sled" (537)

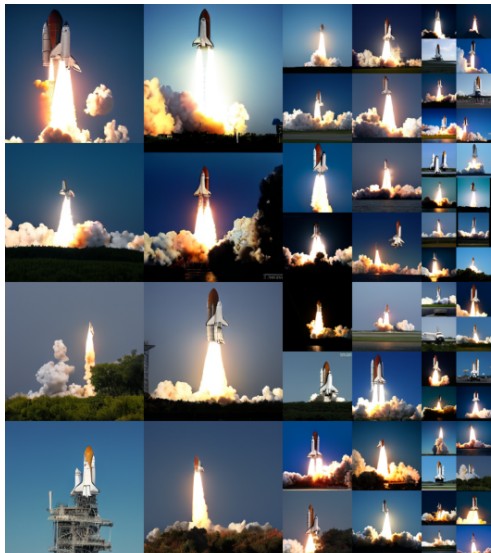

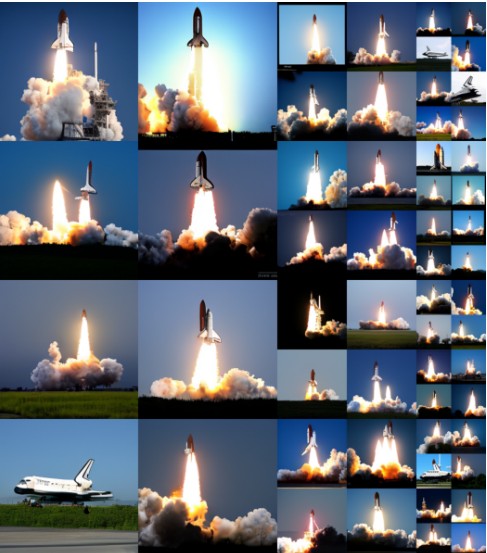

Figure 52: **Uncurated MDiT-B samples.**
100 DDIM steps using $\eta = 1.0$, cfg=4.0
Without Variance Matching.
Class label = "space shuttle" (812)

Figure 53: **Uncurated MDiT-B samples.**
100 DDIM steps using $\eta = 1.0$, cfg=4.0
With Variance Matching.
Class label = "space shuttle" (812)

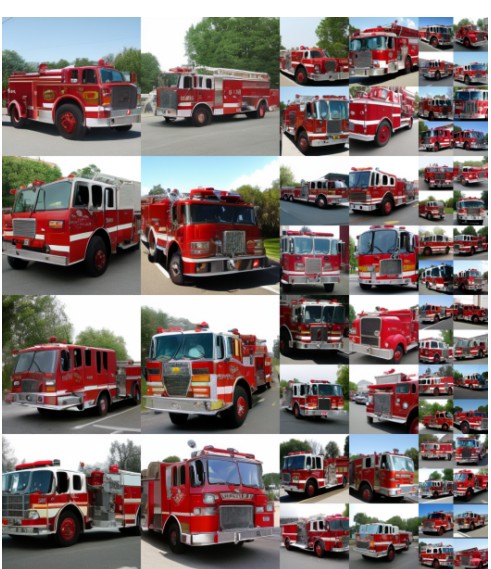

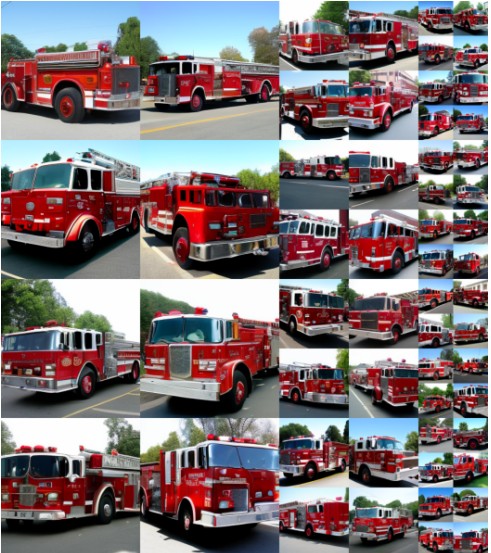

Figure 54: **Uncurated MDiT-B samples.**
100 DDIM steps using $\eta = 1.0$, cfg=4.0
Without Variance Matching.
Class label = "fire engine" (555)

Figure 55: **Uncurated MDiT-B samples.**
100 DDIM steps using $\eta = 1.0$, cfg=4.0
With Variance Matching.
Class label = "fire engine" (555)

## L.5 Class Specific ImageNet for MDiT-L and MDiT-XL

This appendix presents unaltered class-specific images for MDiT-L and MDiT-XL, with minimal reordering to highlight diverse examples in larger displays. MDiT-L is shown after 600k training steps (FID=3.32) and again after an additional 200k steps without variance matching (FID=2.88). Both versions of MDiT-XL (eps and rf), are shown at 1M training steps. Notably, MDiT-XL-rf employs a CFG of 3.0, chosen to avoid over-saturation often exacerbated by rectified flows, a concern similarly addressed in DDPM through our use of 3-channel guidance, as discussed in Peebles & Xie (2022).

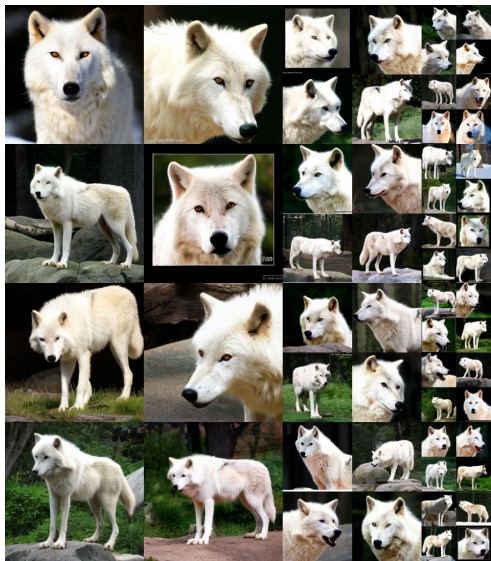

Figure 56: **Uncurated MDiT-L samples.**
100 DDIM steps using $\eta = 1.0$, cfg=4.0
Trained with variance matching.
Class label = "arctic wolf" (270)

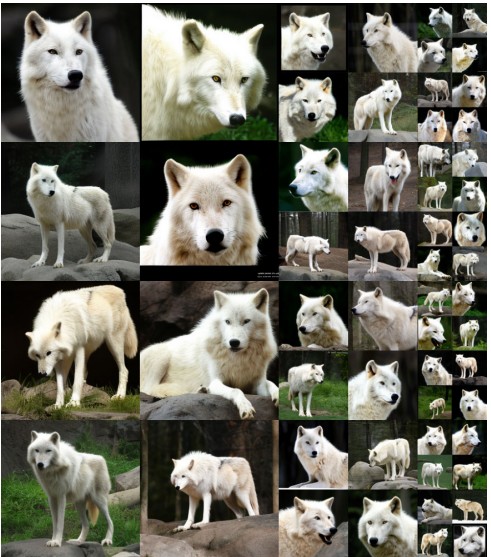

Figure 57: **Uncurated MDiT-L samples.**
100 DDIM steps using $\eta = 1.0$, cfg=4.0
Resumed without variance matching.
Class label = "arctic wolf" (270)

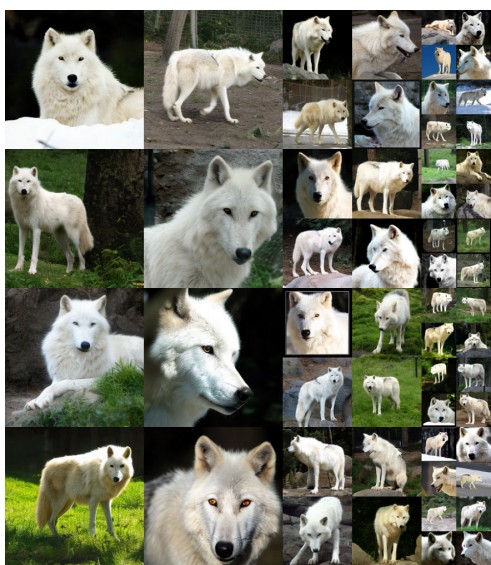

Figure 58: **Uncurated MDiT-XL samples.**
150 DDPM steps using cfg=4.0
Trained with $\epsilon$ prediction.
Class label = "arctic wolf" (270)

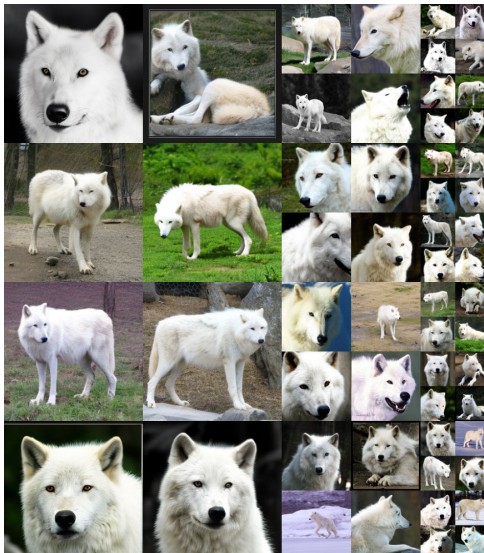

Figure 59: **Uncurated MDiT-XL samples.**
100 Euler steps using $\sigma_s = 0.1$, cfg=3.0
Trained with rectified flows.
Class label = "arctic wolf" (270)

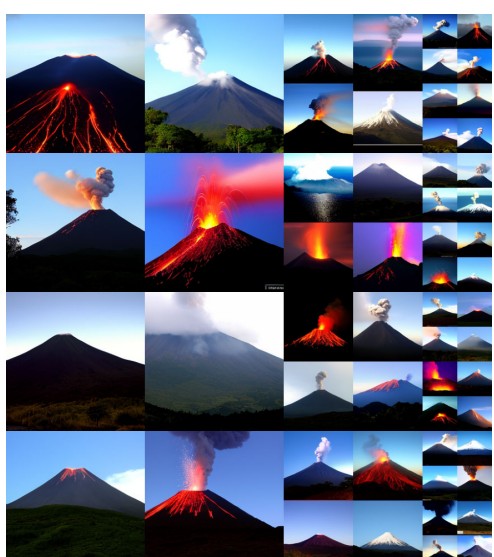

Figure 60: **Uncurated MDiT-L samples.**
100 DDIM steps using $\eta = 1.0$, cfg=4.0
Trained with variance matching.
Class label = "volcano" (980)

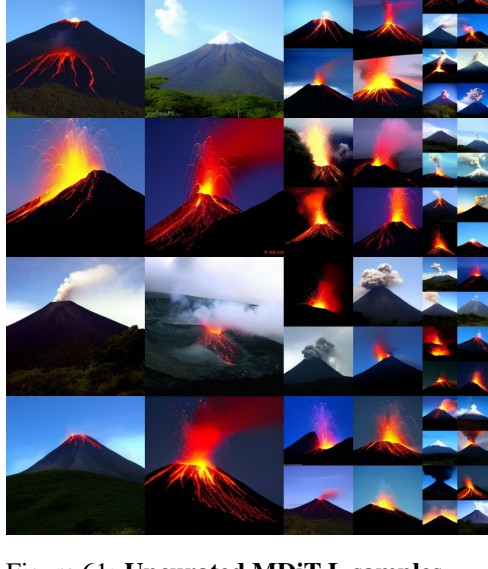

Figure 61: **Uncurated MDiT-L samples.**
100 DDIM steps using $\eta = 1.0$, cfg=4.0
Resumed without variance matching.
Class label = "volcano" (980)

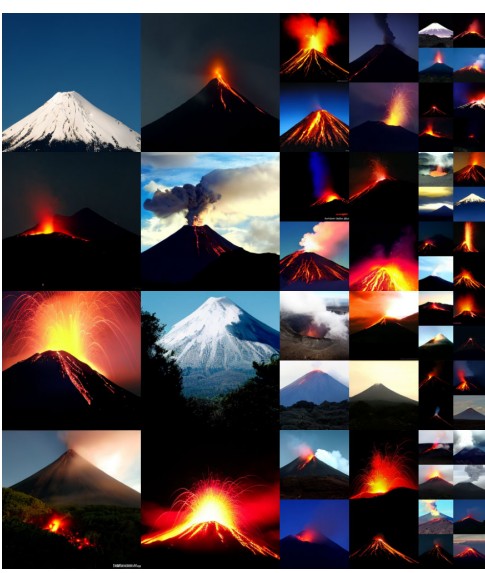

Figure 62: **Uncurated MDiT-XL samples.**
150 DDPM steps using cfg=4.0
Trained with $\epsilon$ prediction.
Class label = "volcano" (980)

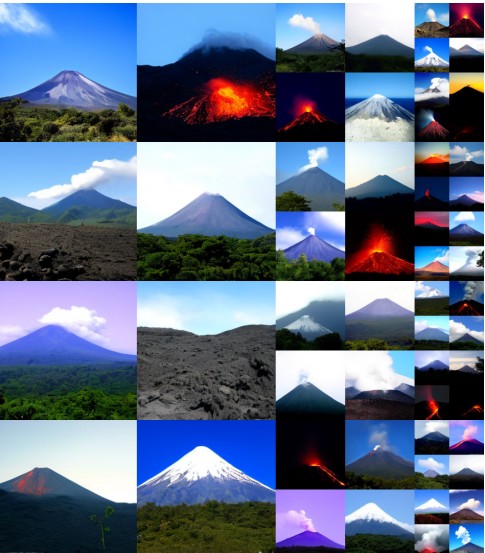

Figure 63: **Uncurated MDiT-XL samples.**
100 Euler steps using $\sigma_s = 0.1$, cfg=3.0
Trained with rectified flows.
Class label = "volcano" (980)

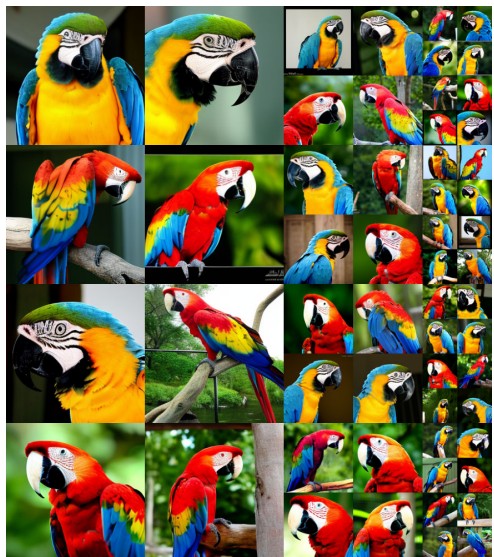

Figure 64: **Uncurated MDiT-L samples.**
100 DDIM steps using $\eta = 1.0$, cfg=4.0
Trained with variance matching.
Class label = "macaw" (88)

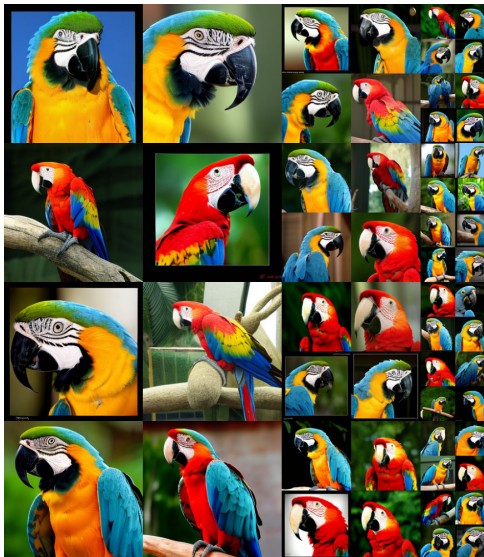

Figure 65: **Uncurated MDiT-L samples.**
100 DDIM steps using $\eta = 1.0$, cfg=4.0
Resumed without variance matching.
Class label = "macaw" (88)

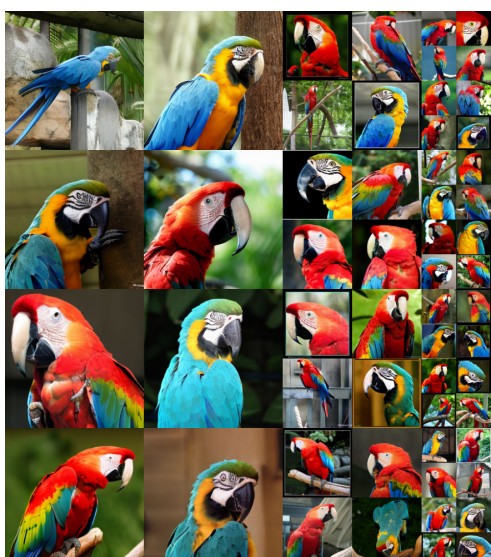

Figure 66: **Uncurated MDiT-XL samples.**
150 DDPM steps using cfg=4.0
Trained with $\epsilon$ prediction.
Class label = "macaw" (88)

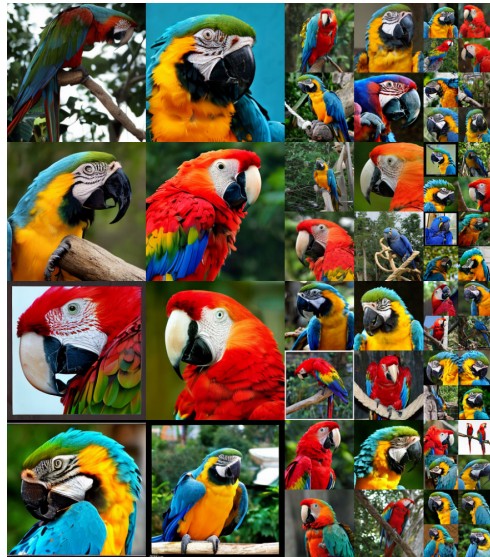

Figure 67: **Uncurated MDiT-XL samples.**
100 Euler steps using $\sigma_s = 0.1$, cfg=3.0
Trained with rectified flows.
Class label = "macaw" (88)

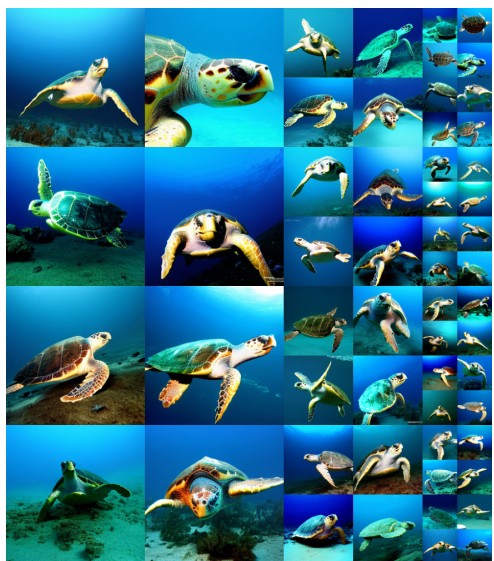

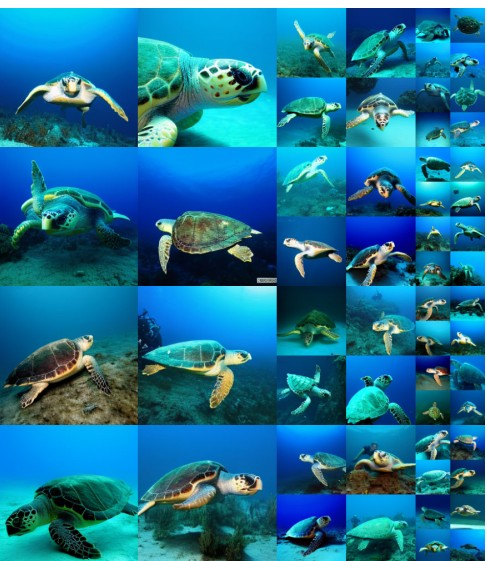

Figure 68: **Uncurated MDiT-L samples.** 100 DDIM steps using $\eta = 1.0$, cfg=4.0 Trained with variance matching. Class label = "loggerhead sea turtle" (33)

Figure 69: **Uncurated MDiT-L samples.** 100 DDIM steps using $\eta = 1.0$, cfg=4.0 Resumed without variance matching. Class label = "loggerhead sea turtle" (33)

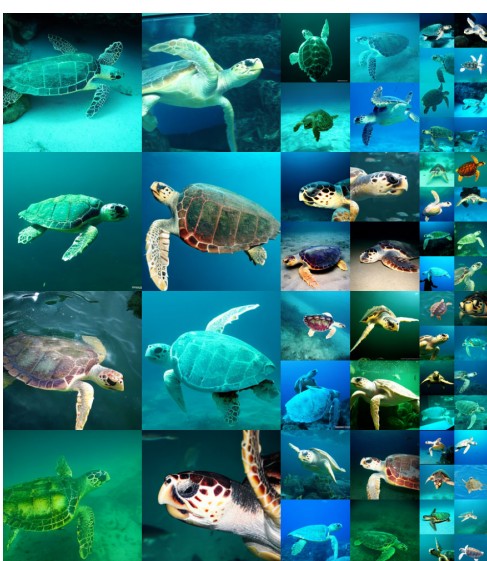

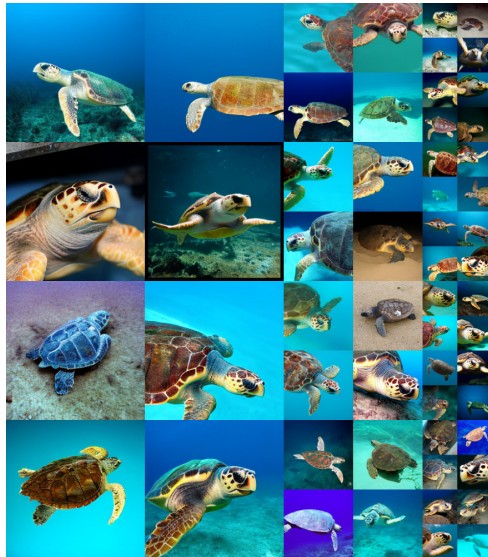

Figure 70: **Uncurated MDiT-XL samples.** 150 DDPM steps using cfg=4.0 Trained with $\epsilon$ prediction. Class label = "loggerhead sea turtle" (33)

Figure 71: **Uncurated MDiT-XL samples.** 100 Euler steps using $\sigma_s = 0.1$, cfg=3.0 Trained with rectified flows. Class label = "loggerhead sea turtle" (33)

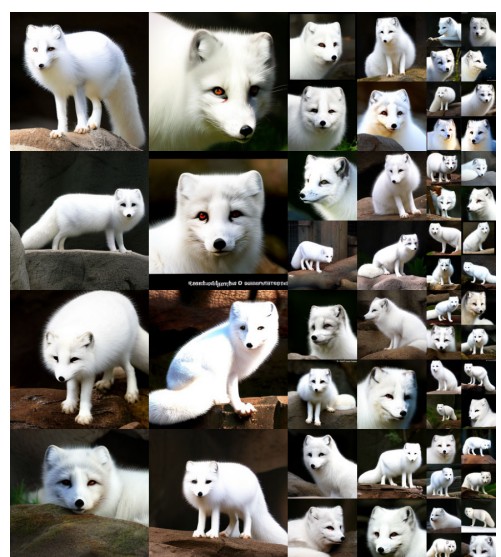 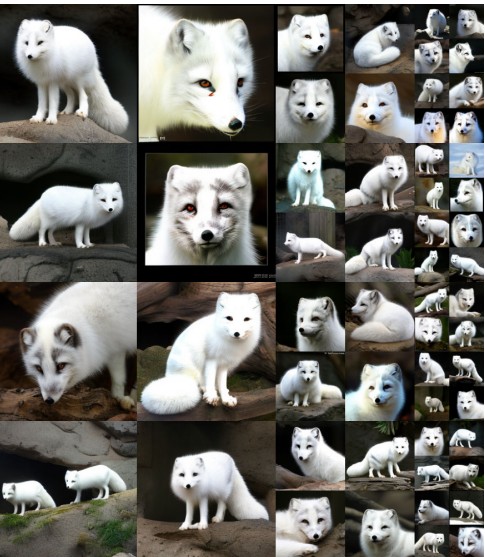

Figure 72: **Uncurated MDiT-L samples.**
100 DDIM steps using $\eta = 1.0$, cfg=4.0
Trained with variance matching.
Class label = "arctic fox" (279)

Figure 73: **Uncurated MDiT-L samples.**
100 DDIM steps using $\eta = 1.0$, cfg=4.0
Resumed without variance matching.
Class label = "arctic fox" (279)

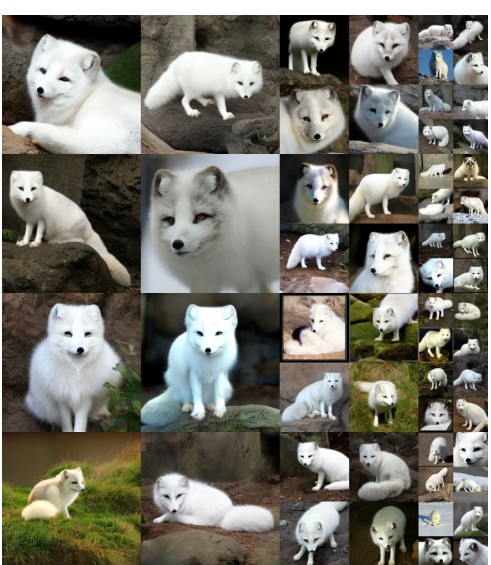 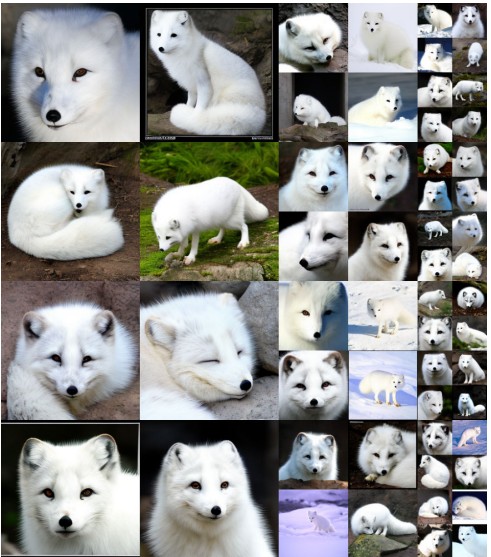

Figure 74: **Uncurated MDiT-XL samples.**
150 DDPM steps using cfg=4.0
Trained with $\epsilon$ prediction.
Class label = "arctic fox" (279)

Figure 75: **Uncurated MDiT-XL samples.**
100 Euler steps using $\sigma_s = 0.1$, cfg=3.0
Trained with rectified flows.
Class label = "arctic fox" (279)

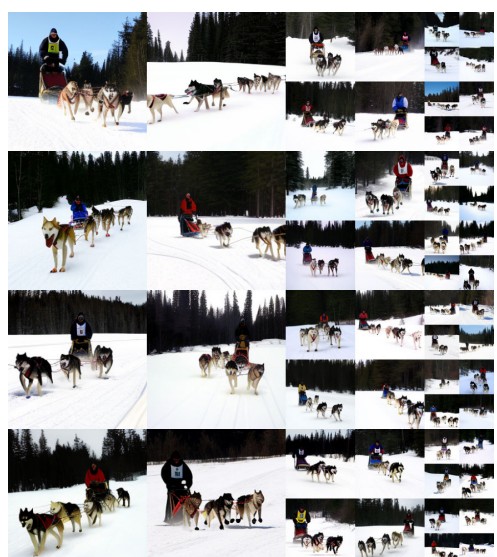

Figure 76: **Uncurated MDiT-L samples.**
100 DDIM steps using $\eta = 1.0$, cfg=4.0
Trained with variance matching.
Class label = "dog sled" (537)

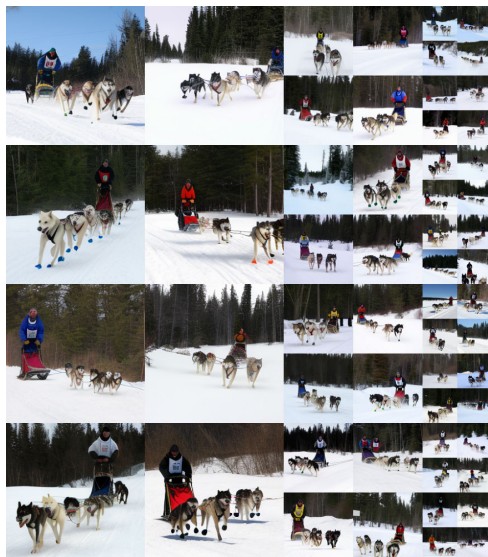

Figure 77: **Uncurated MDiT-L samples.**
100 DDIM steps using $\eta = 1.0$, cfg=4.0
Resumed without variance matching.
Class label = "dog sled" (537)

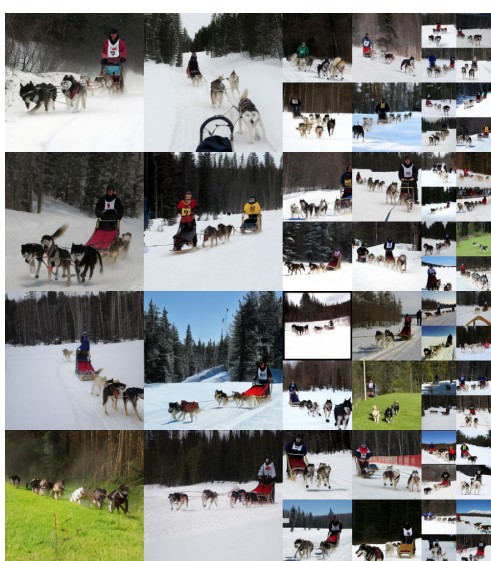

Figure 78: **Uncurated MDiT-XL samples.**
150 DDPM steps using cfg=4.0
Trained with $\epsilon$ prediction.
Class label = "dog sled" (537)

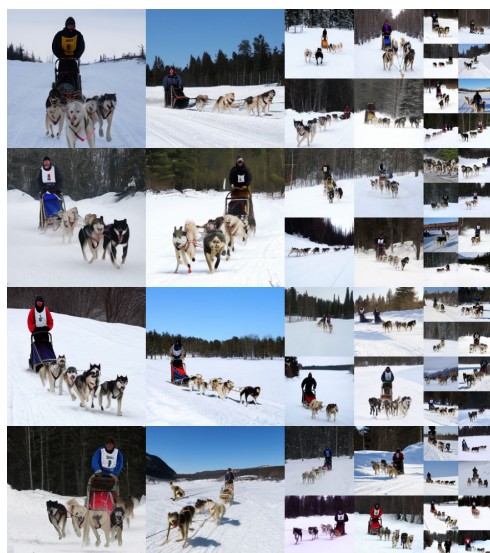

Figure 79: **Uncurated MDiT-XL samples.**
100 Euler steps using $\sigma_s = 0.1$, cfg=3.0
Trained with rectified flows.
Class label = "dog sled" (537)

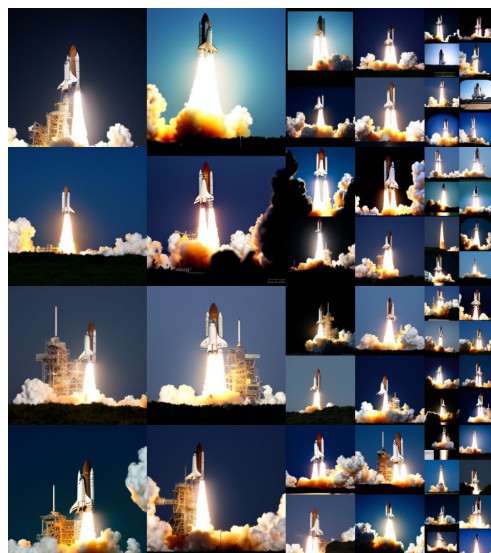

Figure 80: **Uncurated MDiT-L samples.**
100 DDIM steps using $\eta = 1.0$, cfg=4.0
Trained with variance matching.
Class label = "space shuttle" (812)

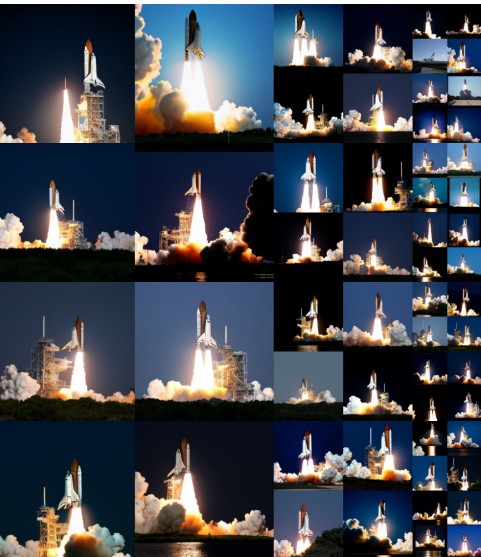

Figure 81: **Uncurated MDiT-L samples.**
100 DDIM steps using $\eta = 1.0$, cfg=4.0
Resumed without variance matching.
Class label = "space shuttle" (812)

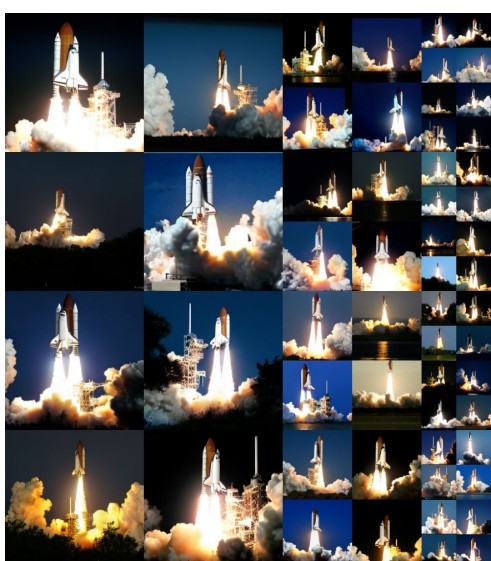

Figure 82: **Uncurated MDiT-XL samples.**
150 DDPM steps using cfg=4.0
Trained with $\epsilon$ prediction.
Class label = "space shuttle" (812)

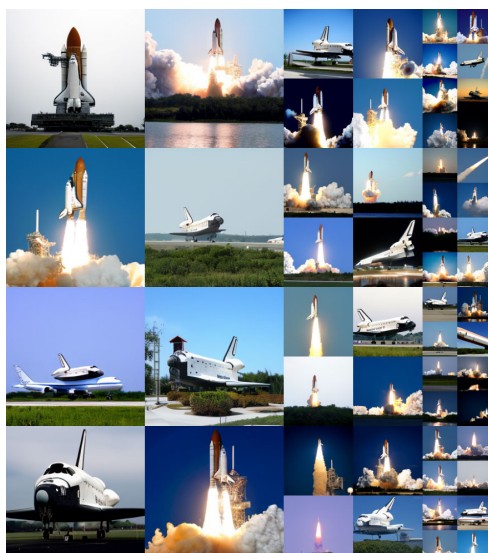

Figure 83: **Uncurated MDiT-XL samples.**
100 Euler steps using $\sigma_s = 0.1$, cfg=3.0
Trained with rectified flows.
Class label = "space shuttle" (812)

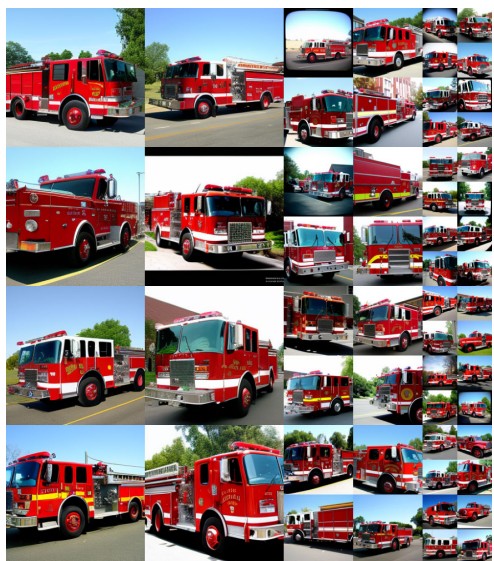

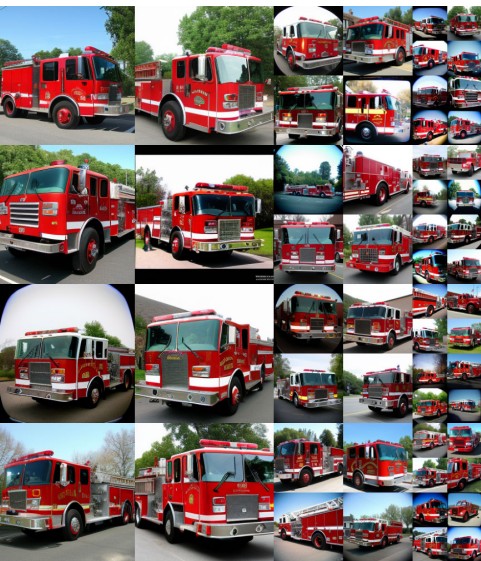

Figure 84: **Uncurated MDiT-L samples.**
100 DDIM steps using $\eta = 1.0$, cfg=4.0
Trained with variance matching.
Class label = "fire engine" (555)

Figure 85: **Uncurated MDiT-L samples.**
100 DDIM steps using $\eta = 1.0$, cfg=4.0
Resumed without variance matching.
Class label = "fire engine" (555)

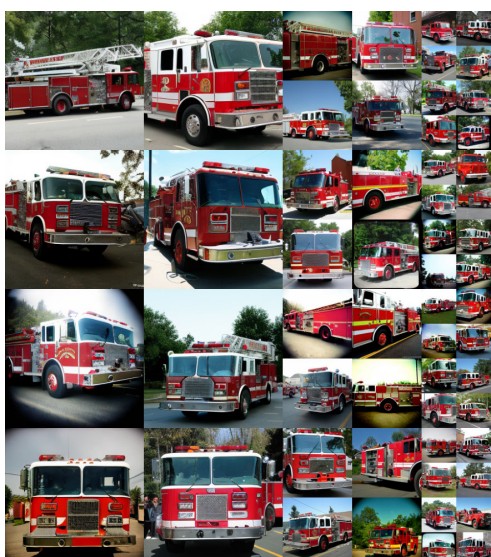

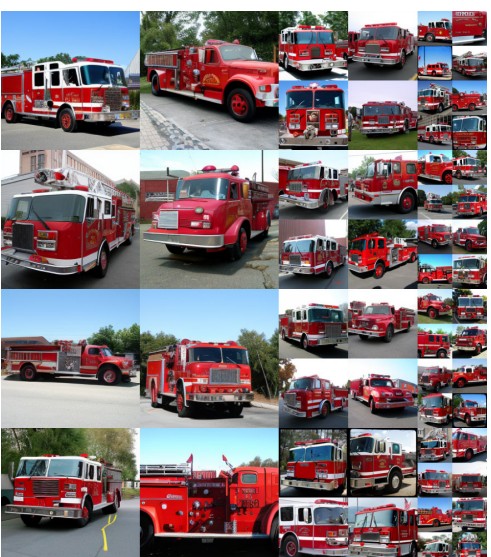

Figure 86: **Uncurated MDiT-XL samples.**
150 DDPM steps using cfg=4.0
Trained with $\epsilon$ prediction.
Class label = "fire engine" (555)

Figure 87: **Uncurated MDiT-XL samples.**
100 Euler steps using $\sigma_s = 0.1$, cfg=3.0
Trained with rectified flows.
Class label = "fire engine" (555)

## M VALIDATING EXPLAINABILITY THROUGH DESTRUCTIVE TESTING

This section explores the predictive power of the explainability analysis proposed in Section 4 by validating the expected contributions of each block and its respective position embeddings to overall image composition. To achieve this, we perform a series of qualitative experiments where portions of the model are selectively disabled (replaced with Identity transformations), effectively disrupting their functionality. We term this process *destructive testing* as it often results in degraded image composition, consistent with our hypothesis about the critical roles of certain components. For simplicity and clarity, we primarily focus on the MDiT-B model trained on ImageNet with configuration {2,4,0,9}, as previously shown in Figure 5c. An enlarged version of this figure is presented in Figure 88, providing a detailed view of the complex magnitudes of the Q and K vectors within each self-attention head.

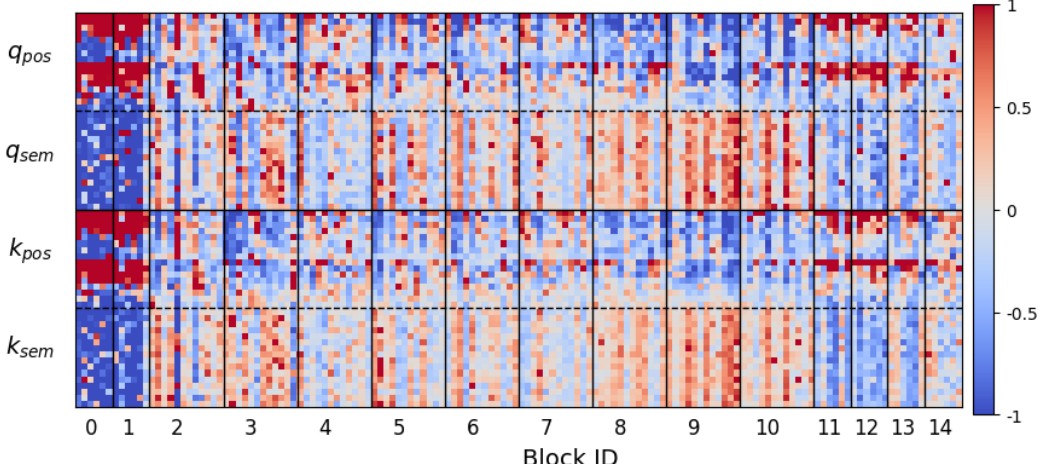

Figure 88: Complex magnitude ($|| \cdot ||^2 - 2$) of Q and K vectors of the MHSA heads for the MDiT-B model trained on ImageNet with configuration {2,4,0,0}. Red and Blue indicates strong and weak activation, respectively. Per vector channels are (from top to bottom): x-position, y-position, and semantic features. Highest semantic focus occurs in blocks 8 and 9.

In the following subsections, we employ three main destructive testing techniques: (1) disabling the RoPE position embeddings in each block, (2) replacing individual blocks with identity transformations to disable them entirely, and (3) adjusting the neighborhood attention kernel size in the outer blocks, reducing it from $k = 7$ to $k = 5$ and $k = 3$.

### M.1 DISABLING ROPE

Disabling the RoPE embeddings for each self-attention layer provides a mechanism to evaluate the impact of positional focus in individual attention heads. This is achieved by bypassing the rotation operation for each block, effectively setting the rotation angle for each channel to $\theta_i = 0$. From the probing visualization in Figure 88, we can outline the following expectations:

- **Blocks B0 and B1:** Both blocks exhibit strong positional focus, but their impact differs due to their location in the transformer stack. B0, being earlier, likely processes fine, localized features and is expected to have minimal visible disruption. In contrast, B1, which processes slightly more complex but still local features, may show subtle disruptions in feature extraction without significantly affecting overall image composition.
- **Blocks B11–B14:** These blocks, identified as hybrid and locally focused, are expected to exhibit minor local consistency changes. However, based on the patching behavior described in Appendix E, disruptions in local features may propagate due to regularity in the core output, necessitating "smoothing" by subsequent layers.
- **Block B4:** As it exhibits the lowest semantic focus, this block is predicted to show the most significant structural changes.

- **Blocks B8–B10:** These blocks, with a strong semantic focus, are expected to display minimal structural changes.

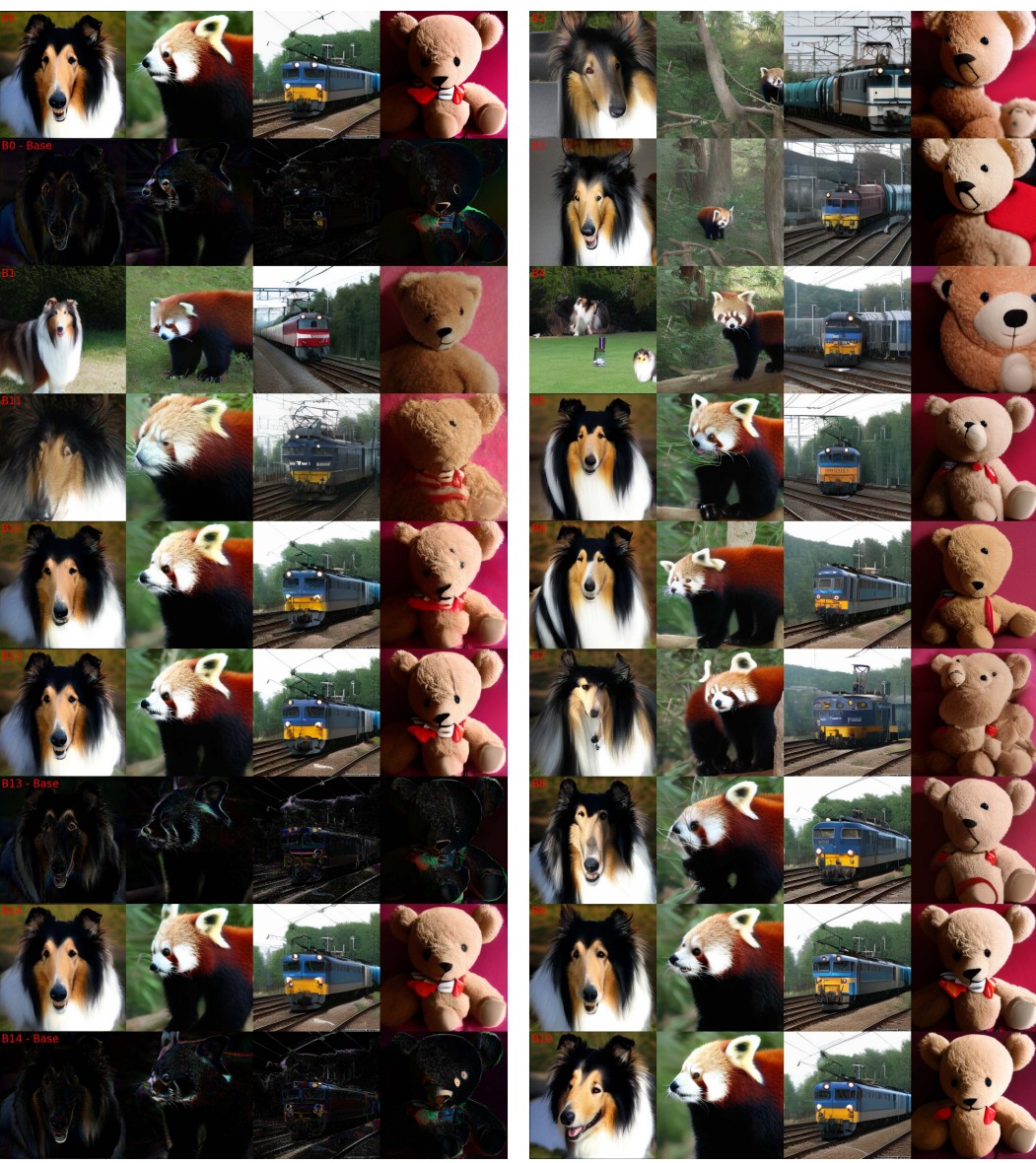

(a) Outer Blocks of {2,4,0,9}.     (b) Inner Blocks of {2,4,0,9}.

Figure 89: Visual impact of disabling RoPE embeddings for each block (independently). Block ID is shown in the upper left corner in red text, with deltas represented by "B$id$ - Base" to show small deviations with the outer blocks. Using 50 DDIM steps; cfg=4.

The results of this test are presented in Figure 89, confirming the expectations outlined above. Notably, the minimal structural changes observed in Blocks B8–B10 suggest that these blocks could potentially operate without RoPE embeddings altogether. Such an omission could enable the model to focus more strongly on semantic processing in these layers while reducing the computational overhead associated with RoPE. Furthermore, the pronounced local feature focus in Blocks B0 and B12–B14 indicates that these layers might benefit from a reduced neighborhood attention window size, providing an additional avenue for computational optimization.

## M.2 DISABLING BLOCKS

Disabling entire blocks provides a measure of the contribution each block makes to the overall image generation process. This is achieved by individually replacing each block with an identity transform, such that $h = x$, effectively skipping the attention and feed-forward layers entirely. Using the probing visualization from Figure 88, we outline the following expectations:

- **Blocks B0 and B1:** These blocks are responsible for low-level feature extraction from the input. However, their long-range positional focus suggests an additional role in denoising the input, as substantiated by the results from the previous section. Consequently, removing either block is likely to result in total image collapse.

- **Blocks B11-B14:** These blocks refine low-level features due to their predominantly hybrid focus. While removing them is expected to preserve overall image coherence, it may introduce errors in progressively higher spatial frequencies (e.g., finer details). Notably, if Block B11 is responsible for smoothing patch artifacts from the core, its removal could lead to near-complete image collapse.

- **Blocks B8 and B9:** With their high semantic focus, these blocks are expected to induce the greatest semantic disruption when removed. Block B8 may also exhibit slightly more spatial disruption due to its marginally lower semantic and increased positional focus.

The results of this test confirm these expectations. Figure 90 illustrates the observed disruptions, highlighting the specific roles of each block in maintaining image coherence and refinement. These findings suggest that increasing the capacity of Blocks B8 and B9, given their critical role in semantic processing, may yield greater improvements in generation quality compared to augmenting other blocks.

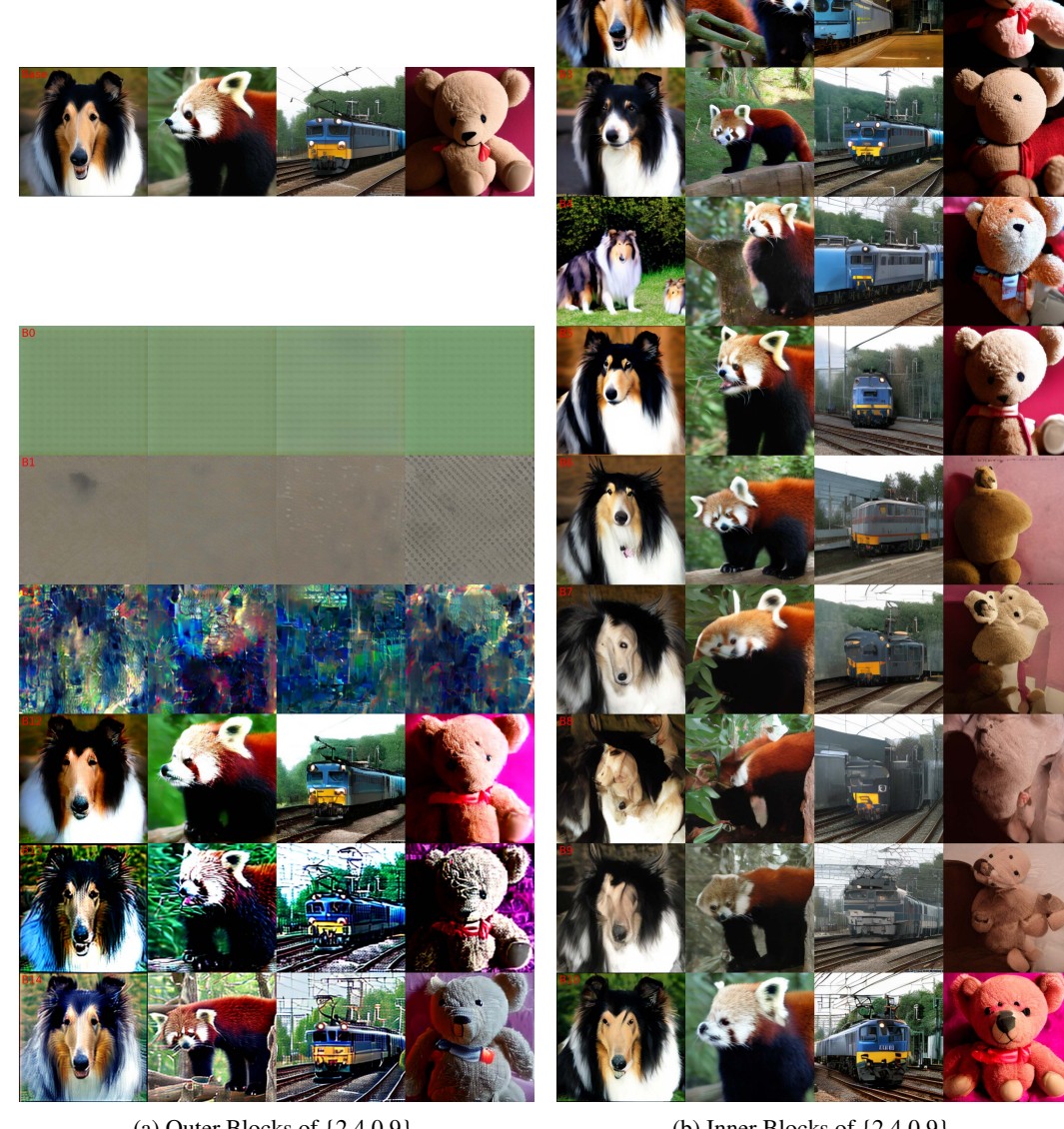

(a) Outer Blocks of {2,4,0,9}.    (b) Inner Blocks of {2,4,0,9}.

Figure 90: Visual impact of disabling each block (independently), replacing it with the identity transform $h = x$. Block ID is shown in the upper left corner in red text. Using 50 DDIM steps; cfg=4.

## M.3    ADJUSTING NEIGHBORHOOD KERNEL SIZE

All of the outer blocks in MDiT utilize neighborhood attention (Natten) to mitigate the $\mathcal{O}(N^2)$ complexity associated with increased sequence length. Natten also provides an efficient way to test the impact of spatial feature scale on each block by adjusting the neighborhood kernel size. Specifically, we focus on reducing the kernel size, limiting the scale of features each block can attend to. The baseline case uses $k = 7$, and we evaluate $k = 5$ and $k = 3$. Using the probing visualization from Figure 88, we outline the following expectations:

- **Blocks B0 and B1**: These blocks are expected to show the strongest deviations due to their high positional focus across the entire kernel size (notably, the first six channels for $k = 7$ and $f = 16$). Block B0, however, may exhibit slightly lower deviation for $k = 5$ compared to B1, given the reduced magnitude in channels 5 and 6.

- **Blocks B11-B13:** Due to their hybrid focus and lower magnitudes in the higher channels, these blocks should show less deviation than B0 and B1 for $k = 5$, with noticeable deviations emerging at $k = 3$.

- **Block B14:** With the least positional focus among the outer blocks, B14 is expected to exhibit minimal deviation, with only minor changes at $k = 3$.

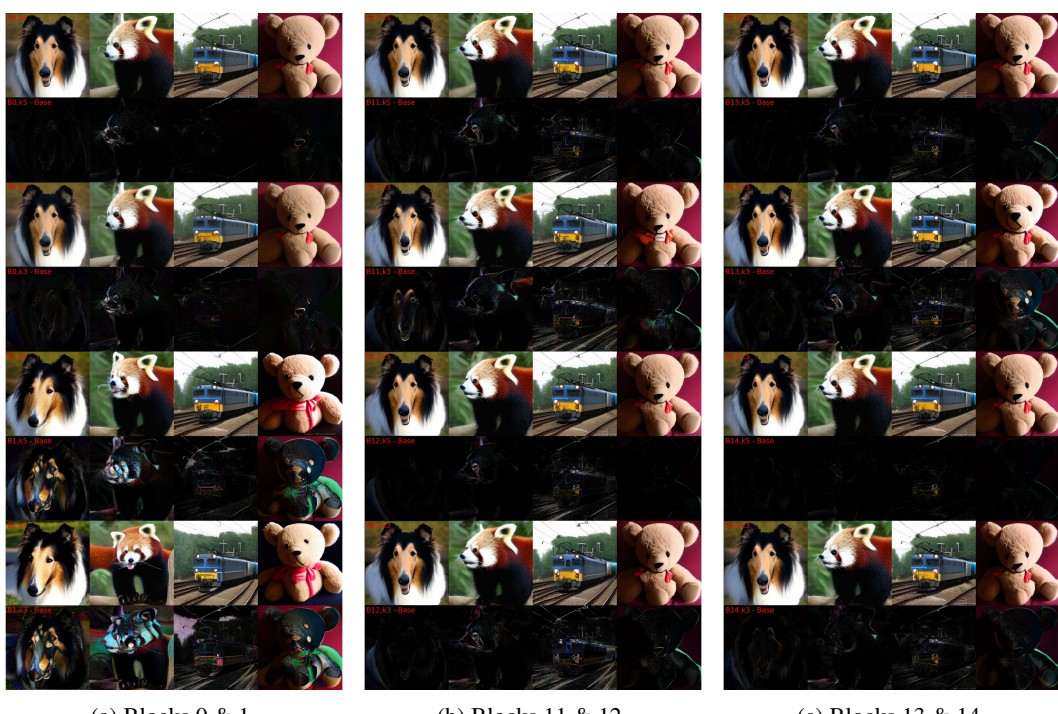

(a) Blocks 0 & 1.        (b) Blocks 11 & 12.        (c) Blocks 13 & 14.

Figure 91: Visual impact of adjusting the Natten kernel size for the outer blocks (independently). Block ID and kernel size is shown in the upper left corner in red text as "B$id$,k$size$", with $k = 5$ and $k = 3$. Also showing deviations from the baseline case with $k = 7$. Using 50 DDIM steps; cfg=4.

The results, presented in Figure 91, confirm these expectations. This analysis suggests that a smaller kernel size of $k = 5$ can be used for Blocks B11–B13, with Block B14 capable of operating with $k = 3$. This adjustment could significantly reduce the computational overhead in the outer blocks, as the neighborhood attention complexity scales with $\mathcal{O}(N \cdot k^2)$.

