# OpenReview forum: "Multi-Scale Image Diffusion Transformers: Explainability Leads to Faster Training"
_ICLR.cc/2025/Conference — Submitted to ICLR 2025_

### Official Review · Reviewer_d2qx · 2024-11-04

**Soundness:** 3
**Presentation:** 4
**Contribution:** 3
**Rating:** 8
**Confidence:** 4

**Summary:**

This paper proposes the Multi-Scale Diffusion Transformer (MDiT), a novel architecture that introduces heterogeneous, asymmetric, scale-specific transformer blocks to diffusion transformers (DiTs) for image synthesis. MDiT addresses the slow convergence issues in traditional DiTs by reintroducing inductive biases typically found in convolutional models, such as multi-scale features and translation invariance, to accelerate training. Using explainability techniques, the authors analyze MDiT’s architectural behavior, showing it functions like a semantic autoencoder, learning image structures more effectively and speeding up training by up to 7x compared to state-of-the-art models. Additionally, a variance matching regularization method is introduced to correct discrepancies in sample variance, enhancing image quality and further accelerating convergence.

**Strengths:**

- This paper is well-written and easy to follow.
- The proposed architecture is novel. By integrating heterogeneous, multi-scale transformer blocks, MDiT adds flexibility and specificity to DiTs.
- The variance matching regularization reduces discrepancies in variance for latent diffusion models, leading to enhanced image contrast and vibrancy.
- The efficiency of the proposed model is remarkable.

**Weaknesses:**

Overall the paper is great! A few minor weaknesses include:
- The MDiT is a little complex, with multiple configurations. This makes it challenging to find the optimal config.
- MDiT particularly focuses on DiT architecture. Are some of the techniques also applicable to UNet-based architecture?
- While MDiT shows promising results on FFHQ and ImageNet, additional tests on diverse and large-scale benchmarks are encouraged.

**Questions:**

This paper includes a comprehensive evaluation of different prospects. No further questions were raised from me.

---

> ### Author Response · Authors · 2024-11-24
> **Response to Reviewer d2qx**
>
> Thank you for your positive review and for recognizing the contributions of our work. We are grateful for your encouraging comments on the clarity of our presentation and the novelty of our proposed architecture. Below, we address the specific points raised in your feedback to provide further context and clarification.
>
> ---
>
> ## W1. Complexity of MDiT and Configuration Challenges
>
> > The MDiT is a little complex, with multiple configurations. This makes it challenging to find the optimal config.
>
> We agree that MDiT introduces a degree of complexity due to its flexibility and configurability. That said, we believe MDiT is no more complex than typical U-Net-based models (e.g., LDM). Specifically, the scaling of hidden dimensions (including the aggregate block attention heads) and the values of {M, N, K, L} mirrors the adjustments typically made to residual blocks and U-Net level scaling in architectures like LDM.
>
> To simplify parameter selection, we have followed specific hyperparameter rules similar to the best practices used in U-Nets. These guidelines were followed to scale MDiT configurations between B-scale, L-scale, and XL-scale, with only minor adjustments made to balance parameter count and FLOPs for fair comparisons with baselines like DiT.
>
> For easier accessibility, we have detailed these guidelines in **Appendix G.6**.
>
> ---
>
> ## W2. Applicability to U-Net-Based Architectures
>
> > MDiT particularly focuses on DiT architecture. Are some of the techniques also applicable to UNet-based architecture?
>
> We appreciate this thoughtful question and agree that certain techniques from MDiT could be applicable to Conv-U-Net architectures. Specifically:
>
> - **Analogous Structure to LDM:** Applying the architectural principles of MDiT to Conv-U-Nets would result in a structure resembling LDM, with interleaved ResNet and transformer blocks.
> - **Adaptation of Aggregate Blocks:** The aggregate blocks in MDiT could be adapted to replace the middle U-Net level in Conv-U-Nets. This would require the use of non-overlapping kernel downsampling (as in MDiT) to avoid excess overhead. However, the subsequent ResNet kernels are likely to smooth out any spatial artifacts caused by this optimization.
>
> While our primary focus was on adapting the strengths of U-Nets to DiT models, this exploration also opens an opportunity to apply the insights back to U-Nets. This could be especially beneficial in application domains where transformers are less practical, such as ultra-high-resolution tasks where sequence length becomes a limiting factor.
>
> ---
>
> ## W3. Evaluation on Larger Benchmarks
>
> > While MDiT shows promising results on FFHQ and ImageNet, additional tests on diverse and large-scale benchmarks are encouraged.
>
> We agree that evaluating on more diverse and large-scale benchmarks would provide further validation of MDiT’s generalizability. As detailed in **Appendix C**, we performed experiments with MDiT-L on CC3M (text-to-image generation). While our limited training budget allowed only ~20 epochs, the results were promising given these constraints. To better emphasize this experiment, we have clarified its reference in **Section 3** of the main text.
>
> These findings suggest that MDiT is scalable and applicable to diverse domains, though we acknowledge that future work should extend evaluations to additional large-scale and diverse benchmarks.
>
> ---
>
> Thank you again for your thoughtful and constructive feedback. We hope these clarifications address your concerns and further highlight the contributions and strengths of our work.
>
>
> [LDM] Rombach et al. (2021). High-Resolution Image Synthesis with Latent Diffusion Models. CVPR’22.

---

> > ### Comment · Reviewer_d2qx · 2024-12-02
> > **Follow up**
> >
> > Thanks to the authors for providing the response. My concerns have been addressed and I am still learning towards acceptance for this paper.

---

### Official Review · Reviewer_GJVN · 2024-11-05

**Soundness:** 3
**Presentation:** 3
**Contribution:** 3
**Rating:** 6
**Confidence:** 4

**Summary:**

The paper proposes a novel method called the Multi-Scale Diffusion Transformer (MDiT), which aims to address the challenges of high computational demands and slow convergence rates in diffusion models for image synthesis. MDiT incorporates a heterogeneous, asymmetric, scale-specific transformer block design that leverages inductive biases to improve training efficiency. The paper demonstrates that by reintroducing structural biases, the MDiT can achieve a significant increase in convergence speed on FFHQ-256x256 and ImageNet-256x256, with a notable reduction in computational requirements. Additionally, the paper introduces a variance matching regularization technique to enhance image quality and further accelerate convergence.

I have read the response of the authors and the comments of other reviewers. I would keep my original score.

**Strengths:**

1. MDiT achieves faster training times and reduces computational costs, which leverages distinct transformer blocks for image feature processing.
2. This paper introduces explainable AI techniques to understand and optimize the process of diffusion transformers.
3. This paper also proposes a variance matching regularization module  to correct sample variance discrepancies.

**Weaknesses:**

1. What are the specific contributions of each architectural choice (e.g., partial head Rotary Positional Embeddings, Aggregate Blocks) to the overall performance?
2. This paper shows a lot of visualization generation results. Are there any failure cases?
3. How does the MDiT architecture scale with increasing model size, and can it be flexibly adapted to different image resolutions and datasets?

**Questions:**

For details, please see weaknesses.

---

> ### Author Response · Authors · 2024-11-24
> **Response to Reviewer GJVN (1/2)**
>
> Thank you for the positive review and for recognizing our contributions, particularly in terms of training efficiency and the explainability technique. Below, we address the specific weaknesses raised and provide clarifications, along with additional experimental results included in the revised manuscript.
>
> ---
>
> ## W1: Contributions of Architectural Choices
>
> > What are the specific contributions of each architectural choice (e.g., partial head Rotary Positional Embeddings, Aggregate Blocks) to the overall performance?
>
> We thank the reviewer for this question and have included an ablation summary in **Section 5.2** and expanded **Appendix F** to clarify the contributions of the individual architectural components.
>
> - **Partial Head Rotary Positional Embeddings (RoPE):** While this feature does not directly improve FID, it allows for zero-shot scaling to larger resolutions and aspect ratios, as shown in **Figure 1** and detailed in **Appendix I**. Adjusting the RoPE frequency (or equivalently reducing $r_{dim}$) results in a modest improvement of 0.7 FID points. However, this configuration was not used in our other experiments.
>
> - **Aggregate Blocks:** These offer an **~1 point** FID improvement over the addition of outer layers (as shown in **Section 5.2**, **Appendix F**, and **Figure 7**). Beyond quantitative gains, aggregate blocks improve medium-scale structural representation, as supported by spectral power analysis in Figure 3. Anecdotally, this improvement reduces errors in quadruped limb generation, such as extra or missing limbs.
>
>
> We hope these additions clarify the distinct roles of these components in improving the architecture’s overall performance.
>
> ---
>
> ## W2: Failure Cases
>
> > This paper shows a lot of visualization generation results. Are there any failure cases?
>
> We appreciate your concern regarding failure cases in the visualizations. Overall, we do not observe any new or additional failure cases beyond those typically seen in smaller diffusion models trained on FFHQ and ImageNet, such as occasional artifacts and feature hallucinations. These instances are visible in some of the uncurated samples provided in **Appendix L**.
>
> Importantly, our analysis does not reveal any unique failure modes introduced by the MDiT architecture compared to established models like DiT. Anecdotally, MDiT may improve coherency and reduce fine detail errors, but these observations are challenging to quantify.

---

> > ### Author Response · Authors · 2024-11-24
> > **Response to Reviewer GJVN (2/2)**
> >
> > ## W3: Scalability
> >
> > > How does the MDiT architecture scale with increasing model size…
> >
> > We appreciate the reviewer raising this important question regarding scalability. To address this, we conducted additional experiments, and the results have been included in **Appendix B.5**. Below is a summary of key findings:
> >
> >
> > - **Baseline Inference Scalability:** MDiT scales linearly with increasing resolution, comparable to DiT at resolutions up to 1024x1024, at which point self-attention begins to dominate computational costs. While MDiT exhibits slightly higher overhead at lower resolutions, it becomes more FLOP-efficient than DiT at 2048x2048.
> >
> > - **Linear Inference Scalability:** In **Appendix J.2**, we explore generating higher-resolution images using neighborhood attention and fine-tuning core patch embedding layers to 4x4. This approach retains image quality through the full latent scale outer layers and remains FFN-bound (linear) up to 2048x2048, requiring significantly fewer FLOPs than DiT.
> >
> > - **Training Scalability:** MDiT exhibits scaling behavior consistent with DiT but achieves lower overall FLOPs due to the improved training efficiency. We validate this by fitting the scaling law proposed by Henighan et al. (2020), yielding a similar scaling exponent (~-0.6) for both model families. This highlights that MDiT preserves the scalability characteristics of diffusion transformers.
> >
> > > … can it be flexibly adapted to different image resolutions and datasets?
> >
> > We explored this idea, showing MDiT is able to effectively handle higher-resolution tasks and extend to new datasets with minimal modifications:
> >
> > - **Higher Resolutions:** As shown in **Appendix I**, MDiT is capable of generating zero-shot images at higher resolutions and arbitrary aspect ratios. However, FFHQ-trained MDiT models encounter directional bias in self-attention heads at out-of-distribution resolutions (**Appendix I.4**). This issue is not observed in ImageNet-trained models. Fine-tuning specific components, such as aggregate blocks (**Appendix J.1**) or patch embeddings (**Appendix J.2**), allows for efficient adaptation to higher resolutions.
> >
> > - **Dataset Generalization:** In **Appendix C**, we demonstrate MDiT's flexibility by training an MDiT-L model on CC3M (text-to-image generation). By replacing the class embedding with text conditioning input, MDiT adapts to this new modality with minimal architectural changes, underscoring its efficiency and generalizability.
> >
> >
> > We hope this clarifies MDiT’s scalability across model sizes, resolutions, and datasets.
> >
> > ---
> >
> > Thank you for your thoughtful questions and relevant suggestions, which have helped us improve the clarity and comprehensiveness of our manuscript.
> >
> > Henighan et al. (2020). Scaling Laws for Autoregressive Generative Modeling. arXiv:2010.14701

---

### Official Review · Reviewer_JShh · 2024-11-06

**Soundness:** 2
**Presentation:** 2
**Contribution:** 2
**Rating:** 3
**Confidence:** 4

**Summary:**

This paper introduces the Multi-Scale Diffusion Transformer (MDiT), which integrates diverse, scale-specific transformer blocks into diffusion models to incorporate structural biases and enhance encoding-decoding behavior, akin to semantic autoencoders. The MDiT significantly speeds up training—up to 7x faster on ImageNet datasets—while improving image vibrancy and contrast, and reducing computational demands and memory use in image synthesis tasks.

**Strengths:**

1. The proposed MDiT has a significant speed improvement compared with the original DiT.

2. The paper incorprates many recent advencements of LLMs into diffusion transformers, yielding improved performance.

**Weaknesses:**

1. Some core contributions are overclaimed. For example, the new components like ROPE and GLU were initially introduced in large language models (LLMs). While these modules have been adopted in many recent visual models and show superior performances, MDiT is not the first to incorporate them, so its architectural design does not provide too many innovations. Also, although the paper emphasizes its processing of image features in the full latent space, the core component of the model, self-attention, actually receives downsampled representations, that is, the features only enter the attention layer after passing through pixel-shuffle as shown in equation (1).

2. The motivation of the paper is not sufficiently substantiated. DiT originally replaced the common UNet structure in latent diffusion models with plain Transformers, which, at higher image resolutions, leads to increased computation but also improves performance and scalability. This paper attempts to revert DiT back to a U-shaped structure, but no substantial benefits have been observed.

3. Important ablation studies are missing: the authors have not provided a clear ablation to show where their performance improvements come from. For instance, it is unclear which part of the model—Llama-like components or the new structural design—has a greater impact on performance.

4. Minor comments: the paper is not very easy to understand, with many details of the model described overly concise in Section 3.

**Questions:**

-- update after rebuttal --

I have carefully read other reviewers' comments as well as the authors' rebuttal. The current version of the paper still cannot convince me to give an accept. As the authors claimed, the LLM-like components are not the innovation of the paper and they foucs on hierarchical designs for Diffusion Transformers. However, the authors did not provide sufficient reasons for me to accept the view that a hierarchical structure is superior to a plain structure. In fact, latent diffusion originally utilized a hierarchical UNet, while DiT transformed it into a plain structure and discovered better scalability. This paper seems to discuss a "Revenge of UNet" concept, but there's nothing novel about the model's approach; it merely combines some recent tricks. I am not surprised that the method proposed in this paper can outperform DiT in certain scenarios, as hierarchical models are often easier to train and converge, which is true for both visual understanding and generative tasks. However, the biggest issue with the U-shape architecture is that it is difficult to scale up (e.g., in terms of parameters) and inconvenient to integrate into multiple modalities, which I consider more crucial in designing architecture. Therefore, based on my unresolved concerns, I maintain my original score.

---

> ### Author Response · Authors · 2024-11-24
> **Response to Reviewer JShh (1/4)**
>
> Thank you for your detailed review, we sincerely appreciate your feedback and for highlighting areas where our methodology can be improved. Below, we address your concerns and clarify misunderstandings.
>
> ---
>
> ## W1. Clarification of Contributions and Architectural Design
>
> > Some core contributions are overclaimed. For example, the new components like ROPE and GLU were initially introduced in large language models (LLMs). While these modules have been adopted in many recent visual models and show superior performances, MDiT is not the first to incorporate them, so its architectural design does not provide too many innovations.
>
> We thank the reviewer for pointing this out and appreciate the opportunity to clarify. RoPE and GLU were not intended to be presented as core architectural contributions of MDiT. To make this distinction clearer, we have revised **Section 3** of the manuscript to explicitly outline the key architectural contributions of MDiT, which focus on the multi-scale architecture:
>
>  - **The shallow U-Net structure:** This design incorporates distinct outer-level and core layers to effectively capture features at multiple spatial scales.
> - **Aggregate blocks:** The blocks are interleaved within the core to enhance the model’s ability to capture medium-scale details and improve structural coherence.
>
> We acknowledge that RoPE and GLU were initially introduced in LLMs and have since been incorporated into recent visual models. As the reviewer notes, these components are not novel to MDiT. While they play important roles in supporting the model’s functionality, their contributions are secondary to the multi-scale design. Specifically:
>
> - **RoPE:** Ablation studies (**Appendix F.1, Table 10**) demonstrate that RoPE provides no FID improvement and results in a minor D-FID degradation compared to sinusoidal embeddings. However, it enables zero-shot scaling to larger resolutions, which is discussed in **Appendix I**.
>
> - **GLU:** Our added ablation studies (**Appendix F**) reveal that the effectiveness of GLU depends on the conditioning strategy. DiT employs conditioning via layer norm modulation (AdaLN) and output gates on the FFN and MHSA layers. However, combining GLU with these gates introduces a destructive interaction, degrading performance. When the gates are removed, GLU helps recover the lost performance and provides further incremental gains. Importantly, the multi-scale design achieves comparable or greater improvements independently, with GLU further enhancing performance in the absence of conditioning gates.
>
> We thank the reviewer for highlighting this distinction and hope that the clarifications in the revised manuscript address any potential ambiguity. We believe that MDiT reflects a distinct architectural direction, combining the strengths of U-Net-style inductive biases with the scalability of transformers. These refinements further emphasize that MDiT’s key innovations lie in its multi-scale architectural design, supported by auxiliary components such as RoPE and GLU.

---

> ### Author Response · Authors · 2024-11-24
> **Response to Reviewer JShh (2/4)**
>
> > Also, although the paper emphasizes its processing of image features in the full latent space, the core component of the model, self-attention, actually receives downsampled representations, that is, the features only enter the attention layer after passing through pixel-shuffle as shown in equation (1).
>
> We thank the reviewer for raising this point and would like to clarify a potential misunderstanding regarding the application of full latent space processing in relation to equation (1). MDiT consists of three distinct types of blocks, each designed to process features at specific resolutions:
>
> - **Outer MDiT blocks:** These operate at full latent resolution and employ neighborhood self-attention to capture fine-grained details.
> - **Inner MDiT blocks:** These process features at a 2× downsampled resolution and incorporate both cross-attention for conditioning and full self-attention.
> - **Aggregate blocks:** These operate at the same resolution as the inner MDiT blocks (2× downsampled); however, the self-attention layer within the aggregate blocks works at a further downsampled resolution (4× from the input). This design ensures the aggregation of global representations while maintaining computational efficiency.
>
> For instance, MDiT-B employs two aggregate blocks (K=4), while MDiT-L and MDiT-XL utilize four aggregate blocks (K=8). Importantly, *self-attention in the outer blocks* is performed at the *full latent resolution*, supporting the model's ability to process image features at this scale.
>
> We believe that equation (1) in Section 3.3 may have contributed to the misunderstanding. In the revised manuscript, this section has been retitled **"Aggregate Blocks: Enhancing Structure at Medium Scales"** and revised, to more clearly describe the aggregate blocks’ behavior, including the resolution at which their self-attention layers operate.
>
> We hope this clarification resolves any potential confusion regarding the resolutions at which different attention layers operate in MDiT. Thank you again for highlighting this point.

---

> ### Author Response · Authors · 2024-11-24
> **Response to Reviewer JShh (3/4)**
>
> ## W2. Motivation and Substantial Benefits
>
> > The motivation of the paper is not sufficiently substantiated. DiT originally replaced the common UNet structure in latent diffusion models with plain Transformers, which, at higher image resolutions, leads to increased computation but also improves performance and scalability. This paper attempts to revert DiT back to a U-shaped structure, but no substantial benefits have been observed.
>
> We appreciate the reviewer’s comments and welcome the opportunity to clarify the motivation and benefits of MDiT’s architecture. Below, we summarize the key advantages provided by our approach:
>
> - **Improved training efficiency:** MDiT achieves up to a 7x speedup in training on ImageNet while maintaining competitive performance with DiT. This efficiency is particularly valuable for practical applications, as it significantly reduces computational demands without compromising image generation quality.
> - **Scalability for high-resolution generation:** MDiT’s shallow U-shaped structure enables more efficient scaling to high resolutions compared to isotropic transformer models like DiT. For instance, **Appendix J.2** demonstrates MDiT generating 512×512 images with patch finetuning and drop-in neighborhood attention. This behavior, made possible by the core’s downsampling, allows superior inference scalability up to 2048×2048 (**Appendix B.5**).
> - **Enhanced image fidelity:** MDiT improves the ability to generate fine-grained details and maintain structural coherence across scales. Although quantitative metrics (e.g., FID, D-FID, sFID) may not fully capture these improvements, **Figure 1** provides qualitative evidence. For example, the nose on the “arctic wolf” (bottom row, 3rd from the left) is exactly one latent pixel in size, where all finer details are sub-latent scale. Notably, other diffusion transformers with 2×2 patch embeddings typically rely on larger resolutions to achieve similar levels of fidelity, as they struggle to capture finer scales directly.
> - **Structural coherence:** MDiT also enhances structural consistency, as we have observed a reduction in extra-limb failures for quadrupeds during our evaluations. While this observation is anecdotal and lacks quantitative backing, we attribute this improvement to the inclusion of both aggregate blocks and variance matching regularization. Notably, the aggregate blocks operate primarily at the 1/3​ latent resolution (**Figure 3**), aligning with the approximate scale between the front and back legs of the “arctic wolf” in Figure 1.
>
> These aspects highlight the substantial benefits of MDiT compared to isotropic transformers like DiT. We believe this architecture offers a distinct balance of computational efficiency, scalability, and structural fidelity, providing a meaningful contribution to the field.

---

> > ### Author Response · Authors · 2024-11-24
> > **Response to Reviewer JShh (4/4)**
> >
> > ## W3. Ablation Studies
> >
> > > Important ablation studies are missing: the authors have not provided a clear ablation to show where their performance improvements come from. For instance, it is unclear which part of the model—Llama-like components or the new structural design—has a greater impact on performance.
> >
> > We appreciate the reviewer’s suggestion and have expanded our ablation studies in the revised manuscript to provide a clearer breakdown of the contributions to performance improvements. A summary table of these results is now included in **Section 5.2**, and detailed experiments are provided in **Appendix F**.
> >
> > Our ablation studies confirm that the multi-scale architecture is the primary driver of performance improvements, consistently outperforming configurations without these components. For example, adding only the outer and aggregate blocks to the baseline - without GLU, Cross-Attention, or RoPE - improves FID by **7.85 points** and D-FID by **119 points**, demonstrating the independent effectiveness of the multi-scale design.
> >
> > In contrast, the contributions of the LLaMA-like components (removing bias terms, RMS Norm, GLU, and RoPE) are mixed. Some, such as RoPE, have minimal or no significant effect, while others, such as GLU, depend on the conditioning strategy. Specifically, GLU improves performance when the conditioning gates from DiT-style blocks are removed; however, combining GLU with conditioning gates introduces a destructive interaction that degrades performance.
> >
> > Ablations show that removing the conditioning gates *reduces performance* by approximately **12 points** in FID and **145 points** in D-FID. Adding GLU to this configuration results in a net improvement of **7.35 points** in FID and **102 points** in D-FID compared to the baseline. Conversely, retaining the conditioning gates while also adding GLU *degrades* FID by **0.89 points** and D-FID by **7 points**.
> >
> > These expanded studies demonstrate that the multi-scale design provides the most consistent and significant gains, with GLU offering complementary improvements when paired appropriately.
> >
> > We hope this clarification addresses the reviewer’s concerns and provides a clearer understanding of the contributions from different architectural components to MDiT’s overall performance.
> >
> > ---
> >
> > ## W4. Presentation and Clarity
> >
> > > The paper is not very easy to understand, with many details of the model described overly concise in Section 3.
> >
> > We appreciate the reviewer’s feedback regarding **Section 3** and have revised this section to improve clarity while highlighting MDiT's architectural contributions.
> >
> > ---
> >
> > Thank you once again for your thoughtful comments and suggestions, which have helped us refine our manuscript and address potential ambiguities.

---

### Official Review · Reviewer_x7Ca · 2024-11-13

**Soundness:** 2
**Presentation:** 1
**Contribution:** 2
**Rating:** 3
**Confidence:** 5

**Summary:**

This paper proposes a multitude of modifications to diffusion models for image generation that are based on vision transformers.  A primary focus is on architectural details of the transformer network which is being trained to denoise.  A combined result of these modifications is faster training for similar generation quality in comparison to baseline diffusion models.

**Strengths:**

Explores many design choices inherent in diffusion transformers, including both model architecture and regularization.

Extensive experiments, including thorough documentation in the appendix, quantify the consequences of the proposed changes.

The cumulative result of the many modifications is faster training and faster inference (fewer sampling steps) for similar quality image generation (measured by FID on ImageNet) as recently published diffusion-based image generators.

**Weaknesses:**

The many tweaks to model architecture and regularization seem to be individually minor and not obviously related to each other.  These tweaks include: patch resolution (1x1 vs 2x2 embedding), time embedding as a token, cross-attention for class conditioning, rotary positional embeddings, layer normalization on Q and K vectors, number of attention heads, and a variance-matching regularization term in the loss.  While the paper provides individual justifications for these design elements, there is an absence of high-level motivating principles to explain them.  The different tweaks are not obviously related, giving the impression that the chosen collection of changes may be akin to a product of architecture search.

There appears to be a disconnect between the analysis and the claims of explainability put forth in the title and introduction.  It is unclear why this architecture is any more or less explainable than existing transformer-based diffusion models.  While Section 4 presents a story about explainability based on probing network behavior, it is unclear whether this post-hoc analysis yields any predictive power or is a heuristic that supports confirmation bias.

In terms of presentation, the main text needs to better communicate what impact each architecture design decision has on the overall system (both in isolation and, if there are dependencies, in combination with others).  Perhaps these messages are buried in the 28 pages of appendices with additional text, tables, and figures.  However, it is not presented in an accessible manner in the main paper, reinforcing the impression that results stem from a collection of engineering tweaks, rather than fundamental insights.

**Questions:**

Is there a coherent motivation connecting the many proposed architectural changes?  How large of a design space was explored to arrive at the chosen configuration?  Do the claims regarding explainability yield any predictive power in terms of model behavior or design trade-offs (and if so, can you provide experimental validation)?

---

Post-rebuttal: As explained in the discussion below, the author response does not resolve my concerns.

---

> ### Author Response · Authors · 2024-11-24
> **Response to Reviewer x7Ca (1/4)**
>
> Thank you for the thoughtful and constructive feedback, as well as for raising several interesting and thought-provoking questions. Your insights have been valuable in helping us improve and clarify our work.
>
> ---
>
> ## W1. Architectural Motivation
>
> > Is there a coherent motivation connecting the many proposed architectural changes?
>
> We appreciate the reviewer’s thoughtful question and acknowledge that the presented architecture may initially appear to result from an extensive architecture search.
>
> While MDiT incorporates insights gained through exploratory design, this exploration was hypothesis-driven and guided by clear goals to address the limitations of isotropic diffusion transformers like DiT. Specifically, the design choices were motivated by the following principles, as outlined in our revised **Section 3**:
>
> 1. **Parameter efficiency:** Reducing unnecessary parameters while maintaining or improving performance.
> 2. **Multi-scale feature representation:** Addressing the inherent isotropic nature of diffusion transformers to better capture features across scales.
> 3. **Flexibility:** Enabling the architecture to adapt to diverse modalities, including text conditioning and zero-shot aspect ratio changes.
>
> In addition to these goals, ensuring stability during training was a critical consideration throughout the design process. Architectural components such as normalized Q and K vectors (as is now the de-facto standard in vision models) proved essential for preventing divergence or instability during training.
>
> Each of the design choices supports one or more of these principles:
>
> 1. **Parameter efficiency:**
>     - The use of LLaMA blocks rather than DiT blocks significantly reduces parameters by removing biases and a large portion of AdaLN modulation.
>     - The time embedding token removes redundancies in the blocks with cross-attention, reducing unnecessary parameters (3× Ada-scale + CA > 1× token projection).
> 2. **Flexibility:**
>     - Cross-attention for class conditioning provides straightforward support for text-based conditioning, as demonstrated in **Appendix C**.
>     - Rotary position embeddings (RoPE) enable zero-shot aspect ratio and scale changes, as discussed in **Appendix I**.
> 3. **Multi-scale feature representation:**
>     - Changing the patch resolution (from 2×2 to 1×1) allows the model to resolve fine details that are otherwise unresolvable with larger patch embeddings.
>     - Variance matching regularization provides more informative gradients for structural coherence.
>
>
> The overarching narrative of MDiT is further clarified in the revised **Section 3**, which positions the architecture as a testbed to explore the guiding question posed in the introduction: *“Can such biases be explicitly reintroduced to diffusion transformers while maintaining their generality and enhancing training efficiency?”*
>
> We hope this clarifies the high-level design principles that guided MDiT’s development. As the reviewer notes, individual choices were justified in the paper. This response, along with the new revision, highlights how they collectively support the overarching goals of the architecture.
>
>
>
> > How large of a design space was explored to arrive at the chosen configuration?
>
> The explored design space was larger than what is presented in the paper, as part of our effort to build intuition for the core model behavior. Notably, much of this exploration was conducted using text-to-image (T2I) models trained on MS-COCO, a simple yet representative task, with training stopped at 50k steps. This allowed us to evaluate configurations visually (via generated samples and activation inspection) before committing to full-scale evaluations.
>
> This exploration led to key insights, including that isotropic transformers often exhibited a semantic-autoencoder-like behavior and that the capacity of the patch embedding layer played a significant role in this dynamic. These observations informed several decisions, such as the use of cross-attention for class conditioning to remain consistent with the text-based modality.
>
> It is worth noting that the first configurations subjected to FID evaluations were the *parallel* and *serial* variants (**Table 10** in **Appendix F.1**), which essentially served as an “unblinding” to the broader exploratory process. From this stage onward, our design decisions were evaluated more rigorously to ensure alignment with the overarching goals outlined in **Section 3**.
>
> We would be happy to provide additional details about this process if the reviewer wishes.

---

> ### Author Response · Authors · 2024-11-24
> **Response to Reviewer x7Ca (2/4)**
>
> ## W2. Connection with Explainability
>
> We thank the reviewer for raising this important point and for their detailed observations regarding the role of explainability in our work. We recognize that the causal relationship between explainability and performance optimization may not have been fully clear.
>
> The primary role of the explainability framework in this work is to provide insights into the network’s depth-wise functional behavior, enabling architectural optimizations that improve training efficiency and image synthesis quality. While the title could be interpreted to suggest the reverse causal relationship (i.e., explainability inherently improving training), this is not the intent of the work. Instead, explainability was used as a tool to guide the design process, consistent with the contributions outlined in the introduction.
>
> With that said, we believe MDiT is inherently more explainable than other isotropic diffusion transformers for several reasons:
> - **Multi-scale architecture:** The U-Net-style design suggests that features are processed at specific spatial resolutions, rather than being superimposed at a single scale. For example, in isotropic models with 2×2 patch embeddings, finer 1×1 features must be encoded alongside others within the same token. MDiT’s multi-scale design mitigates this by compartmentalizing features across resolutions (supported by **Appendix M**).
> - **Rotary position embeddings (RoPE) and layer normalization:** These components enable and support the explainability analyses in Section 4. RoPE introduces interpretable spatial priors, while layer normalization ensures consistent scaling (L2 norm) across attention heads, facilitating meaningful comparisons between blocks. Without these elements, this additional explainability method would not be possible.
> - **Removal of feature suppression gates:** By removing the gates from DiT blocks, MDiT prevents the use of an explicit feature suppression mechanism, such as learned zero modulation. Instead, removing these gates encourages active processing at all timesteps without relying on explicit gating.
>
> These aspects collectively make MDiT more interpretable and amenable to explainability techniques compared to isotropic transformer-based diffusion models. While these benefits were not the primary focus of the work, they represent an inherent advantage of the architecture.

---

> ### Author Response · Authors · 2024-11-24
> **Response to Reviewer x7Ca (3/4)**
>
> > Do the claims regarding explainability yield any predictive power in terms of model behavior or design trade-offs (and if so, can you provide experimental validation)?
>
> This is an excellent question, and we agree that demonstrating predictive power is critical to validating the utility of our explainability framework.
>
> ---
>
> ### 1. Predictive Power for Model Behavior and Design Trade-offs:
>
> One example of predictive power comes from the analysis presented in **Figure 5b and 5c**, where we predicted that introducing a scale separation between fine-grained (1×1 latent) and coarser (2×2 downsample) resolutions would shift feature extraction behavior to the finer scale. This behavior mirrors the hierarchical processing seen in conv-nets, such as edge detection at initial layers. Indeed, this prediction was validated by comparing configurations like {0,0,0,12} and {2,4,0,9}. However, this results in increased FLOPs as finer scales, processing the full sequence length $H\times W$ rather than the downsampled $H \times W / 4$.
>
> To mitigate this, we introduced neighborhood attention for fine-grained processing, predicting that a relatively small kernel size (e.g. $k=7$) would be sufficient for extracting low-level features. This choice, derived from the insights provided by our explainability analysis, was confirmed to be effective.
>
> ---
>
> ### 2. Experimental Validation via Destructive Testing:
>
> We further validated the predictive power of the explainability framework through destructive testing, detailed in **Appendix M**. In these experiments, we systematically ablated architectural components (e.g., positional encodings, entire blocks, and the neighborhood attention kernel size) and verified that the visual impacts aligned with predictions made by interpreting the complex magnitude plot from **Figure 5c**. This analysis focused on the {2,4,0,9} configuration, allowing us to test the framework’s ability to explain and anticipate changes in model behavior.
>
> The results confirmed that the observed outcomes matched the predictions in all tested cases. For example:
> - Ablating positional encodings in certain blocks had minimal impact for semantically focused blocks but significantly affected composition for position-focused blocks.
> - Reducing the neighborhood attention kernel size produced varying effects, depending on how far the position focus extended within each block (e.g., channels 0–2, 0–4, 0–6).
>
> These findings suggest actionable architectural design choices, such as selectively removing position embeddings in certain blocks to increase capacity, and reducing the neighborhood kernel size to lower FLOPs. While we did not re-train or fine-tune the model to verify these changes due to limited resources, the destructive testing experiments demonstrate that the explainability framework offers hypothesis-generating insights into architectural design and potential trade-offs.
>
> ---
>
> ### 3. Broader Applications of Attention Probing:
>
> Beyond MDiT, the attention probing framework has broader predictive applications. For instance, it offers insights into the interaction between RoPE and the self-attention mechanism's ability to balance positional and semantic information. This was demonstrated in experiments with LLaMA-3 models (**Appendix K**), where predictions about RoPE behavior at longer context lengths were validated.
>
> Specifically, the training context length establishes an effective $r_{dim}$ boundary: frequencies below this threshold are effectively ignored by the model, as they were never utilized during training. As expected, when comparing the base model against a fine-tuned version trained on longer contexts, differences in the attention probe complex magnitude were minimal at higher frequencies (shorter distances) but increased at lower frequencies (longer distances). These results offer insights into how training context length shapes the effective $r_{dim}$ and its interaction with positional encoding.
>
> ---
>
> Collectively, these results demonstrate that the explainability framework provides predictive insights into model behavior and design trade-offs. While certain predictions remain exploratory, the strong alignment between predicted and observed outcomes validates the framework’s ability to guide architectural decisions and improve our understanding of both transformer-based diffusion models and transformers more broadly, as evidenced by the LLaMA results.

---

> > ### Author Response · Authors · 2024-11-24
> > **Response to Reviewer x7Ca (4/4)**
> >
> > ## W3. Communicating Architectural Choices and Dependencies
> >
> > We appreciate the reviewer’s feedback regarding the presentation of the architectural design decisions and their impacts.
> >
> > To improve accessibility, we have included a summary ablation table in **Section 5.2** that highlights the performance contributions of the primary components, such as LLaMA blocks, RoPE, cross-attention (MDiT blocks), and the multi-scale design. This table provides a high-level overview of the relative impact of each component, with more detailed ablations and analysis presented in **Appendix F**.
> >
> > In **Appendix F**, we further explore potential interdependencies between components. While many adjustments appear to impact performance independently (e.g., the multi-scale design), we identified a notable dependency between the DiT condition gates, GLU FFN, and cross-attention:
> >
> > - Removing conditioning gates while retaining the baseline GeLU FFN leads to a significant performance degradation, which can be mitigated by adding cross-attention (less effective) or using a GeGLU FFN (more effective).
> > - Adding a GeGLU FFN alongside the conditioning gates results in degraded performance, likely due to a destructive interaction (both apply direct feature modulation).
> > - This finding suggests that the conditioning gates and GeGLU FFN function as direct feature suppression mechanisms, while cross-attention provides this functionality indirectly.
> > - Notably, GeGLU without conditioning gates represents the minimal configuration within this interdependency to achieve significant gains over the baseline.
> >
> > While our revisions aim to prioritize clarity and focus in the main text, we recognize that presenting all design decisions in detail could distract the reader from other critical content. The updates to **Section 3**, along with the addition of **Section 5.2** and to **Appendix F**, strike a balance by providing both a high-level overview and deeper technical insights for readers seeking further details. We hope these updates address the reviewer’s concern and provide greater clarity regarding the contributions and impacts of the architectural decisions.
> >
> > ---
> >
> > Thank you again for your thoughtful and constructive feedback, which has been invaluable in improving the quality and clarity of our work.

---

> > > ### Comment · Reviewer_x7Ca · 2024-12-03
> > > **Re: Response to Reviewer x7Ca**
> > >
> > > Having read the other reviews and author response, I remain unconvinced that there is sufficient innovation here. Rather, extensive empirical tweaking of the architecture seems to be the focus, but the individual changes are not particularly novel or interesting. I concur with Reviewer JShh, who makes similar points (e.g., that components like ROPE are borrowed from elsewhere).
> > >
> > > The author response actually highlights, in some respects, the degree to which the architectural optimization is focused on minutiae. For example, multi-scale (a design characteristic the paper promotes as important) is about "changing the patch resolution (from 2×2 to 1×1)". Sure, this might improve performance, but it is hardly an interesting or exciting concept and has zero novelty. Many prior works have explored multi-scale neural architectures in more expansive settings (e.g., spanning a vast range of spatial scales).
> > >
> > > Similarly, there does not seem to be any new advance in explainability (contrary to what the title might suggest), but rather a notion that particular chosen components (e.g., ROPE, removal of gating mechanisms) are inherently more explainable. Given the present limitations of tools for interpreting network behavior, this is at best debatable. However, even if one accepts this argument, the paper is not introducing any novel components or explainability approaches.
> > >
> > > I appreciate the additional experimental summaries offered in the updated appendix, but also maintain my original view that, with the primary contribution being a collection of engineering tweaks, far more work needs to be put into the presentation in order to clearly communicate the key choices in the main text (rather than 55 pages of appendix, a significant portion of which is additional text).

---

> > > > ### Author Response · Authors · 2024-12-04
> > > > **Response to Reviewer x7Ca**
> > > >
> > > > We sincerely thank Reviewer x7Ca for their response and for providing further perspective.
> > > >
> > > > We acknowledge that the reviewer appears to perceive the primary contribution as stemming from the collection of auxiliary components (e.g. RoPE). However, as noted by the reviewer and in Section 3.1, this integration is not novel and was therefore not intended to represent the main contributions of our work. Rather, these components help establish an efficient starting point with properties that augment the following contributions:
> > > > 1. A novel *heterogeneous multi-scale* diffusion transformer architecture.
> > > > 2. The application of a new *explainability technique* that both optimizes and explains the multi-scale design.
> > > > 3. The introduction of a *variance matching regularization scheme*, which improves image quality and provides spatially informative gradients.
> > > >
> > > > These efforts enable significantly faster ImageNet training while the auxiliary components provide secondary benefits (e.g., conditioning flexibility, parameter efficiency, and zero-shot spatial extrapolation). These secondary benefits are thoroughly detailed in the appendix for reproducibility and transparency, allowing the main text to focus on the intended core contributions.
> > > >
> > > > ---
> > > >
> > > > ## Architectural Contribution
> > > >
> > > > We appreciate the reviewer’s feedback and would like to clarify the key architectural contributions of our work: the shallow U-Net and Aggregate blocks. While RoPE was included in our design, it was discussed solely in the context of its role in aiding the attention probing analysis and zero-shot extrapolation. As shown in Section 5.2 and Appendix F, RoPE neither impacts metric performance nor is necessary for the multi-scale architecture, *consistent with its auxiliary nature*.
> > > >
> > > > Regarding patch resolution, we apologize for any misunderstanding. Our response directly addressed the reviewer’s list of architectural changes and clarified that changing patch resolution is not an isolated tweak but *part of the multi-scale motivation*. Notably, this change alone would result in an uninteresting isotropic network.
> > > >
> > > > Our key architectural contributions, described in Section 3 and illustrated in Figure 2, collectively form the multi-scale architecture:
> > > > - The **shallow U-Net** introduces a finer-grained resolution beyond the standard isotropic case: covering **both** 1x1 and the typical 2x2 patch resolution levels.
> > > > - The **Aggregate blocks** provide an efficient mechanism for coarse feature processing (4x4 level) without the computational overhead of adding deeper U-Net levels.
> > > >
> > > > We acknowledge the reviewer’s point that similar multi-scale architectures exist in convolutional U-Nets and classification vision transformers (discussed in Section 1). However, to our knowledge, the successful and efficient integration of these concepts into diffusion transformers *is novel*. Prior works have relied either on isotropic networks or full multi-level symmetric U-Nets, both of which use larger input patch embeddings and omit the full input resolution scale. In contrast, we demonstrate that incorporating this outermost resolution scale into the hierarchy significantly improves training efficiency.
> > > >
> > > > ---
> > > >
> > > > ## Explainability Contribution
> > > >
> > > > We appreciate the reviewer’s perspective and their acknowledgment of the limitations of current tools for interpreting network behavior. Our attention probing analysis (Section 4) provides a valuable lens for understanding where the transformer focuses on spatial and semantic features. This level of detail is not easily obtainable using existing methods and helps to contextualize the spatial behavior of RoPE-based transformers.
> > > >
> > > > The utility of our explainability technique is demonstrated through:
> > > > - The observed shift in short-range feature focus when adding the outer U-Net level (Section 4.1), highlighting the importance of capturing *both* typical and full input resolutions.
> > > > - Its ability to provide insights into zero-shot extrapolation failures in LLMs (Section 4.1, Appendix K).
> > > > - The experimental predictions detailed in Appendix M, complementing the discussion in Section 3.2 and acknowledged by the reviewer.
> > > >
> > > > This analysis extends the current toolkit for analyzing network behavior and provides new insights into transformer models.
> > > >
> > > > ---
> > > >
> > > > ## Practical Contribution
> > > >
> > > > In addition to the architectural and explainability insights, our work offers a significant practical contribution. As noted in the reviewer’s initial review, we have extensively quantified the effects of various architectural components and demonstrated faster training. This advances progress toward overcoming the training bottleneck in diffusion models. At a minimum, we believe our work establishes a new baseline for efficient training and provides a foundation for future research to build upon and compare against.
> > > >
> > > > ---
> > > >
> > > > We sincerely appreciate the reviewer’s time and feedback, which provided us with the opportunity to clarify and better emphasize the contributions of our work.

---

### Author Response · Authors · 2024-11-24
**Response to All Reviewers**

We sincerely thank the reviewers for their thoughtful feedback and constructive comments, which have greatly helped us refine our work. Below, we provide an overview of the key revisions made to the paper in response to the feedback received:

- **Section 3:** Revised to better communicate the motivation of our architecture, the core architectural contributions, and to improve readability of the model design [x7Ca, JShh].
- **Section 5.2 “Architectural Ablations”:** Added a summary of how each of the major model changes impacts performance [x7Ca, JShh, GJVN].
- **Appendix F “Additional Ablations”:** Updated to provide more detailed ablation steps, helping isolate potential interdependencies between architectural choices [x7Ca, JShh, GJVN].
- **Appendix B.5 “Computational Scaling Behavior”:** Added to explore the scalability of MDiT in comparison to DiT, both for image resolutions and training FLOPs [GJVN].
- **Appendix G.6 “Guidelines for Parameter Selection in MDiT”:** Added to clearly list the architectural hyperparameter guidelines we used to simplify the MDiT experiments [d2qx].
- **Appendix M “Validating Explainability through Destructive Testing”:** Added to verify the predictive power of the explainability analysis and directly connect these results to architectural implications [x7Ca].

These revisions address the primary concerns and suggestions raised by the reviewers, and we hope they clarify and strengthen the contributions of our work.

Thank you again for your detailed reviews and thoughtful engagement with our submission. We look forward to further feedback during this discussion phase.

---

### Meta-Review · Area_Chair_jykb · 2024-12-16

**Metareview:**

This paper proposes a novel multi-scale transformer architecture and regularization techniques for generative diffusion models, which leads to significantly faster training as compared to single-scale architectures.
Strengths reported by the reviewers include the novel architecture, it's ability to accelerate training, and extensive experiments.
Reported weaknesses include limited novelty of each proposed change wrt baselines, and lack of clarity of the contribution of each component of changes wrt baseline architectures, and a small number of datasets for experiments.

**Additional Comments On Reviewer Discussion:**

In response to the reviews the authors provided an extensive rebuttal and submitted an updated version of the manuscript with additional experimental results. This additional material has only in part been able to address the reviewer concerns, in particular regarding the novelty of the presented approach. Additional scaling experiments and ablations are useful, but unfortunately measured at points in training far from the best performance. The final recommendations are mixed (two reject and two (marginal) accept). The two reviewers with negative recommendations acknowledged the rebuttal, but maintained their recommendation. Taking into account all the material from the reviews, and responses, it seems that the paper can still be significantly improved by focussing on the improved training and inference speed, and reorganizing the content and taking more of the quantitative results to the main paper, and deferring some of the analysis to the supplementary, while also shortening the latter.

---

### Decision · Program_Chairs · 2025-01-22

Reject